# Self-driven electrical triggering system activates tunneling nanotube highways to enhance drug delivery in bladder cancer therapy

Zhijun Liu[1,2,3], Ravindra Joshi [4,5], Zhongguo Zhou[6], Fulin Liu [7,8], Ying Gong[1], Mingyan Sun[1], Xiuxiu Li[1], Tao Jiang[1], Liang Zou[9], Siyuan Wang[10], Yi Shi[7,8,11] ✉, Zong-Hong Lin [4,5] ✉ & Yang-Bao Miao [1] ✉

Bladder cancer ranks among the most prevalent malignancies affecting the urinary system worldwide. Despite advances in treatment, poor drug penetration and uncontrolled release continue to impede the effectiveness of chemotherapy for this disease. To overcome these obstacles, we have developed a self-driven electrical triggering system, which leverages intravesical pressure to produce electricity. This electronic trigger system can rapidly transport hydroxycamptothecin through tunneling nanotubes, acting as a high-speed channel, thereby enhancing the drug absorption by tumor cells. Additionally, the voltage generated by this system effectively induces reactive oxygen species (ROS), further promoting the eradication of bladder cancer cells. In orthotopic female animal models of bladder cancer, our results indicate that an intravesical pressure-driven system in the bladder generates electricity to facilitate drug release and rapid diffusion through a tunneling nanotube highway, while also effectively generating ROS to eliminate bladder cancer cells. This self-driven electrical trigger system, coupled with a tunneling nanotube highway to transport drugs, offers renewed hope for bladder cancer treatment. With its potential to transform current therapeutic approaches, this system is poised for deeper exploration in research and clinical settings.

Bladder cancer ranks as the fourth most prevalent cancer and is projected to be the ninth leading cause of cancer-related mortality among men[1,2]. In 2023, bladder cancer is expected to claim the lives of 16,710 individuals in the United States, constituting 2.7% of all cancer-related deaths[3]. Presently, intravesical chemotherapy stands as a primary therapeutic modality for managing bladder cancer[4]. Despite its effectiveness, intravesical chemotherapy is accompanied by a spectrum of side effects, encompassing urinary tract irritation, infections, and cystitis[5,6]. Such adverse effects not only pose the risk of bladder cancer recurrence but also potentially exacerbate disease progression[7,8].

Primarily, these challenges stem from the limitations of drug targeting and inadequate penetration[9]. Therefore, achieving effective delivery of permeability-enhancing anti-cancer drugs into bladder cancers is crucial for improving the efficacy of intravesical bladder cancer treatment[9–11].

The burgeoning advancement of nanotechnology has ushered in a new era for enhancing drug efficacy in anti-tumor therapy[12]. Presently, researchers are dedicated to addressing the biomedical challenges associated with traditional anti-tumor methodologies and spearheading the exploration and development of nanomedicine carriers and

novel nanomedicines[13–15]. These endeavors offer substantial promise in augmenting efficacy while mitigating toxicities and side effects.

Among the emerging systems under investigation, piezoelectric bionanomaterials, encompassing ferroelectric, piezoelectric and pyroelectric materials, have garnered considerable attention in anti-tumor applications[16–19]. By harnessing forces, light, magnetism, or thermal and electrical stimuli, these materials generate polarized charges at a microscopic level, establishing a piezoelectric potential and an inherent electric field[20–24]. Grounded in piezoelectric and pie-zoelectric photoelectric theories, the polarized charges can directly target substances within the tumor microenvironment, such as $H_2O$, instigating the generation of copious reactive oxygen species (ROS) during charge polarization, thereby efficaciously eradicating tumor cells[17,25–27]. Concurrently, the polarization charge from piezoelectric materials aids in carrier separation and impedes recombination[28,29]. Moreover, the generation of electric charges assists in overcoming physiological barriers within organisms, such as tight epithelial cell layers, thus furnishing a robust theoretical framework for the efficient treatment of bladder cancer[30,31].

Hence, we propose a self-driven electrical triggering system (BTO-CPT/FA) (Fig. 1). The integration of piezoelectric particles, such as barium titanate (BTO), hydroxycamptothecin (CPT)[32,33], and a targeting agent, folic acid (FA)[34], underpins this system's potential to revolutionize bladder cancer treatment. This system is carefully designed to enhance the delivery and absorption of CPT through tunneling nanotubes by utilizing the self-generated electrical system induced by intravesical pressure. This targeting approach not only ensures precise drug release but also evidently improves therapeutic outcomes.

Additionally, this system's ability to generate ROS under con-trolled voltage further enhances its efficacy in eliminating bladder cancer cells[35]. In preclinical studies using animal models of orthotopic bladder cancer, this approach demonstrated that the improved drug penetration, release, and reactive ROS generation were achieved through the electrically triggered enhancing transport of tunneling nanotube highways, which led to a notable reduction in tumor pro-gression. The self-generating electrical trigger system, featuring high-speed drug delivery through tunneling nanotubes and the production of ROS via self-driven electricity, presents a promising breakthrough for fighting against bladder cancer. This approach offers renewed optimism for improving treatment outcomes and enhancing the quality of life for bladder cancer patients globally.

## Results

### Characterization of the self-driven electrical triggering system (BTO-CPT/FA)

To attain the requisite attributes for effective cancer treatment, pre-cise control over nanoparticle size is imperative. In this investigation, we synthesized BTO NPs via hydrothermal reaction between Ba(OH)$_2$ and tetrabutyl titanate, yielding nanoparticles with a particle size of less than 100 nm. Figure 2a depicts a schematic diagram illustrating the synthesis process of the BTO-CPT/FA nanoplatform. It's well-established that smaller nanoparticles exhibit enhanced internaliza-tion by cancer cells owing to their size and surface characteristics[36,37]. Moreover, nanoparticle sizes ranging from 50 nm to 200 nm facilitate tumor targeting[38]. Additionally, the piezoelectric properties of BTO nanoparticles (NPs) can be modulated by temperature variation. To enhance the piezoelectric characteristics of BTO NPs, tetragonal BTO NPs with superior piezoelectric properties were obtained after being performed at 800 °C for 5 hours. The surface of the tetragonal BTO NPs became hydrophilic after treatment with hydrogen peroxide solution (Supplementary Fig. 1). The hydrophilic surface by modified with hydroxylation to establish a basis for chemical reaction. Following this, the BTO-CPT/FA nanoplatforms were synthesized via a two-step Mitsunobu reaction.

Schematic illustration of CPT and FA combined on the surface of tetragonal phase BTO after Mitsunobu reaction (Fig. 2b). In the SEM and TEM images (Fig. 2c, d), the average size of BTO NPs measures approximately 72.8 ± 14.9 nm (width), 94.0 ± 14.0 nm (length), respectively, exhibiting a distribution of monodisperse properties. Further analysis with Image J revealed an aspect ratio varying from 1.1 to 1.4, indicating the lack of central symmetry in the piezoelectric material—an essential characteristic for achieving its piezoelectric properties[39] (Fig. 2e).

To enable efficient drug attachment, the surface of the piezo-electric BTO NPs was rendered hydrophilic through an oxidation process. Subsequently, the drug (CPT) and the targeting agent, FA-PEG$_{2000}$-COOH, were covalently attached via the Mitsunobu reaction, forming a self-triggered electrical system (BTO-CPT/FA). The energy dispersive X-ray (EDX) spectroscopy elemental mapping image dis-tinctly illustrated the uniform dispersion of Ba, Ti, and O elements within the BTO-CPT/FA (Fig. 2f, g).

Dynamic light scattering (DLS) analysis indicates that the hydro-dynamic size of BTO-TCP/FA in deionized water is approximately 150 nm (Fig. 2h). This size is discernibly larger than the nanoparticle radius observed via SEM or TEM, likely due to the presence of a hydration layer, which is consistent with the results observed for BTO alone[40]. The findings also show no detectable size change after adding the small-molecule drug CPT and targeting agents. However, a notable change in Zeta potential was observed, transitioning from −34.9 ± 0.6 mV for BTO to −28.7 ± 0.9 mV after CPT modification, and subsequently to −27.1 ± 1.1 mV upon FA-PEG$_{2000}$-COOH modification.

To confirm that the chemotherapeutic agent and targeting reagent were loaded on the BTO, characterization of UV-vis spectro-photometry and FT−IR spectroscopy was performed. UV-vis spectro-photometry revealed an absorption peak at 384 nm, characteristic peak of CPT, confirming its attachment to the BTO NPs (Fig. 2i). To optimize the loading concentrations of CPT and FA in the BTO-CPT/FA nanoplatform, thermogravimetric analysis (TGA) was employed (Supplementary Table 1 and Supplementary Fig. 7). Based on the economic feasibility and loading efficiency, we selected the synthesis condition with number "②" that is "0.01 mmol CPT and 100 mg BTO", the mass fraction of CPT and FA on BTO-CPT/FA was 6.2%, 5.7%, respectively, and the BTO-CPT/FA nanoplatform was prepared based on the ratio. Furthermore, the loading rate of Cy5.5 on BTO was 8.9% through UV-vis spectrum (Supplementary Fig. 8).

FT−IR analysis verified the successful formation of BTO-CPT and BTO-CPT/FA through covalent conjugation between hydroxyl-functionalized BTO and the CPT and FA. In the case of BTO-CPT, the appearance of a characteristic ether bond (C−O−C) stretching peak at 1150 cm$^{-1}$, along with the disappearance of the H−O−H bending vibration peak at 1630 cm$^{-1}$, confirmed covalent bonding between CPT and BTO NPs. Moreover, the persistence of the lactone C=O peak at 1650 cm$^{-1}$ indicated that CPT was chemically conjugated rather than physically adsorbed. For BTO-CPT/FA, the emergence of a distinct ester carbonyl (C=O) peak at 1730 cm$^{-1}$, intensified C−O−C vibrations at 1250 cm$^{-1}$, and the disappearance of the free carboxylic acid peak at 1700 cm$^{-1}$ collectively confirmed the esterification reaction between FA-PEG-COOH and BTO NPs (Fig. 2j).

To investigate whether CPT binds to BTO via chemical coupling, the ζ-potential of BTO nanoparticles and CPT in acetone was measured (Supplementary Fig. 2a). The near-zero ζ-potential value indicated minimal electrostatic interaction between the two, suggesting that electrostatic forces play a negligible role in the binding process. Fur-thermore, although the absorbance of the supernatant decreased meaningfully after two washing steps (Supplementary Fig. 2b), the characteristic absorbance of CPT at 384 nm in the precipitate remained largely unchanged (Supplementary Fig. 2c), with over 85% of CPT retained even after seven washing cycles (Supplementary Fig. 2d). These results suggest that CPT is not merely adsorbed through

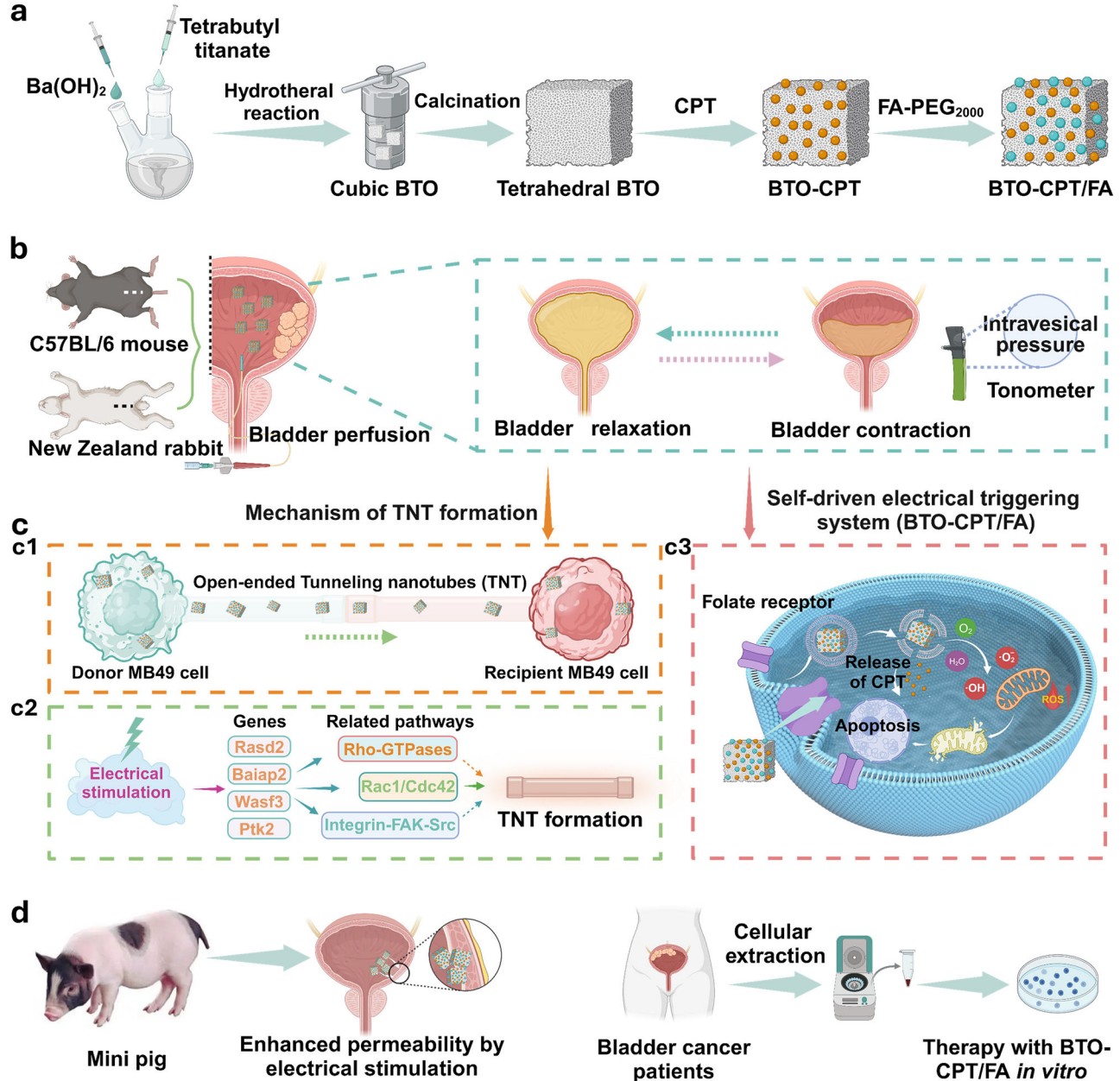

**Fig. 1 | Construction and operating mechanism of the self-driven electrical triggering system (BTO-CPT/FA) for orthotopic bladder cancer treatment.**
**a** Schematic illustration of the preparation of the BTO-CPT/FA nanoplatform. **b** In orthotopic bladder cancer animal models (C57BL/6 mouse or New Zealand rabbit), BTO-CPT/FA for tumor therapy through bladder perfusion. **c** The delivery mechanisms of nanodrug driven by piezoelectric effects: the formation of a high-speed transport pathway between bladder cancer cells via tunneling nanotubes with a structure of open-ended (**c1**), the related genes involved in the formation of tunneling nanotube under electrical stimulation (**c2**), and BTO-CPT/FA leverages intravesical pressure-induced spontaneous electric fields to release CPT and facilitate drug transport through tunneling nanotubes, enhancing the delivery and penetration of CPT, meanwhile, the generated ROS also contribute to tumor cell apoptosis (**c3**). **d** An enhanced nanodrug permeability in the minipig bladder mediated by piezoelectric effects after treatment with BTO-CPT/FA (left), and the marked inhibitory effects of BTO-CPT/FA for bladder cancer cells from clinical patient samples (right). Figure created in BioRender. Xixi, L. (2025) https://BioRender.com/x1i5irp.

nonspecific interactions but is likely anchored onto the BTO surface through stable chemical coupling. In addition, the large specific surface area of BTO facilitates initial CPT loading prior to washing.

To verify the piezoelectric properties, the crystal structures of these NPs were examined using X-ray Diffraction (XRD), showing the patterns were consistent with simulations, notably, a distinct split peak appears between 44° and 46°, indicating that the BTO possesses a tetragonal phase structure (PDF#05-0626)[41,42], which is favorable for generating piezoelectric properties (Fig. 2K, l).

## Performance characterization of self-driven electrical triggering system

To evaluate the piezoelectric response of the synthesized particles, BTO-CPT/FA samples were securely mounted on an electro-mechanical measurement device for mechanical cycling tests (Fig. 3a). The measurement system consisted of a piezoelectric module, a linear reciprocating motion platform, and an electrical signal detection workstation (Supplementary Fig. 3). Upon initiation of linear reciprocating motion, voltage signals generated by

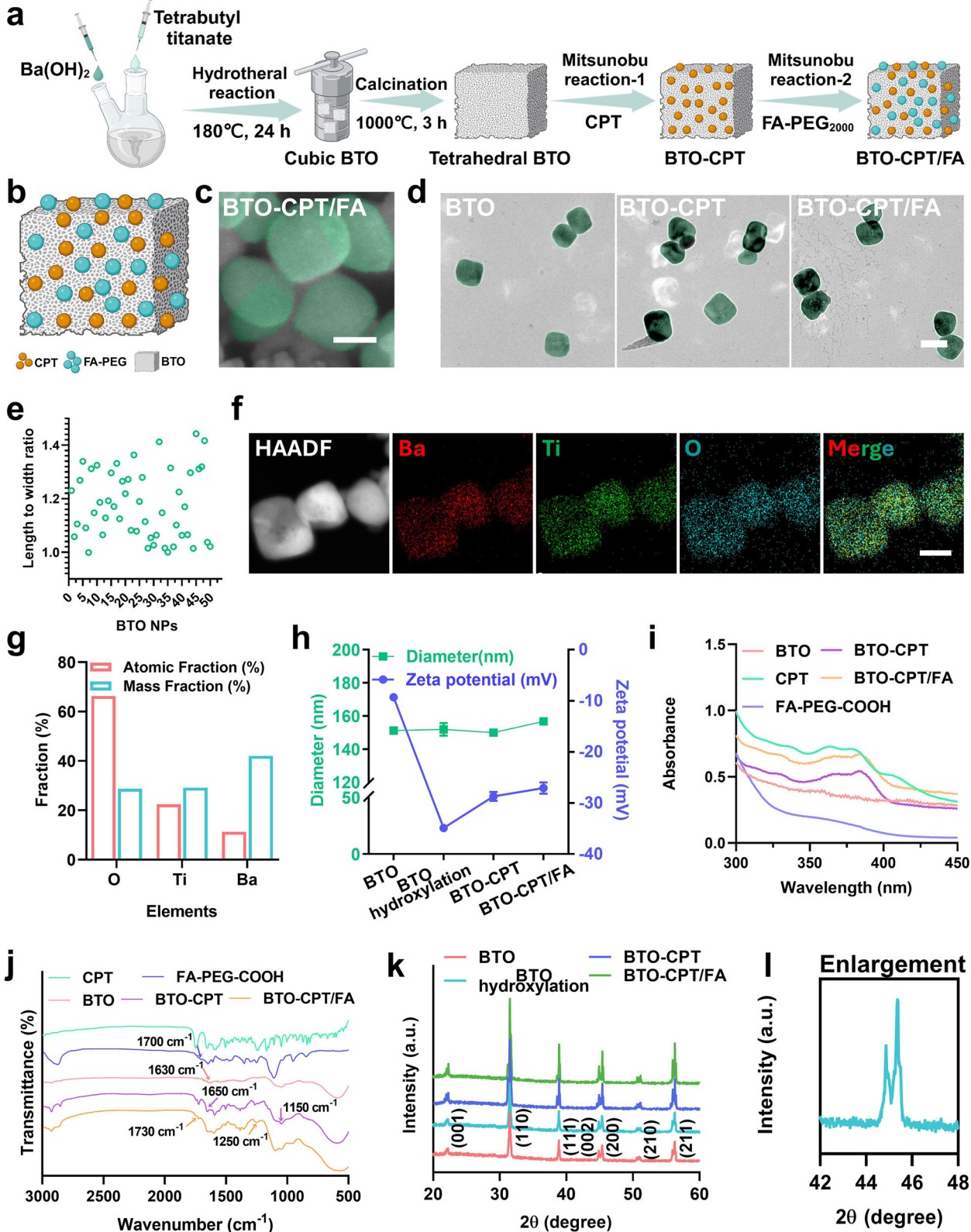

the piezoelectric module were continuously recorded over a 300-second period. The results demonstrated that under cyclic mechanical stimulation, the BTO-CPT/FA nanocomposites exhibited a robust ability to generate pulsed electrical signals, effectively simulating the dynamic intravesical pressure fluctuations that occur during bladder filling (Fig. 3b).

The output of potential generated by BTO-CPT/FA was found to be robust. Moreover, mechanical cycling tests demonstrated that BTO/CPT NPs and BTO-CPT/FA displayed piezoelectric output potentials comparable to those of BTO NPs, as evidenced by the pulsed electrical signals they produced (Fig. 3b). These analytical findings suggest that the incorporation of the CPT and the targeting agent folic

**Fig. 2 | Characterization of the self-driven electrical triggering system (BTO-CPT/FA).** **a** Schematic illustrating the synthesis of BTO-CPT/FA nanoplatform through hydrothermal reaction, calcination and the Mitsunobu reaction, respectively. **b** Schematic diagram of the structure of BTO-CPT/FA nanoplatform. **c** SEM image of BTO-CPT/FA nanoplatform, scale bar = 50 nm. **d** TEM images of BTO, BTO-CPT and BTO-CPT/FA, scale bar = 100 nm. **e** Analysis of length-to-width ratios for BTO-CPT/FA (*n* = 50). **f** High Angle Annular Dark Field (HAADF)-TEM image of BTO-CPT/FA and the corresponding elemental map, scale bar = 50 nm. **g** Mass percent and atomic percent of Ba, Ti and O in BTO-CPT/FA nanoplatform. **h** Hydrodynamic size and ξ-potential of different NPs (*n* = 3 technical samples examined over three independent experiments). **i** Ultraviolet-visible (UV-Vis) absorption spectra of BTO-CPT/FA nanoplatform. **j** Fourier Transform Infrared (FT–IR) spectra of BTO-CPT/FA nanoplatform. **k** X-ray diffraction (XRD) patterns of different NPs in BTO-CPT/FA nanoplatform. **l** Magnified XRD profiles in the range from 42° to 46°. The experiment was repeated three times, and the results were consistent (**c**, **d**, **f**). Source data are provided as a Source Data file. Figure 2a, b were created in BioRender. Xixi, L. (2025) https://BioRender.com/frctjq2, https://BioRender.com/3ee3gd8, respectively.

acid does not notably alter the crystal structure and piezoelectric response of BTO NPs.

To ensure precise verification, the piezoelectric behavior of the self-driven triggering system (BTO-CPT/FA) was examined at the nanoscale. A direct piezoelectric response force microscope was employed to characterize this behavior (Fig. 3c). A representative image provided a detailed contrast of the morphology, phase, and amplitude of the electricity-generating trigger system within a scanning area of 1 × 1 µm². The piezoelectric and ferroelectric characteristics were further confirmed by Fig. 3d, e which shows well-defined amplitude-voltage butterfly loops generated by the tip bias voltage (±8 V) and typical 180° phase change hysteresis loop. Figure 3f shows the corresponding atomic force microscope (AFM) image of BTO.

A notable contrast between the amplitude curve and phase distribution was evident, confirming that the self-driven electrical triggering system exhibited inherent piezoelectric properties. Additionally, in the absence of direct current, both BTO and BTO-CPT/FA generated distinct phase loops and piezo-amplitude butterfly loops, further demonstrating polarization reversal. The experimental data indicate that the self-driven electrical triggering system exhibits strong piezoelectricity and near-complete polarization characteristics at this scale. Furthermore, the modifications with FA and CPT did not impair the original piezoelectric properties of BTO (Fig. 3g–i).

To accurately evaluate the efficacy of the self-driven electrical triggering system in generating ROS in situ, in vitro experiments were conducted using ultrasound to simulate the intravesical pressure associated with bladder cancer. This method enabled the quantitative assessment of ROS production by the system. Malachite green was employed as a ROS indicator, as ROS can cleave its C = C and C = N bonds, leading to dye degradation (Fig. 3j). As shown in Fig. 3k, l, BTO-CPT/FA induced time-dependent degradation of malachite green under ultrasound stimulation, whereas no marked degradation was observed in the control group. Additionally, the decreased absorbance of malachite green at 618 nm further confirmed that BTO-CPT/FA effectively induced dye degradation upon ultrasound exposure—a widely accepted method for evaluating ROS generation (Fig. 3m).

To eliminate the potential influence of dissolved oxygen in deionized water on malachite green degradation, experiments were repeated using deoxygenated deionized water. As shown in Supplementary Fig. 6a, b, the control group showed negligible degradation, while the degradation rate in the BTO-CPT/FA group under ultrasound was notably reduced in deoxygenated conditions. Furthermore, the degradation efficiency of malachite green under normoxic and hypoxic conditions was compared between the control and BTO-CPT/FA groups (Supplementary Fig. 6c, d). In both groups, the degradation under hypoxia was slower than that under normoxia.

These findings suggest that dissolved oxygen plays a critical role in ultrasound-triggered malachite green degradation. The decreased degradation rate observed in deoxygenated conditions may be attributed to: (a) reduced potential for ultrasound-induced oxidation reactions in the absence of oxygen, and (b) the diminished capability of ultrasound to induce ROS generation—specifically, superoxide anion radicals—from BTO under low-oxygen conditions (as illustrated in Fig. 3j). To replicate the bladder environment, artificial urine was utilized. The results shown in Fig. 3n, o demonstrated that BTO-CPT/FA

appreciably degraded malachite green under ultrasound, whereas the control group exhibited no degradation.

To further investigate the role of ROS in this degradation, ROS scavengers including tri-n-butylamine (TBA), 1,4-benzoquinone (BQ), and ethylenediaminetetraacetic acid disodium salt (EDTA-Na) were introduced into the malachite green/BTO-CPT/FA reaction system. Following ultrasound exposure, malachite green degradation was markedly inhibited in the presence of these scavengers, indicating their effectiveness in suppressing ROS generation and confirming the critical role of ROS in the degradation process (Fig. 3p).

To further explore the mechanism by which electrical stimulation triggers CPT release from BTO-CPT/FA nanoplatforms, under the force of a linear reciprocating motion device to simulate bladder contraction and expansion, and the release of the drug was monitored (Supplementary Fig. 4b, Supplementary Movie. 1). As shown in Fig. 3q and Supplementary Fig. 4a, after seven days of continuous monitoring, there was no obvious CPT release was detected in the control group, while notably higher CPT release was monitored in the polarized BTO-CPT group, indicating that mechanical force induced the polarized BTO produced a piezoelectric effect during reciprocating motion and stimulating drug release. In summary, this experimental design eliminates other complex factors, such as the impact of ultrasound on the accuracy of the experimental results. Furthermore, X-ray diffraction (XRD) analysis confirmed the distinct crystal structures of polarized versus non-polarized BTO. Specifically, the Supplementary Fig. 4c reveals that the polarized BTO exhibited a characteristic split peak between 44° and 46°, indicative of successful polarization and the presence of a tetragonal phase, whereas this feature was absent in the non-polarized counterpart.

To ensure effective intravesical drug delivery, it is essential to evaluate the stability of BTO-CPT/FA in the bladder environment. Stability was assessed through in vitro experiments using simulated urine at pH 4.7, incubated at 37 °C for five days (Fig. 3r). Additionally, the ζ potential of BTO-CPT/FA was monitored throughout the five-day incubation (Supplementary Fig. 5). The results showed no obvious changes in ζ potential, indicating good colloidal stability under simulated physiological conditions. The above results indicated that throughout the incubation period, both the particle size and polymer dispersity index (PDI) remained stable. These findings confirm the robust stability of BTO-CPT/FA under acidic conditions, emphasizing the resilience of the self-driven electrical triggering system and its suitability for effective drug delivery in bladder conditions.

## Evaluation of the antitumor effects of self-driven electrical triggering system in vitro

Effective treatment of bladder cancer requires efficient uptake and intracellular transport of NPs by bladder cancer cells. In this study, mouse bladder cancer cells (MB49) were used to evaluate the cellular uptake of the prepared BTO-CPT/FA (Fig. 4a). To validate the therapeutic efficacy of the self-driven electrical triggering system, the impact of various concentrations of BTO-CPT/FA on MB49 cells was systematically assessed using an in vitro model (Fig. 4d). The primary goal was to investigate their antitumor effects in vitro.

Remarkably, within just 2 hours, MB49 tumor cells exhibited notable uptake of BTO-CPT/FA, primarily due to the high expression of

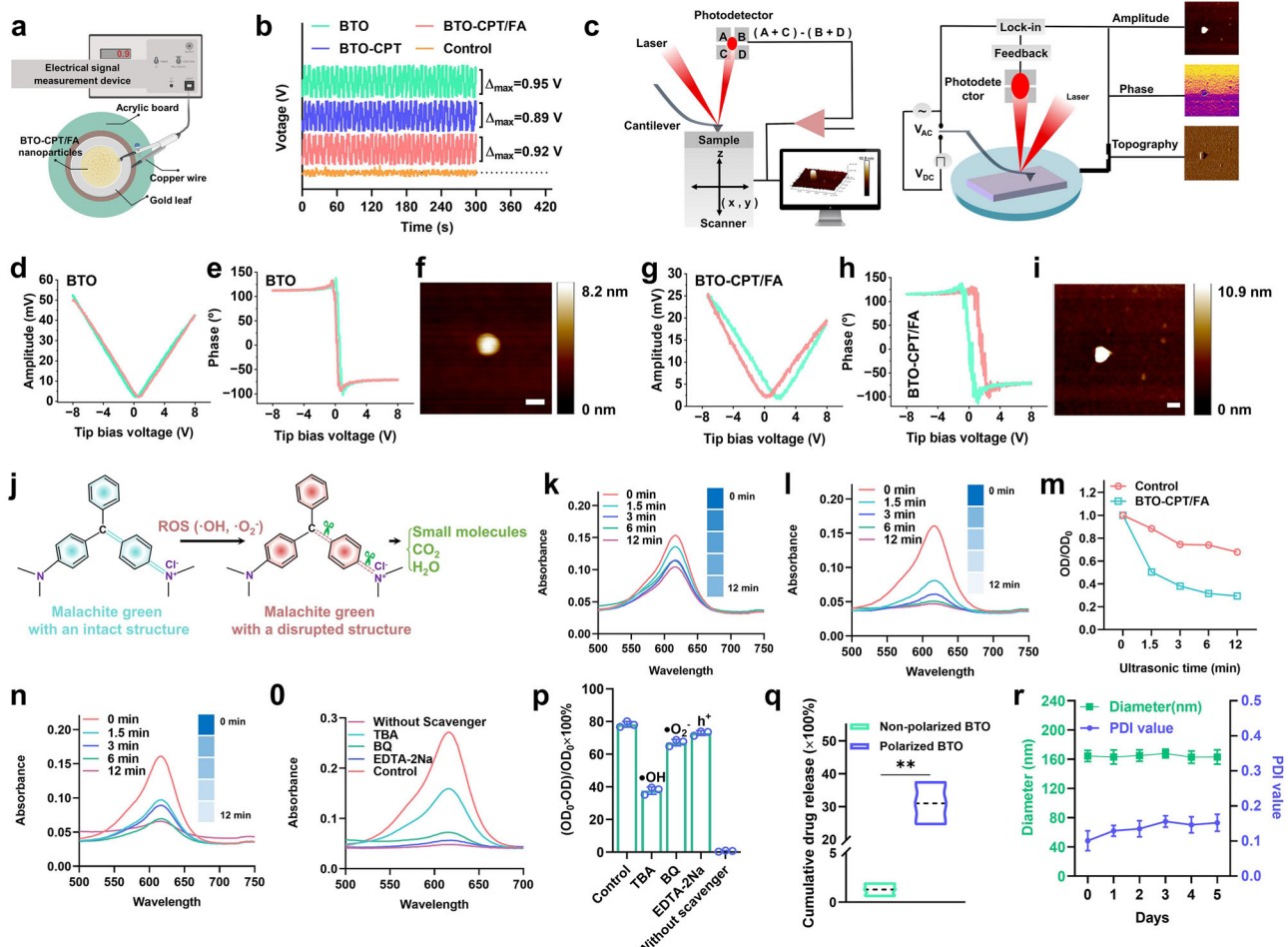

**Fig. 3 | Performance evaluation of self-driven electrical triggering system.**
**a** Schematic diagram of the device for testing the voltage generated by the self-driven electrical triggering system. **b** The curve of voltage generated by BTO-CPT/FA under linear reciprocation. **c** Schematic illustration of piezoelectric force microscopy (PFM). **d** Amplitude curve of BTOs. **e** Phase distribution of BTOs. **f** Surface topography of BTOs, scale bar = 100 nm. **g** Amplitude curve of BTO-CPT/FA. **h** Phase distribution of BTO-CPT/FA. **i** Surface topography of BTO-CPT/FA, scale bar = 100 nm. **j** Schematic diagram of the degradation of malachite green treatment with BTO-CPT/FA under ultrasound. **k** UV-vis absorption curve of malachite green after being treated by the control group under ultrasound in deionized water. **l** UV-vis absorption curve of malachite green after being treated with BTO-CPT/FA under ultrasound in deionized water. **m** UV-vis absorption curve of malachite green at 618 nm after treatment by control or BTO-CPT/FA. **n** UV-vis absorption curve of malachite green after being treated with BTO-CPT/FA group under ultrasound in artificial urine. **o** UV-vis absorption curve of malachite green after being treated with BTO-CPT/FA in the presence of inhibitors under ultrasound in artificial urine. **p** The effect of inhibitors on the degradation of malachite green after treatment with BTO-CPT/FA under ultrasound in deionized water ($n = 3$ independent experiments). **q** The cumulative release amount of CPT from BTO-CPT/FA after one week of monitoring ($n = 3$ independent experiments). The bottom, middle, and top of each box plot represented the 25th, 50th, and 75th percentiles. **$p$-value = 0.0013. **r** Stability testing of BTO-CPT/FA in artificial urine ($n = 3$ independent experiments). The experiment was repeated three times, and the results were consistent (**f**, **i**). Data are presented as mean values ± SD. p values are calculated using t-tests without adjustments (**q**). Source data are provided as a Source Data file. Figure 3a was created in BioRender. Xixi, L. (2025) https://BioRender.com/3yf5705.

folate receptor alpha on the MB49 tumor cell surface, which facilitates the effective uptake of folate-conjugated NPs. In contrast, BTO NPs without folic acid exhibited notably lower uptake (Fig. 4b). Additionally, compared to BTO-Cy5.5 without folic acid coating, BTO-Cy5.5/FA nanoparticles showed an evident improvement in uptake. Specifically, the distribution of BTO-CPT/FA in MB49 cells was 2.5 times higher than that of BTO-Cy5.5 after 2 hours (Fig. 4c).

Furthermore, using ultrasound to simulate intravesical pressure and promote drug release considerably enhanced the therapeutic efficacy against bladder cancer. Confocal laser scanning microscopy (CLSM) was used to visualize live and dead MB49 cells after treatment with different groups. The fluorescence intensity of Calcein-AM and PI in each group, as shown in Fig. 4e, reflects cell viability. Notably, the PI fluorescence intensity was substantially higher in the BTO-CPT/FA + US group compared to the BTO + US and BTO-CPT + US groups, indicating a higher level of cell death (Fig. 4f). This suggests a marked enhancement in therapeutic effect achieved through folic acid targeting.

Specifically, BTO-CPT/FA exposed to ultrasound-simulated intravesical pressure exhibited a marked increase in ROS levels, demonstrating highly efficient ROS generation under these conditions. To further assess the capacity of bladder cancer cells to generate sufficient ROS under simulated intravesical pressure for tumor destruction, we used 2,7-Dichlorodihydrofluorescein diacetate (DCFH-DA), a fluorescent dye that reflects cellular redox processes, to measure ROS levels. As shown in Fig. 4g, h, BTO-CPT/FA generated a robust ROS response under intravesical pressure, promoting the destruction of simulated bladder cancer cells. In contrast, minimal ROS signals were detected in the CPT and BTO-CPT/FA groups without ultrasound, confirming that ultrasound activation is essential for triggering the piezoelectric catalytic effect. This underscores the potential of combining ultrasound-triggered drug release with folic acid targeting to enhance the efficacy of bladder cancer treatment. Importantly, the cytotoxicity of BTO-CPT/FA under simulated intravesical pressure was greater than that of BTO and BTO-CPT,

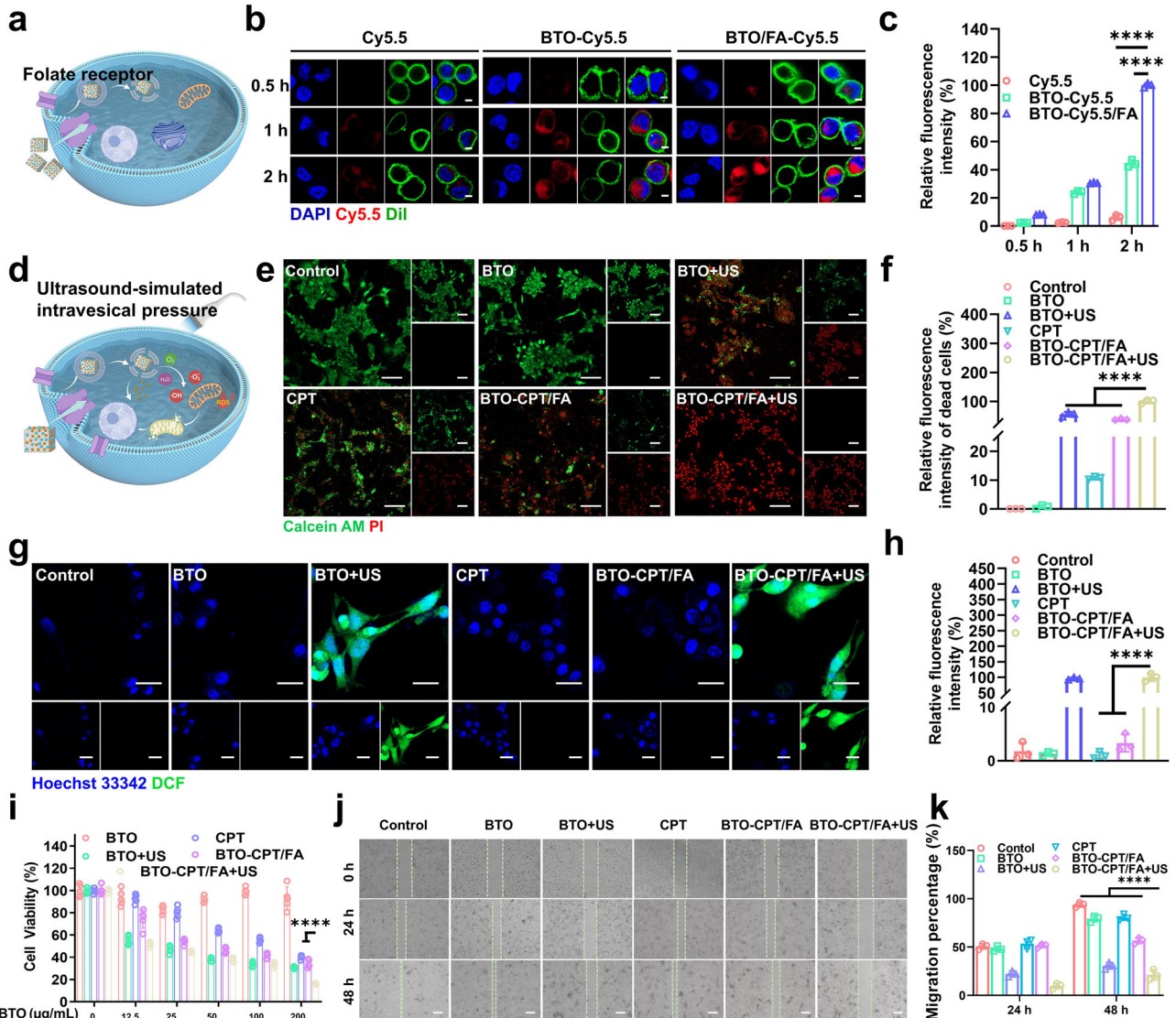

**Fig. 4 | Evaluation of the antitumor effects of the BTO-CPT/FA in vitro.**
**a** Schematic representation of specific uptake mediated by folate receptor.
**b** Confocal images of MB49 cells treated with different groups, scale bar = 5 μm.
**c** Quantitative analysis of Cy5.5 fluorescence intensity ($n = 3$ independent biological experiments). ****$p < 0.0001$. **d** Schematic diagram of the generation of ROS and the release of CPT from BTO-CPT/FA nanoplatform under ultrasound. **e** Confocal images of live-dead cell staining for MB49 cells after treatment with different groups, scale bar = 100 μm. **f** Quantitative analysis of PI fluorescence intensity ($n = 3$ biologically independent experiments). ****$p < 0.0001$. **g** The DCF images of MB49 cells after being treated with different groups, scale bar = 25 μm. **h** Quantitative analysis of the fluorescence intensity of DCF ($n = 3$ biologically independent experiments). ****$p < 0.0001$. **i** Cell viability of MB49 cells after treated with different groups, assessed using an MTT assay ($n = 5$ biologically independent experiments). ****$p < 0.0001$. **j** The confocal images of the Migration of MB49 cells after treated with different groups, scale bar = 100 μm. **k** Calculation of migration rates for MB49 cells ($n = 3$ biologically independent experiments). ****$p < 0.0001$. Data are presented as mean values ± SD. $p$ values are calculated via one-way ANOVA with Tukey's multiple comparisons test without adjustments (**f**, **h**). $p$-values are calculated via two-way ANOVA with Tukey's multiple comparisons test without adjustments (**c**, **i**, **k**). Source data are provided as a Source Data file. Figure 4a, d were created in BioRender. Xixi, L. (2025) https://BioRender.com/qgs5unc.

suggesting that the nanoparticles' ability to self-generate electricity and their targeting function play crucial roles in tumor eradication (Fig. 4i). In addition, the IC50 value of BTO-CPT/FA is 50.51 μg/mL under ultrasound, and the related IC50 curve was uploaded to revised manuscript (Supplementary Fig. 9). To further validate the efficacy of BTO-CPT/FA in inhibiting the invasion and migration of bladder cancer cells under simulated intravesical pressure, we conducted scratch assays. As shown in Fig. 4j, k, after 48 hours, the mobility of cells treated with BTO-CPT/FA + US was reduced by 80% compared to the positive control. These results indicate that BTO-CPT/FA effectively hinders the invasion and migration of cancer cells under conditions mimicking intravesical pressure. This effect is likely due to the nanoparticles' intrinsic electrical activity and their ability

to generate ROS, which contributes to the elimination of bladder cancer cells.

## Biosafety and biological applicability evaluation of BTO-CPT/FA nano-system
A high biosafety and biological applicability are essential prerequisites for the application of BTO-CPT/FA nano-system in tumor treatment effectively. In order to evaluate comprehensively biosafety and biological applicability of BTO-CPT/FA nano-system, the following experiments were designed.

Initially, the sensitivity of normal bladder epithelial cells and bladder cancer cells to ROS is one aspect. As shown in Supplementary Fig. 10a, through live/dead cell staining experiments, it was found that

BTO-CPT/FA has weak toxicity to MB49 cells without ultrasound, but notably induced cell death under ultrasound. This process induced a marked piezoelectric effect under ultrasound, producing abundant ROS, disrupting the redox balance of cancer cells, and accelerating tumor cell apoptosis. We selected bladder epithelial cells SV-HUC-1 (a type of bladder epithelial cell) as normal cells to investigate the effects of ROS on it under the same conditions. As shown in Supplementary Fig. 10b, BTO-CPT/FA had almost no cytotoxicity for SV-HUC-1 cells without ultrasound application. However, under ultrasound, the apoptosis induced by BTO-CPT/FA was discernibly lower than that in MB49 cells. Therefore, it is evident from the above results that MB49 cells are more sensitive to ROS induced by the piezoelectric effect at the same concentration.

Subsequently, the mitochondrial membrane potential and calcium ion channels of bladder epithelial cells and bladder cancer cells were evaluated after treatment under different conditions. As shown in the Supplementary Fig. 11, for tumor cells MB49, the mitochondrial membrane potential remains complete in the absence of ultrasound, indicating that BTO-CPT/FA does not cause evident changes in membrane potential. However, under ultrasound, the cells exhibit considerable apoptosis, and the membrane potential shows marked changes compared to the non-ultrasound, suggesting that ROS generated by the piezoelectric effect induced by ultrasound severely disrupts membrane integrity. On the contrary, for bladder epithelial cells SV-HUC-1, there were no obvious apoptosis phenomena regardless of ultrasound application, and the cell membranes remained relatively integral. Additionally, to validate the differential activation of calcium ion channels by the BTO-CPT/FA piezoelectric nano-system in tumor cells and normal epithelial cells, cells were stained by calcium ion probe Fluo-4 AM under different conditions. As shown in the Supplementary Fig. 12, for MB49 cells, there was no obvious green fluorescence in the absence of ultrasound, indicating that the calcium ion channel is not activated; after ultrasound application, clear green fluorescence is visible within the cells, indicating that calcium ion channels are sizably activated. However, for bladder epithelial cells SV-HUC-1, regardless of whether ultrasound treatment is administered, no obvious green fluorescence is visible within the cells, indicating that calcium ion channels are not activated under these conditions. In summary, cancer cells respond more sensitively to ROS generated by the BTO-CPT/FA piezoelectric nano-system, disrupting membrane potential and activating calcium ion channels, and specifically reducing the toxic side effects on normal bladder epithelial cells, and also demonstrate that the piezoelectric BTO-CPT/FA nano-system is feasible for biological applicability.

In addition, histological and biochemical assays were important consideration that should not be disregarded. As shown in Supplementary Fig. 25a, through hematoxylin-eosin (HE) staining experiments, in healthy mouse model, intravesical infusion of the BTO-CPT/FA piezoelectric nano-system into mouse bladders via the urethra did not cause any toxic side effects on bladder tissue. In contrast, when the same amount of BTO-CPT/FA piezoelectric nano-system was perfused into mice with in orthotopic bladder cancer mice, it induced noteworthy apoptosis, producing cellular debris (green dashed circles), and infiltration of neutrophils (indigo arrows) and lymphocytes (yellow arrows) was observed (Supplementary Fig. 25b). This demonstrates that normal bladder tissue exhibits higher tolerance to the same dose of BTO-CPT/FA, aligning with our experimental safety design principles. In addition, as shown in Supplementary Fig. 26, through the analysis of blood parameters to demonstrate the hematological safety of BTO-CPT/FA nano-system. There were no indicative changes in blood-related parameters, whether in healthy mice or in orthotopic bladder cancer mice, clearly demonstrating the excellent hematological safety of BTO-CPT/FA. Furthermore, this result is similar to the results of the biocompatibility evaluation of the heart, liver, spleen, lungs, and kidneys of mice and rabbits before and after treatment with the BTO-CPT/FA piezoelectric nanomaterial system described in the manuscript.

## Investigation of the endocytosis mechanism of the BTO-FA nano-system

We systematically investigated the endocytosis mechanism of our delivery system and classical delivery pathways, including passive diffusion, clathrin-mediated endocytosis, and caveolin-dependent endocytosis.

Initially, the mesoporous silica (MS NPs) and gold nanoparticles (Au NPs) were synthesized, which are two types of nanosized carriers that rely on clathrin-mediated endocytosis and caveolin-mediated endocytosis, respectively. As shown in the Supplementary Fig. 13a, b, MS NPs with a particle size of approximately 40 nm was prepared via the gel-sol method, and Au NPs with a particle size of approximately 33 nm were prepared via the self-assembly method. Both nanocarriers exhibit excellent monodispersity and uniform particle size, providing a solid structural foundation for subsequent endocytosis experiment.

Subsequently, chlorpromazine (CPZ) and methyl-β-cyclodextrin (MCD) were selected as inhibitors of endocytosis mediated by clathrin and caveolin, respectively, to investigate their influence on the endocytosis efficiency of different nano-systems. As shown in Supplementary Fig. 13b, f, comparison with the control group revealed that endocytosis of MS NPs was detectably inhibited after CPZ pretreatment, with a reduction in endocytosis efficiency of 24.4%. Similarly, the endocytosis efficiency of Au NPs decreased by 12.6 % after MCD pretreatment compared to the control group (Supplementary Fig. 13c, f). Meanwhile, we investigated the feasibility of these two classical internalization pathways in the BTO-CPT/FA nano-system. The experimental results indicated that after pretreatment with CPZ and MCD, respectively, there was no discernible decrease in the endocytosis efficiency of BTO-FA NPs (Supplementary Fig. 13e, f), suggested that this nano-system relied on the specific binding of folate and folate receptors to achieve effective endocytosis. Additionally, we investigated the influence of temperature on the endocytosis of BTO-FA NPs, as shown in Supplementary Fig. 13d, f, when the temperature was lowered to 4 °C, the fluorescence intensity of BTO-FA NPs decreased by 45.7%, indicating that the endocytosis in this system is energy-dependent.

In conclusion, we systematically investigated the differences among the nano-delivery system designed in this manuscript and classical delivery pathways, we can conclude that the endocytosis of BTO-FA NPs is an energy-dependent process mediated by folate receptors, and that TNTs play a key role in the transport of nanodrugs among cells.

## The mechanism of the system for high-speed drug delivery through the tunneling nanotube

Intercellular connections play a crucial role in facilitating material transport and signal exchange between cells. The effective and intelligent utilization of these connections holds noteworthy promise for advancing nanomedicine-based tumor therapies. Tunneling nanotubes (TNTs) are one such connection that can be leveraged for this purpose. Given the efficacy of BTO-CPT/FA in treating bladder cancer cells, we further explored whether the self-generating electrical triggering system could activate the tunneling nanotube network under simulated intravesical pressure, thereby enhancing the targeted delivery of drugs. To investigate this, we examined whether BTO-CPT/FA-mediated therapy could improve drug uptake by tumor cells. CLSM images (Fig. 5a) revealed the presence of tunneling nanotubes between MB49 cells.

To further evaluate the enhanced penetration of BTO-CPT/FA within the tumor microenvironment, tunneling nanotubes were visualized using SEM (Fig. 5b, c). The results clearly demonstrate that, under ultrasound, BTO-CPT/FA successfully navigate through these

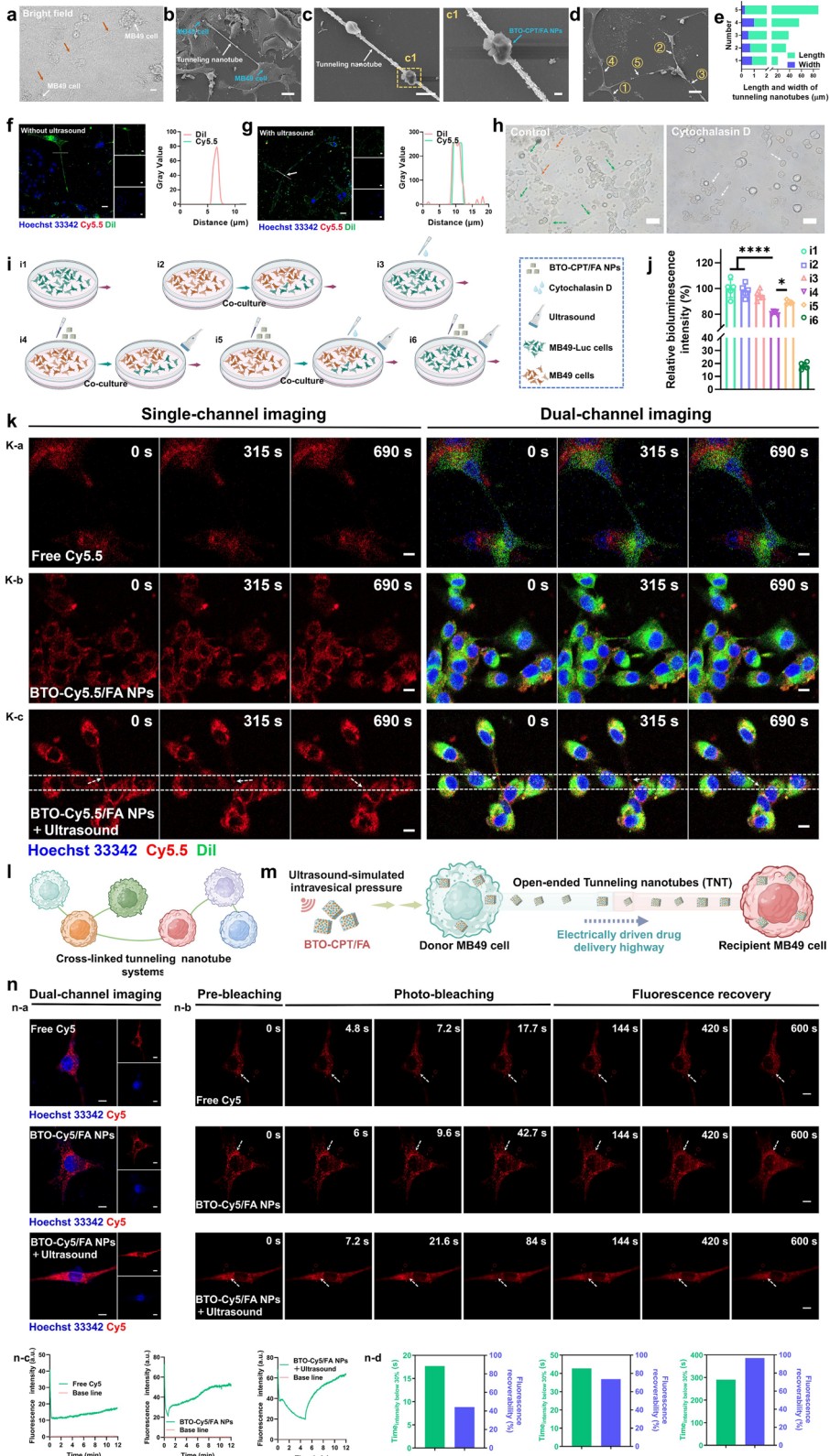

nanotubes, facilitating their entry into tumor cells. This highlights the crucial role of intravesical pressure in activating the tunneling nanotube network and promoting the thorough infiltration of tumors by BTO-CPT/FA, emphasizing their potential for more effective therapeutic delivery.

Additionally, the cross-linked tunneling nanotube systems among MB49 cells were observed through SEM (Fig. 5d, i). As shown in Fig. 5e,

the length and width were measured by Image J, and the length of the tunneling nanotubes was measured randomly, from 20 μm to 80 μm, while the width was frequently less than 1 μm. Moreover, bladder cancer cells can exchange materials through TNTs, and in the presence of self-driven electrical stimulation, these structures enable the controlled delivery of BTO-CPT/FA. A suggestive observation emerged when comparing BTO-CPT/FA treatment alone with BTO-CPT/FA exposed to

**Fig. 5 | High-speed delivery of self-driven electrical triggering system through tunneling nanotubes (TNTs). a** CLSM image showing TNTs between MB49 cells, scale bar = 20 μm. **b** SEM image showing TNTs between MB49 cells, scale bar = 20 μm. **c** Enlarged SEM image of a TNT, scale bar = 5 μm, which the BTO-CPT/FA was indicated in period c1, scale bar = 1 μm. **d** SEM image of cross-linked TNTs systems, scale bar = 20 μm. **e** Quantitative analysis of the length and width of TNTs. **f** (left) CLSM image illustrating BTO-CPT/FA that has not yet entered the TNT, scale bar = 20 μm. (right) Fluorescence intensity analysis along the line scan from Fig. 5f. **g** (left) CLSM image showing BTO-CPT/FA entering the TNT, scale bar = 20 μm. (right) Fluorescence intensity analysis along the line scan from Fig. 5g. **h** CLSM images of MB49 cells after being treated with control or Cytochalasin D, scale bar = 100 μm. **i, j** Effect of Cytochalasin D on the transport of BTO-CPT/FA through TNTs. (left) Schematic representation of the effect of cytochalasin D on the transport of BTO-CPT/FA. (right) Relative bioluminescence intensity of MB49-Luc cells in different groups ($n$ = 5 independent biological experiments). *$p$-value = 0.0174, ****$p$ < 0.0001. **k** Single particle tracking (SPT) images of BTO-Cy5.5/FA nanoparticles transported among MB49 cells through TNTs. a) Free Cy5.5, b) BTO-Cy5.5/FA NPs and c) BTO-Cy5.5/FA NPs + ultrasound, scale bar = 10 μm. **l** Schematic diagram of the cross-linked TNTs systems. **m** Schematic illustration of the formation of TNTs driven by BTO-CPT/FA under ultrasound. **n** Fluorescence recovery after photobleaching (FRAP) images of BTO-Cy5/FA NPs in MB49 cells. a) Free Cy5, b) BTO-Cy5/FA NPs and c) BTO-Cy5/FA NPs + ultrasound, scale bar = 10 μm. The experiment was repeated three times, and the results were consistent (**a–c, f–h, k, n**). Data are presented as mean values ± SD. $p$ values are calculated via one-way ANOVA with Tukey's multiple comparisons test without adjustments (**j**). Source data are provided as a Source Data file. Figure 5i, 5l, 5m were created in BioRender. Xixi, L. (2025) https://BioRender.com/ub0o582, https://BioRender.com/rey5mxt, respectively.

ultrasound-simulated intravesical pressure, showing a marked increase in nanoparticle accumulation within the tunneling nanotubes under the latter condition, additionally, fluorescence intensity data from line scans of the tunneling nanotubes revealed notable enrichment of Cy5.5 and DiI fluorescence in the BTO-CPT/FA group under ultrasound simulation of intravesical pressure (Fig. 5f, g). This highlights the capacity of self-generating electrical triggering systems to enhance drug penetration across physiological barriers, facilitated by rapid transport through the tunneling nanotube pathways.

Tunneling nanotubes play a crucial role in the high-speed transport of BTO-CPT/FA between MB49 cells. To gain a deeper understanding of the transport mechanism, in order to have a more comprehensive insight into the transport of BTO-Cy5.5/FA NPs among MB49 cells through TNTs, some advanced experimental technologies, such as the single particle tracking (SPT) and the fluorescence recovery after photobleaching (FRAP) based on real-time imaging of live cells, were conducted. In the SPT experiment, the free Cy5.5 and BTO-Cy5.5/FA without ultrasound were selected as control groups; there was an inconspicuous movement of Cy5.5 or BTO-Cy5.5/FA NPs (Fig. 5k-a, b, Supplementary Movie. 2, 3). In contrast, for the ultrasound group, there is an obvious phenomenon that BTO-Cy5/FA nanoparticles removal through TNTs (Fig. 5k-c, Supplementary Movie. 4), meanwhile, the transport rate reached 0.0151 μm/s, which is comparable to the transport of viral particles in fibroblasts or epithelial cells through tunneling nanotubes and similar to the rate of macrophage-driven nanomedicine transport to tumor cells[43–47]. In conclusion, these results suggest that TNTs are important bridges for the high-speed transport of BTO-Cy5/FA nanoparticles among MB49 cells. Ultrasound could induce a piezoelectric effect and enhance the BTO-Cy5/FA nanoparticles transport in (between) cells, which could improve the therapeutic outcomes in tumor treatment.

In addition, in FRAP experiment, free Cy5 and BTO-Cy5/FA without ultrasound were selected as control groups to stimulate passive diffusion, while the BTO-Cy5/FA with ultrasound treatment as experimental group (Fig. 5n-a), specifically, after photo-bleaching, a faster fluorescence decay rate of Cy5 can be observed in the control groups, the time taken for the fluorescence intensity to decay to 30% of its initial value was 12.7 s and 42.7 s in free Cy5 and BTO-Cy5/FA with ultrasound treatment groups (Fig. 5n-b, c, Supplementary Movies 5, 6, 7), respectively, furthermore, after photo-bleaching, fluorescence is highly unlikely to recover. It is pleasing to note that the BTO-Cy5/FA with ultrasound group showed excellent resistance against photo-bleaching and fluorescence recovery performance, the time taken for the fluorescence intensity to decay to 30% of its initial value was 290.6 s (Fig. 5n-d), after photo-bleaching, the fluorescence recovery rate reached 96.55% within 12 min. Based on the above, combined with the experimental results of SPT, ultrasound can induce a piezoelectric effect and non-negligibly improve the transmission of the piezoelectric nanostructure BTO-CPT/FA through the TNTs, increase the communication frequency of nanodrugs in cells

(between cells), prolong the photobleaching time, and increase the fluorescence recovery rate.

To further investigate the role of tunneling nanotubes in drug delivery, we compared conditions that inhibit their formation. Cytochalasin D, a known inhibitor of tunneling nanotube formation, was used to assess its impact. As shown in Fig. 5h, the control group displayed abundant tunneling nanotubes, some connecting cells (green arrows), while others were extending outward (orange arrows), with intact cell morphology. In contrast, Cytochalasin D treatment discernibly reduced the number of tunneling nanotubes, with altered cell morphology (white arrows). This result indicates that Cytochalasin D effectively inhibits tunneling nanotube formation[48].

The ability of BTO-CPT/FA to be delivered via tunneling nanotubes and induce cell death is crucial for enhancing tumor treatment efficacy while maintaining biosafety. To investigate whether BTO-CPT/FA transported through tunneling nanotubes could induce cell death in surrounding cells, MB49 cells treated with BTO-CPT/FA were co-incubated with untreated Luc-MB49 cells (labeled with luciferase), and bioluminescence intensity was measured.

The experimental groups included: i1) MB49-Luc cells, i2) MB49 cells + MB49-Luc cells, i3) Cytochalasin D + MB49-Luc cells, i4) MB49 cells (BTO-CPT/FA NPs) + MB49-Luc cells + ultrasound, i5) MB49 cells (BTO-CPT/FA NPs) + MB49-Luc cells + Cytochalasin D + ultrasound, and i6) MB49-Luc cells (BTO-CPT/FA NPs) + ultrasound (Fig. 5i). As shown in Fig. 5j, the bioluminescence intensity in the i2 group remained unchanged, indicating that the addition of MB49 cells did not affect the bioluminescence of MB49-Luc cells. However, a slight reduction in cell activity was observed after co-incubation with Cytochalasin D, likely due to its inhibitory effect on cell proliferation[49]. More notably, an appreciable decrease in bioluminescence intensity was observed in Luc-MB49 cells co-incubated with BTO-CPT/FA-treated MB49 cells. When Cytochalasin D was added, the death rate of MB49-Luc cells decreased slightly, suggesting that Cytochalasin D inhibited the delivery of BTO-CPT/FA by impairing tunneling nanotube formation.

These results indicate that: (1) BTO-CPT/FA can pass through tunneling nanotubes to surrounding cells, inducing cell death; (2) Cytochalasin D impairs the delivery efficiency of nanomedicine by reducing the formation of intercellular tunneling nanotube networks; and (3) Cytochalasin D may also reduce the cytotoxic effects of nanomedicines by inhibiting cellular uptake of the BTO-CPT/FA. Furthermore, detectable cytotoxicity was observed in MB49-Luc cells co-incubated with BTO-CPT/FA[50,51], consistent with the results from Fig. 4i. In conclusion, these findings suggest that BTO-CPT/FA delivered through tunneling nanotubes can effectively induce cell death, demonstrating their anti-tumor potential.

## Biodistribution of the self-driven electrical triggering system

To assess the precise targeting capability of the self-driven electrical triggering system on the tumor, we replaced the drug with Cy5.5 and tracked its biological distribution following bladder perfusion using an

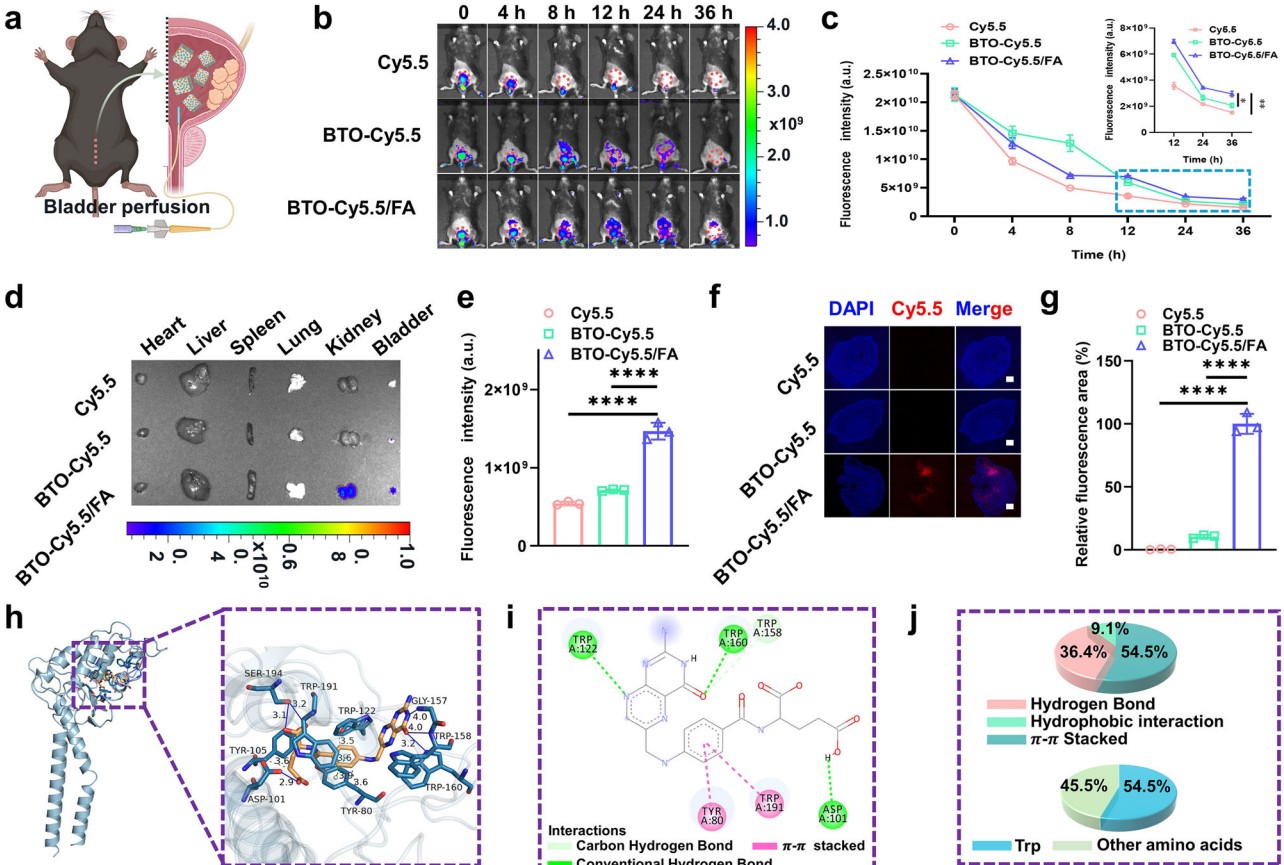

**Fig. 6 | Folate receptor-mediated specific enrichment of self-driven electrical triggering system in orthotopic bladder cancer mice. a** Schematic representation of bladder perfusion of BTO-CPT/FA. **b** In vivo live imaging at different time points post-bladder perfusion across various groups. **c** Quantitative analysis of Cy5.5 fluorescence intensity in bladder regions at different time points (*n* = 3 mice for each group). *p*-value = 0.0144, **p*-value = 0.0064. **d** Ex vivo live imaging of major tissues. **e** Quantitative analysis of Cy5.5 fluorescence intensity in isolated bladder tissue (n = 3 in each group). ****p* < 0.0001. **f** CSLM images of the bladder section, scale bar = 500 μm. **g** Quantitative analysis of Cy5.5 fluorescence intensity

(*n* = 3 in each group). ****p* < 0.0001. **h** Three-dimensional representation of the interaction between folic acid and the folate receptor post-docking. **i** Two-dimensional representation of the interaction between folic acid and the folate receptor post-docking. **j** Analysis of the interaction forces between folic acid and the folate receptor, including the proportion of major amino acid residues. Data are presented as mean values ± SD. *p* values are calculated via one-way ANOVA with Tukey's multiple comparisons test without adjustments (**c, e, g**). Source data are provided as a Source Data file. Figure 6a was created in BioRender. Xixi, L. (2025) https://BioRender.com/w65mvky.

in vivo imaging system (IVIS) (Fig. 6a). Notably, tumor uptake of BTO-Cy5.5 without targeting was decreased over time, while the decline in uptake of BTO-Cy5.5/FA with targeting was markedly slower, furthermore, there was a substantial difference in Cy5.5 fluorescence intensity among the BTO-CPT/FA group and the other two groups in time range of 8 h ~ 36 h. (Fig. 6b, c, and the insert picture of Fig. 6c). The relative fluorescence intensity within specific regions of interest was quantified using an animal in vivo imaging system (IVIS) Lumina II. Further quantitative analysis of tumor accumulation was conducted through ex vivo fluorescent imaging of isolated tumors 36 hours post-bladder perfusion, revealing prolonged tumor retention of BTO-Cy5.5/FA compared to BTO-Cy5.5 and free Cy5.5 groups (Fig. 6d, e).

Additionally, cryosections of tumor tissues confirmed the difference in tumor accumulation between BTO-Cy5.5 and BTO-Cy5.5/FA groups. Compared to BTO-Cy5.5 without FA modification, BTO-Cy5.5/FA exhibited a notable improvement in accumulation, with a five-fold higher particle distribution in tumor tissue (Fig. 6f, g). These findings demonstrate that folate receptor-mediated more accumulation of BTO-CPT/FA in bladder cancer, and has the potential to enhance its antitumor efficacy and support reduced perfusion doses.

To establish the relevance of molecular docking simulations to the experimental system and confirm the ability of FA to bind folate receptors in tumor tissues, we conducted molecular docking studies

(Fig. 6h–j). Both three-dimensional and two-dimensional docking results revealed that hydrogen bonding, hydrophobic interactions, and π-π stacking interactions are the primary forces driving FA-receptor binding, with tryptophan playing a key role in mediating these interactions.

In line with these computational findings, the structural design of the BTO-CPT/FA nanoplatform ensures that FA remains readily accessible for receptor binding. During synthesis, CPT is first chemically coupled to BTO, followed by FA-PEG2000-COOH modification. This design ensures that the FA-PEG2000-COOH group is oriented outward, facilitating preferential binding to folate receptors on tumor cell membranes. Additionally, as FA is conjugated to the terminal end of PEG2000, the flexible linker allows it to extend into the surrounding medium, maintaining an optimal conformation for receptor recognition in both in vitro and in vivo environments.

This structural arrangement ensures that when the nanoplatform encounters tumor cells, FA binds to folate receptors, promoting the accumulation of the nanodrug at the tumor site. Following receptor-mediated endocytosis, CPT is then released, inducing DNA damage. Moreover, the molecular docking simulations confirm that the fundamental molecular recognition process between FA and folate receptors remains intact, even when FA is incorporated into the BTO-CPT/FA nanoplatform. This reinforces the applicability of the computational results to the experimental system.

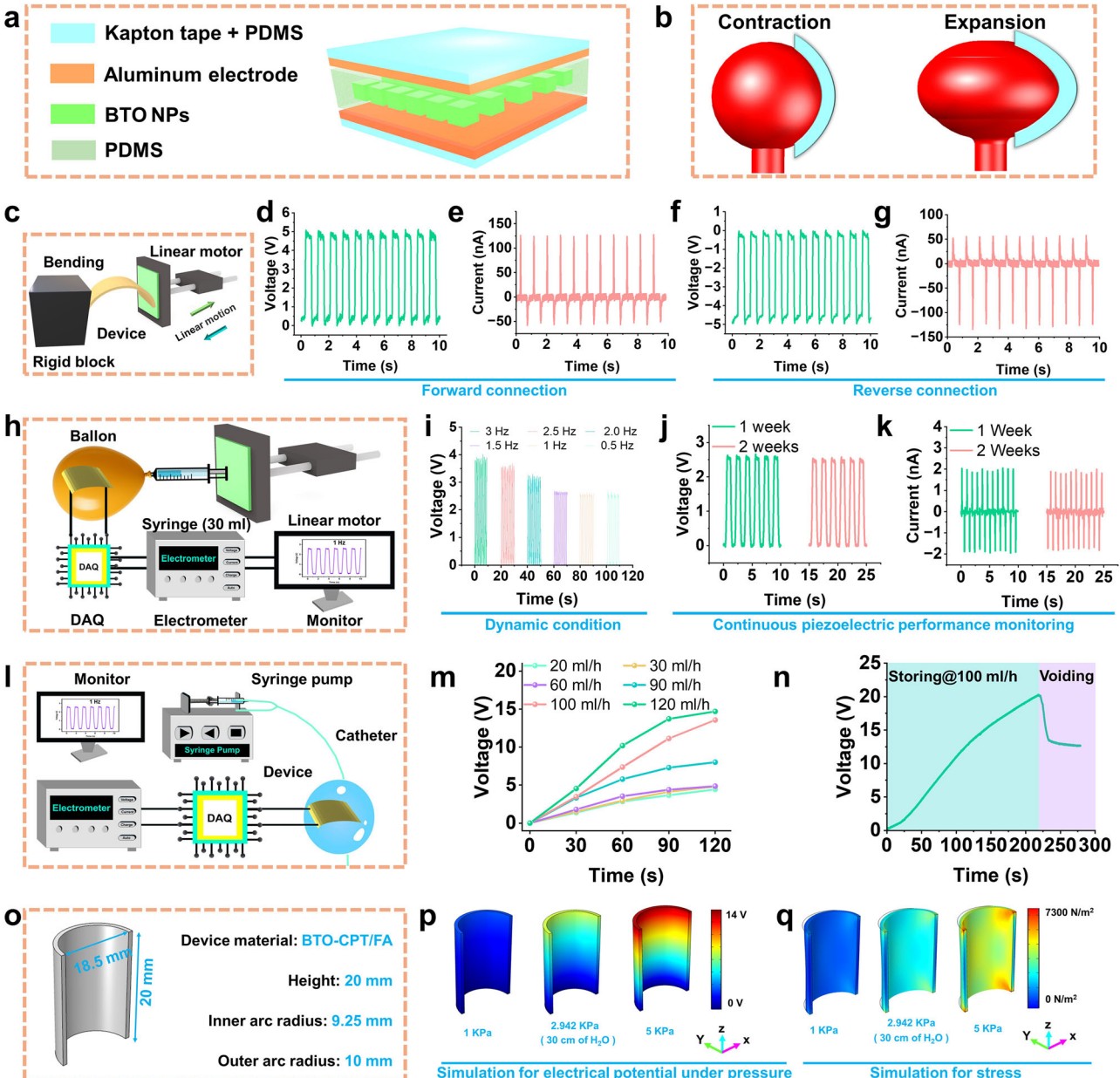

**Fig. 7 | Electromechanical performance of the PDMS + BTO-CPT/FA device for urinary bladder simulation. a** Schematic of the device with a composite PDMS piezoelectric layer integrated with BTO-CPT/FA. **b** Illustration of device bending during biomechanical contraction and expansion of the urinary bladder. **c** Experimental setup for mechanical to electrical energy conversion under bending of BTO based device. **d** and **e** Output voltage and current in forward connection. **f** and **g** Output voltage and current in reverse connection. **h** Setup for simulating bladder biomechanics using a piezoelectric device on a balloon controlled by syringe airflow and a linear motor. **i** Voltage output from the piezoelectric device during simulated bladder motion at different frequencies. **j** and **k** Long-term durability data showing voltage and current outputs of the device. **l** Experimental setup for urinary bladder response simulation using a catheter system. **m** Electrical output at different water flow rates in the catheter. **n** Bladder monitoring using the BTO-CPT/FA device on the catheter balloon during water storage (100 ml/h) and voiding state. **o** Geometrical parameters of the piezoelectric device for finite element analysis. **p** and **q** Simulated piezoelectric potential and stress distribution under pressures ranging from 1 Kpa to 5 Kpa, respectively. Source data are provided as a Source Data file. Figures are created in Blender 4.2. (https://www.blender.org/) and AutoCAD 2024. (https://www.autodesk.com/in/products/autocad/overview).

## Electrical performance evaluation of a simulated self-driven electrical triggering system for bladder cancer treatment

A flexible piezoelectric device responsive to both mechanical load and stimuli is proposed as a promising approach for biomechanical bladder monitoring. The schematic illustrates the layer-by-layer sandwich design employed to fabricate the flexible piezoelectric device (Fig. 7a). The device consists of a piezoelectric layer composed of BTO NPs integrated with polydimethylsiloxane (PDMS). The piezoelectric layer is interfaced with PET/Al electrodes with an Al electrode. The preparation process of the piezoelectric layer containing PDMS and BTO

NPs as shown in Supplementary Fig. 14. Additionally, the fabricated piezoelectric device exhibits high flexibility (Supplementary Fig. 15), ensuring excellent conformal contact with the balloon and catheter used for simulating the urinary bladder. During bladder expansion and contraction,n corresponding to the storage and voiding phases the piezoelectric device undergoes bending deformation as depicted in Fig. 7b and Supplementary Fig. 16.

The electromechanical performance of the device was further analyzed using a device with dimensions (20 × 15 × 0.7 mm³) at a driving frequency of 1 Hz[52]. The weight percentages (wt%) of BTO in

PDMS + BTO composite were optimized at 10 %, 20 %, 30 % and 40 %. As shown in Supplementary Fig. 17 **a-left**, the open-circuit voltage increased with BTO content increasing at 30 wt%, while decreasing at 40 wt%. This decline at higher BTO content can be attributed to the interplay between dielectric constant, piezoelectric coefficient and Young's modulus of PDMS + BTO composite. A similar trend was observed in PDMS + BTO-CPT/FA based device (Supplementary Fig. 17 **a, b-left**).

Bending tests using a linear motor further evaluated the electromechanical response of device (Fig. 7c). In Supplementary Movie 8, the device of PDMS + BTO-CPT/FA piezoelectric layer under reciprocating rectilinear movement force to simulate bladder contraction- expansion. Comparisons of the piezoelectric voltage responses were conducted for devices with 30 wt% BTO in both PDMS + BTO and PDMS + BTO-CPT/FA composites. The peak-to-peak open-circuit voltages were measured at 5 V and 3.99 V, respectively (Supplementary Fig. 17 **a, b-right**).

Additionally, PFM analysis revealed a stronger piezoelectric amplitude in the PDMS + BTO composite compared to the PDMS + BTO-CPT/FA composite (Fig. 3d, g). Furthermore, the output voltage and current of the nanogenerator with various load resistances are shown in Supplementary Fig. 18. The measured output voltage and current at a specific load resistance are multiplied to determine the device's effective output power (P) (Supplementary Fig. 19). Up to a maximum value, output power increases with increasing load resistance before decreasing. At 20 MΩ, the maximum output power is 56 nW.

Switching polarity measurements were performed to confirm that the observed output signals originated from the piezoelectric response. As shown in Fig. 7d–g, the peak-to-peak voltage and current outputs in forward and reverse polarity connections were ±5 V and ±180 nA, respectively. The nearly identical output magnitude under reversed polarity with opposite signal polarity verified that the electrical output was generated by the piezoelectric effect. This reversible switching of signals demonstrated the inherent piezoelectric nature of the device.

The urinary bladder functions as a dynamic reservoir undergoing cyclic filling and voiding, facilitated by complex biomechanical motion with air, along with urine, causing internal pressure variations and structural deformation during storage (expansion) and voiding (contraction). To replicate this behavior, the effects of air and water on bladder biomotion were independently studied using an experimental setup consisting of a rubber balloon attached to a 30 mL syringe (Fig. 7h). The piezoelectric response of the device was evaluated at varying frequencies (0.5–3 Hz). Forward motion of the linear motor pushed air into the balloon, causing expansion, while reverse motion induced contraction. The reverse motion was restricted to maintain the balloon's dome shape.

As shown in Fig. 7i, the peak-to-peak voltage magnitude decreased with declining frequencies and stabilizing at 1 Hz, indicating a frequency-dependent response. Durability testing over one and two weeks revealed stable voltage (2.5 V) and current (4 nA), demonstrating the device's robust long-term performance under cyclic loading, which is critical for bioelectronic applications as illustrated in Fig. 7j, k. The real-time images of PDMS + BTO-CPT/FA piezoelectric device under cyclic loading were shown by the Supplementary Movie 9, Supplementary Fig. 20.

To investigate the piezoelectric response under varying fluid flow rates causing bladder deformation, a customized catheter model was developed as shown in Fig. 7l. The setup included a 10 mL water-filled syringe connected to a catheter (16 Fr/Ch, 5–15 mL/cc). Initially, 5 mL of water was added to the catheter via its non-returning valve to maintain a spherical shape, enabling attachment of the BTO-CPT/FA piezoelectric device using double-sided tape on its top surface. The water flow rate was controlled between 20 ml/h and 120 ml/h using an electromechanical syringe pump generating pressurized fluid momentum that caused balloon expansion. This expansion induced bending in the piezoelectric device, producing an electrical output that increased with higher flow rates, as shown in Fig. 7m.

To mimic bladder storage and voiding functions, the water flow rate was set to 100 ml/h (Fig. 7n, Supplementary Movie 10). During the storage phase, the voltage output increased to 20 V with a non-linear voltage-time relationship observed. For the voiding phase, the syringe pump was manually detached, creating higher intravesical pressure within the catheter compared to the syringe end. This pressure difference caused reverse water flow, pushing the syringe backward and leading to a sudden voltage drop.

Finite element method (FEM) simulations using COMSOL Multiphysics were conducted to analyze the piezoelectric potential distribution under compressive pressures ranging from 1 kPa to 5 kPa, supporting the pressure-voltage relationship observed in the catheter model. For the simulation setup, bottom and top cross-sections of the device were defined as ground and floating potential, respectively, while the vertical edges were fixed (Fig. 7o). Compressive pressure loads were applied to the inner curved surface to replicate bladder deformation. A physics-controlled fine mesh was employed for simulations to ensure high accuracy.

The simulations revealed a progressive increase in piezoelectric potential from the bottom to the top of the device, consistent with the applied compressive pressure. A comparison of piezoelectric responses under varying pressures (Supplementary Fig. 21-a) showed a linear increase in electrical potential with the highest output (-13.6 V) recorded at 5 kPa and the lowest (-2.71 V) at 1 kPa (Fig. 7p). Additionally, the stress distribution under the same pressure conditions demonstrated uniform stress across the device surface, ranging from 0 N/m$^2$ to a maximum of 7300 N/m$^2$ (Fig. 7q, Supplementary Fig. 21b). The simulation results closely aligned with experimental data demonstrating similar voltage generation patterns and confirming the model's validity in capturing the voltage behavior induced by intravesical pressure.

To investigate the piezoelectric performance and durability of BTO-CPT/FA-based piezoelectric nanogenerator device in a urinary bladder for dynamics monitoring, an ex vivo experiment was conducted using a porcine bladder model weighing 54.5 g[53–57]. As illustrated in Supplementary Fig. 22-a, b, 60 mL of deionized (DI) water was injected into the bladder via an 18 FR/CH (5–10 cc/mL) catheter and a 70 mL capacity syringe. During the forward stroke of the syringe, bladder expansion due to increased internal pressure resulted in a corresponding increase in piezoelectric voltage output (-7 V). Conversely, during the backward stroke, bladder contraction led to a decrease in voltage output (Supplementary Fig. 22c).

Additionally, as depicted in Supplementary Fig. 22d, e, DI water was introduced into the bladder at a constant flow rate of 100 mL/h using a syringe pump. During the storage phase, the voltage output increased from 0 V to a peak of 2.86 V, before declining to 1.28 V during the manually induced voiding phase (Supplementary Fig. 22f). These findings validate the sensor's robustness and effectiveness in real-time bladder monitoring, demonstrating its potential for clinical translation in urological healthcare applications.

## Evaluation of the anti-tumor effects in vivo of a self-driven electrical triggering system

To further confirm that the electrical stimulation generated by intravesical pressure can promote ROS production, intravesical pressure of control and orthotopic bladder cancer mice in vivo or in vitro was accurately quantified using a tonometer (Fig. 8a, b). As illustrated in Fig. 8c, the intravesical pressure in the tumor with orthotopic bladder cancer mice was measured at 60 cm H$_2$O, while the pressure in a normal bladder was only 30 cm H$_2$O. Notably, the results obtained from the isolated bladder were consistent with those observed in vivo,

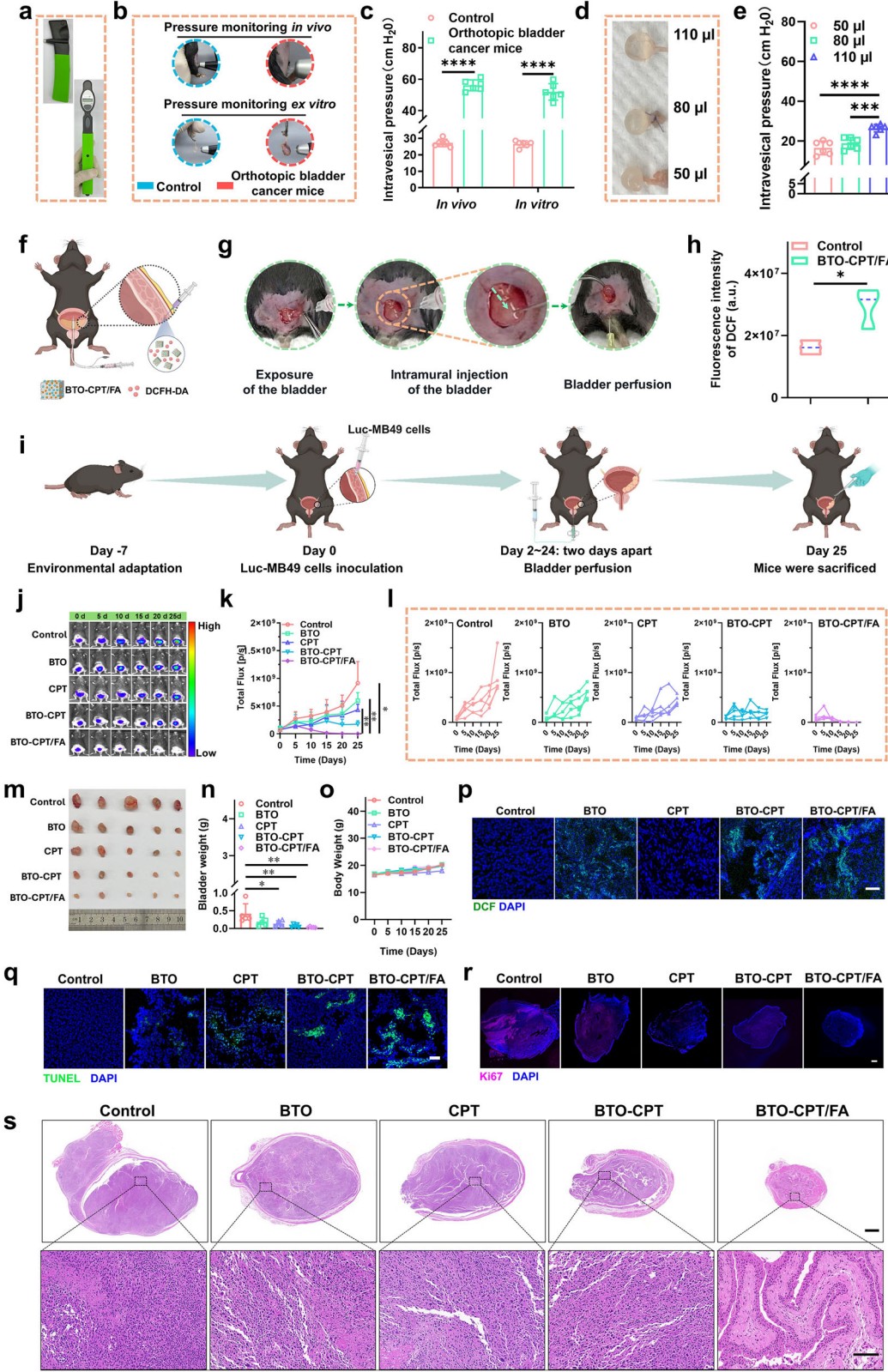

validating the effectiveness of the tonometer in accurately quantifying intravesical pressure both in vivo and ex vitro. As shown in Fig. 8e, it's surprised to observe that the pressure of the ex vivo bladder after perfusion with different volumes of artificial urine showed a positive correlation with the volume of perfusion.

Upon injecting the self-driven triggering system (BTO-CPT/FA) into the bladder wall, we simulated its interaction with the bladder environment, where the intravesical pressure triggered the generation of ROS (Fig. 8f, g). As shown in Fig. 8h, BTO-CPT/FA induced a substantial amount of ROS at the tumor site in response to intravesical pressure, effectively facilitating the destruction of bladder cancer cells. This demonstrates the system's potential for targeted therapeutic efficacy through ROS production triggered by physiological conditions.

**Fig. 8 | Evaluation of antitumor effect of BTO-CPT/FA in vivo. a** Images of the pressure tonometer. **b** Images of pressure testing via pressure tonometer in vivo and ex vivo. **c** Quantitative analysis of intravesical pressure for Fig. 8b ($n = 6$ mice for each group). ****$p < 0.0001$. **d** Images of isolated bladders perfused with artificial urine. **e** Quantitative analysis of intravesical pressure for Fig. 8 d ($n = 6$ mice for each group). ***$p$-value = 0.0007, ****$p < 0.0001$. **f** Schematic illustration of ROS detection in mouse bladder wall. **g** Images shown injecting different groups into the bladder wall. **h** Quantitative analysis of DCF fluorescence intensity after mouse bladder wall incubated with different groups ($n = 3$ in each group). *$p$-value = 0.0285. **i** Therapeutic schedule in MB49 orthotopic bladder cancer mice model. **j** Bioluminescence images of the orthotopic bladder cancer mice in different groups. **k** Bioluminescence curves of orthotopic bladder cancer mice ($n = 5$ mice for each group). *$p$-value = 0.0277 (Control $vs$. BTO-CPT/FA), **$p$-value = 0.0037 (BTO $vs$. BTO-CPT/FA), **$p$-value = 0.0029 (CPT $vs$. BTO-CPT/FA).

**l** Bioluminescence curves of each mouse ($n = 5$ mice for each group). **m** Photographs of the isolated bladders ($n = 5$ in each group). **n** Weights of isolated bladders ($n = 5$ in each group). *$p$-value = 0.0390 (Control $vs$. CPT), **$p$-value = 0.0098 (Control $vs$. BTO-CPT), **$p$-value = 0.0039 (Control $vs$. BTO-CPT/FA). **o** Body weight curves of orthotopic bladder cancer mice (n = 5 mice for each group). **p** CLSM images of ROS detection, scale bar = 50 μm. **q** CLSM images of TUNEL staining, scale bar = 50 μm. **r** CLSM images of Ki-67 staining, scale bar = 500 μm. **s** Panoramic H&E images, scale bar = 100 μm (top), 10 μm (bottom). The experiment was repeated three times, and the results were consistent (**p–s**). Data are presented as mean values ± SD. $p$-values are calculated via two-way ANOVA (**c, k**), t-test (**h**) and one-way ANOVA (**e, n**), respectively. Source data are provided as a Source Data file. Figure 8f, i created in BioRender. Xixi, L. (2025) https://BioRender.com/yc9j5l1, https://BioRender.com/20cg3uq, respectively.

Inspired by the performance of the self-driven electrical trigger system, we further assessed its anticancer efficacy in vivo. To establish an orthotopic bladder cancer model, MB49 bladder cancer cells modified with the luciferase gene were inoculated into the bladder wall of C57BL/6 mice (Fig. 8i), while the surgical procedure for inoculation of MB49 cells within the bladder wall of mice is shown in Supplementary Fig. 23a. This model allowed real-time monitoring of tumor growth using the IVIS Lumina II. The mice were randomly divided into five groups: an untreated control group receiving PBS, and four treatment groups receiving either free CPT, BTO NPs, BTO-CPT NPs, and BTO-CPT/FA at equivalent concentrations (Fig. 8j). To determine the optimal perfusion volume, various volumes of deionized water were perfused into the bladders of mice. An overflow phenomenon was observed when the perfused volume exceeded 50 μL. Therefore, the perfusion volume was set at 50 μL (Supplementary Fig. 23b). Throughout the treatment period, changes in bioluminescence signals, body weight, and tumor size were closely monitored, providing comprehensive insights into the therapeutic effects.

Post-treatment, the bioluminescence curve in the bladder regions of the BTO-CPT/FA group showed a meaningfully slower rate of tumor progression compared to the other groups (Fig. 8k, l). Following treatment, tumors were excised, photographed, and their volumes and weights were measured. While treatment with free CPT, free BTO NPs, and BTO-CPT NPs led to some degree of tumor growth inhibition, evidence of continued tumor growth was observed. In contrast, treatment with BTO-CPT/FA resulted in an evidently enhanced antitumor effect, as demonstrated by substantial reductions in both tumor volume and weight (Fig. 8m, n). Furthermore, BTO-CPT/FA treatment did not lead to detectable weight loss during the bladder therapy period (Fig. 8o), confirming the treatment's safety and therapeutic potential.

Additionally, cryosection analysis of tumor tissues confirmed that BTO-CPT/FA generated a considerably higher amount of ROS within the tumors compared to other groups (Fig. 8p). TUNEL and Ki67 immunofluorescence staining further indicated a marked improvement in tumor suppression in the BTO-CPT/FA group compared to the others (Fig. 8q, r). Finally, H&E staining of isolated bladder tissues, observed via Case Viewer panoramic scanning technology, confirmed that BTO-CPT/FA exhibited the most effective therapeutic impact on tumors (Fig. 8s). As shown in Supplementary Fig. 24, H&E staining of major organs in mice treated with different groups revealed no discernible pathological damage, further emphasizing the minimal toxicity of BTO-CPT/FA. In conclusion, intravesical perfusion of BTO-CPT/FA demonstrates exceptional antitumor efficacy with minimal harm to normal tissues.

## Mechanisms unveiling the anti-tumor potential of self-driven electrical triggering system

To investigate the mechanism by which the self-driven electrical triggering system (BTO-CPT/FA) inhibits bladder cancer progression,

bladder tumor tissues were collected after treatment with BTO-CPT/FA and from untreated controls. RNA was extracted from these tissues for subsequent analysis. The RNA transcriptomic analysis revealed 958 differentially expressed genes in the BTO-CPT/FA-treated group compared to the control group. Of these, 624 genes were evidently downregulated, while 958 genes were upregulated (Fig. 9a).

Furthermore, moreover, the heatmap results in Fig. 9b indicated that tunneling nanotube formation and the enhanced delivery capability between MB49 cells were facilitated by the upregulation of genes such as Mmp2, Rasd2, Wasf3, Prkcb, Prkcq, and Prkcz. Additionally, the downregulation of genes like Ptk2 and Baiap2, which inhibit tunneling nanotube formation, further contributed to this process. These genes are linked to the remodeling of the extracellular matrix (ECM) and the reorganization of the cytoskeleton after treatment with BTO-CPT/FA. Gene Set Enrichment Analysis (GSEA) further revealed that the observed changes in gene expression were positively correlated with ECM glycoproteins and the core matrisome, which play key roles in the formation of new extracellular structures and signaling pathways. This, in turn, promoted the formation of tunneling nanotubes (Figs. 9c, d) following the treatment of orthotopic bladder cancer mice with BTO-CPT/FA.

The Gene Ontology (GO) enrichment analysis revealed that the differentially expressed genes (DEGs) were primarily localized in tumor vasculature, ECM, and cell membrane structures (Fig. 9e). Additionally, GO analysis indicated that BTO-CPT/FA treatment influenced cell growth and several key cellular functions, including protein serine-threonine-tyrosine kinase activity, protein tyrosine kinase activity, and transmembrane receptor protein kinase activity.

Kyoto encyclopedia of genes and genomes (KEGG) pathway analysis highlighted detectable enrichment in several critical signaling pathways, including the PI3K-Akt signaling pathway, GMP-PKG signaling pathway, proteoglycans in cancer, cytokine-cytokine receptor interaction, inflammatory mediator regulation of TRP channels, axon guidance, and gap junctions (Fig. 9f–h). Interestingly, many of these pathways are directly or indirectly involved in tunneling nanotube formation. Following BTO-CPT/FA treatment, the rapid delivery of nanomedicines via cross-linked tunneling nanotubes substantially enhanced the therapeutic efficacy against tumors[58–61].

GO enrichment analysis of differentially expressed genes also revealed considerable enrichment in microtubule-related pathways, such as regulation of spindle microtubule attachment to the kinetochore, tubulin binding, microtubule binding, regulation of microtubule-based processes, and microtubule plus-end binding, following BTO-CPT/FA treatment. Importantly, microtubules are directly involved in the formation and maintenance of tunneling nanotubes. For example, γ-tubulin and microtubule-associated proteins (MAPs) are key components of microtubules that provide biomechanical support for tunneling nanotubes, thereby facilitating their extension. As such, these microtubule-related pathways positively influence tunneling nanotube formation[62–66] (Fig. 9i).

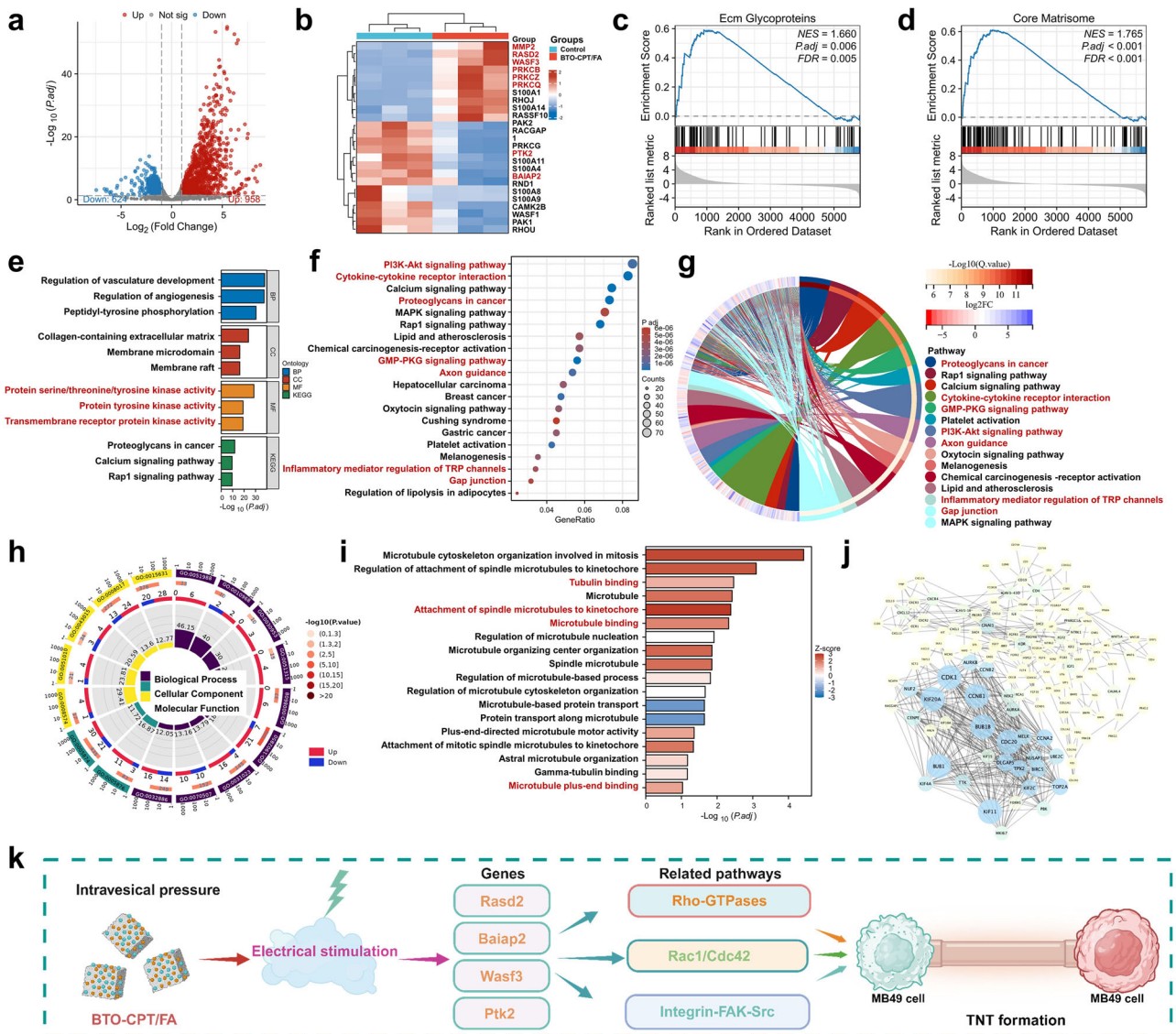

**Fig. 9 | Transcriptomic analysis of tumor tissues treated with self-driven electrical triggering system (BTO-CPT/FA).** **a** A volcano plot illustrating the upregulated and downregulated genes among the differentially expressed genes in the control and BTO-CPT/FA groups. **b** A heatmap of the differentially expressed proteins. **c** and **d** GSEA demonstrating positive enrichment of differentially expressed genes (DEGs) in cellular processes. **e** Gene Ontology (GO) enrichment analysis of DEGs between the control and BTO-CPT/FA groups. **f** Enrichment analysis for DEGs triggered by BTO-CPT/FA treatment. **g** Chordal plots of kyoto encyclopedia of genes and genomes (KEGG) pathway analysis after treatment with BTO-CPT/FA. **h** GO enrichment analysis of the DEGs after treatment with control or BTO-CPT/FA groups. **i** GO enrichment of the DEGs related to microtubules after treatment with BTO-CPT/FA. **j** Functional association networks of variably expressed microtubule-associated proteins. **k** Mechanism of tunneling nanotubes formation among MB49 cells after treated by BTO-CPT/FA. Source data are provided as a Source Data file. Figure 9k was created in BioRender. Xixi, L. (2025) https://BioRender.com/vpcd32z.

Additionally, protein–protein interaction network analysis revealed the expression of key genes associated with tunneling nanotube (TNT) formation (Fig. 9j). Notably, the expression of *CD79A/CD79B* and *CDC20* highlighted a marked phenomenon in TNT formation following treatment with BTO-CPT/FA NPs, particularly aiding in the extension of the cell membrane to form the tubular structures required for TNTs. Furthermore, the increased expression of CCL19/CCR1 and CCL19/CCR6 in the BTO-CPT/FA group may facilitate the establishment of the TNT structure by promoting membrane protrusion and cellular pseudopod formation. These results indicate that key genes and protein expressions were appreciably altered in response to electrical stimulation triggered by intravesical pressure following BTO-CPT/FA treatment. Moreover, signaling pathways closely linked to TNT formation were activated, providing mechanistic support for the high-speed transport of BTO-CPT/FA NPs through

TNTs among tumor cells, and laying the groundwork for improving tumor therapy outcomes (Fig. 9k).

## Evaluation of anti-tumor effects of self-driven electrical triggering system in larger animals

To evaluate the effectiveness of the proposed self-driven electrical triggering system (BTO-CPT/FA) in larger animals, we tested its application in rabbits with orthotopic bladder cancer. The model was established by inoculating Vx2 tumor cells into the rabbit bladder (Fig. 10a). As shown in Fig. 10b, Vx2 tumor cells were successfully inoculated into the rabbit bladder. Following treatment with BTO-CPT/FA, the size of the bladder cancer tissues of rabbits was sizably reduced, as depicted in Fig. 10c, d. H&E staining results (Fig. 10e, f) further confirmed this reduction. These findings demonstrate that BTO-CPT/FA generate electricity under

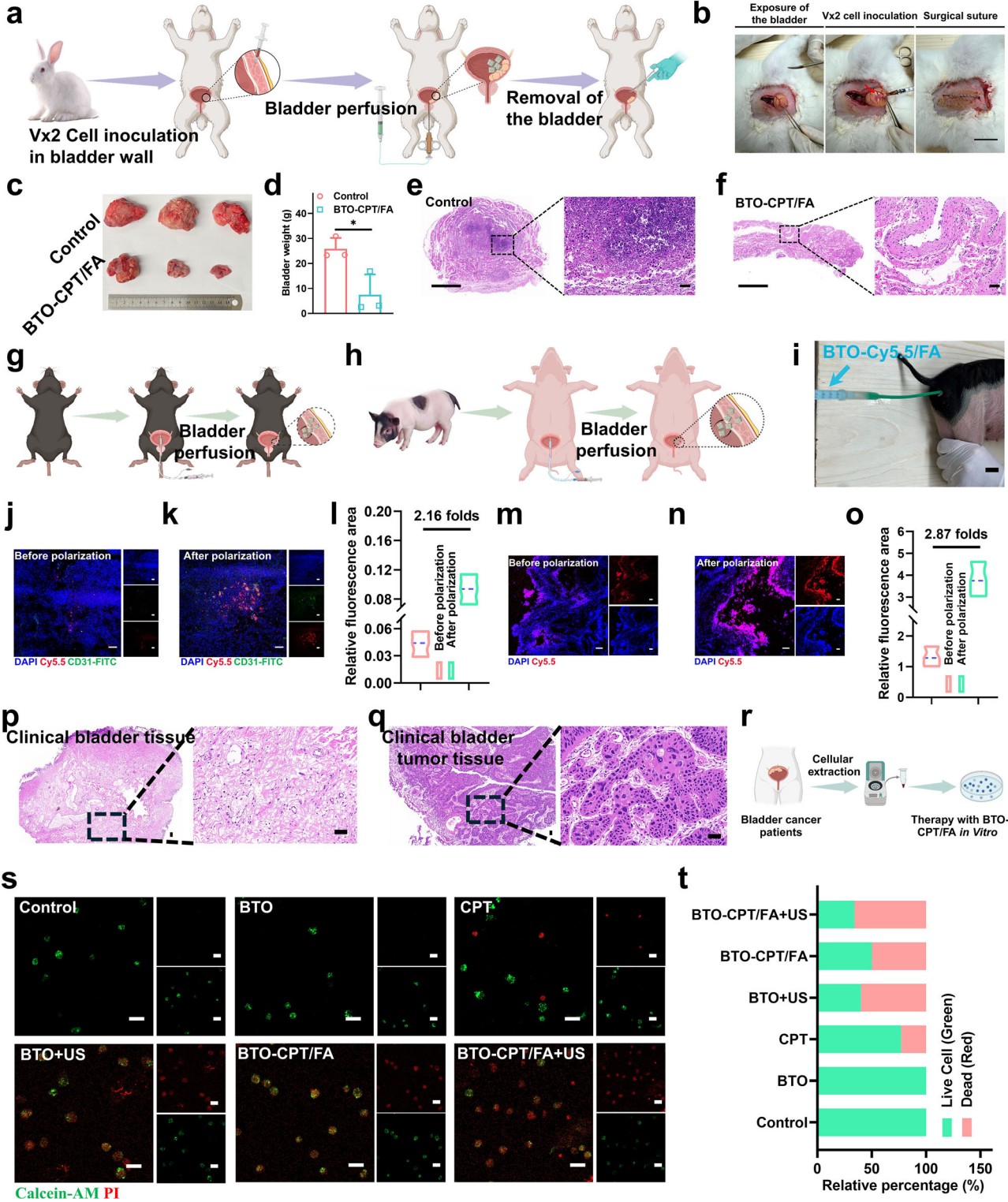

intravesical pressure, which not only facilitates rapid drug transport through tunneling nanotubes but also produces large amounts of ROS, effectively targeting and killing bladder cancer cells.

To evaluate the potential toxicity of BTO-CPT/FA, we conducted a series of blood chemical and blood-related parameter tests (Supplementary Figs. 27, 28). These results showed no discernible deviations from those of healthy rabbits, confirming that BTO-CPT/FA with a safe biocompatibility. Additionally, H&E staining results further supported these findings (Supplementary Fig. 29).

To verify that the self-driven electricity triggering system can effectively overcome physiological barriers, we selected mice and min pigs for validation in vivo. We demonstrated the perfusion process of the self-driven electrical triggering system in the bladders of mice and mini pigs (Fig. 10g–i). Following the bladder permeability experiment, mini pigs were euthanized via intracardiac injection of KCl solution, and the fine structure of their urinary system was analyzed (Supplementary Figs. 30, 31). As shown in Fig. 10j, k, the polarized spontaneous electrical triggering system, exhibiting strong self-driven electrical performance, resulted in non-negligibly greater permeability in the

**Fig. 10 | Evaluation of the therapeutic efficacy of BTO-CPT/FA on orthotopic bladder cancer rabbits and tumor cells from clinical bladder cancer patients, respectively, and assessment of bladder permeability in mice and minipigs.** **a** Therapeutic plan for orthotopic bladder cancer rabbits. **b** Images of rabbit bladder wall inoculated with Vx2 tumor cells, scale bar = 3 cm. **c** Photograph of the isolated bladders after treatment with different groups. **d** Analysis of bladder weight (n = 3 rabbits for each group). *p-value = 0.0260. **e, f** H&E images of rabbits post-treatment with different groups, scale bar =1 mm (left), 50 μm (right). **g, h** Schematic diagram of the permeability evaluation in the bladder of mice and minipigs, respectively. **i** Image showing bladder perfusion for minipigs, scale bar = 2 cm. **j, k** CLSM images of frozen sections of the mouse bladder treated with unpolarized BTO NPs and polarized BTO NPs, respectively, scale bar = 50 μm. **l** Quantitative analysis of the Cy5.5 fluorescence area for Fig. 10j, k (n = 3 in each group). **m, n** CLSM images of frozen sections of the minipig bladder treated with unpolarized BTO NPs and polarized BTO NPs, respectively, scale bar = 50 μm. **o** Quantitative analysis of the Cy5.5 fluorescence area for Fig. 10m, n (n = 3 in each group). **p, q** H&E images of bladder tissue and clinical bladder cancer tissue from clinical bladder cancer patient samples, respectively, scale bar = 50 μm. **r** Flowchart detailing the extraction of tumor cells derived from clinical bladder cancer samples. **s** CLSM images of live-dead cell staining of tumor cells derived from clinical bladder cancer samples after treatment with different groups, scale bar = 20 μm. **t** Quantitative analysis of the proportion of live cells and dead cells after different treatments. The experiment was repeated three times, and the results were consistent (e, f, p, q, s). Data are presented as mean values ± SD. p values are calculated via t-test without adjustments. Source data are provided as a Source Data file. Figure 10a, g, h, r created in BioRender. Xixi, L. (2025) https://BioRender.com/9kog1cx, https://BioRender.com/w9kvl55, https://BioRender.com/91qmy52, respectively.

mouse bladder. The fluorescence intensity of Cy5.5 in the bladder of mice after treatment with polarized BTO-Cy5.5/FA NPs was approximately 2.16-fold higher than in the non-polarized NPs group (with weak self-driven electrical performance) (Fig. 10l). The polarization effect of BTO NPs mediated the bladder permeability in minipigs in consistent with mice was observed, with even better result presentation, which may be attributed to the greater intravesical pressure of the minipig bladder. (Fig. 10m–o).

These findings suggest that the system has meaningful potential to enhance rapid drug delivery to the lesion site during electricity generation. Furthermore, H&E staining of clinical bladder (tumor) tissue samples revealed clear signs of cancer (Fig. 10p, q and Supplementary Table 2). Primary tumor cells were extracted from these clinical samples and treated with different experimental groups (Fig. 10r). Calcein AM/PI staining results showed that BTO-CPT/FA exhibited a more pronounced inhibitory effect on tumor cells compared to the other groups (Fig. 10s, t). These results indicate that the polarized BTO-CPT/FA system induces a ROS storm, which facilitates the release of the chemotherapy drug CPT and enhances bladder tissue penetration. This system demonstrates strong inhibitory effects on both in situ rabbit bladder cancer and cells derived from clinical tumor samples, while exhibiting minimal toxicity to other organs.

## Discussion

The self-driven electrical triggering system (BTO-CPT/FA) presents a groundbreaking approach to overcoming the persistent challenges of chemotherapy for bladder cancer, such as poor drug penetration and uncontrolled release. By harnessing intravesical pressure to generate electric fields, the BTO-CPT/FA system represents a marked advancement in the field of nanomedicine. This spontaneous electric generation facilitates the formation of tunneling nanotubes, which serve as high-speed conduits for drug delivery among bladder cancer cells. The enhanced permeability achieved through these nanotube highways ensures deeper and more uniform drug penetration into the tumor microenvironment, effectively addressing one of the primary limitations of conventional intravesical chemotherapy.

Moreover, the piezoelectric-induced electric fields stimulate the generation of ROS, which play a dual role in inducing apoptosis and disrupting the tumor microenvironment. Orthotopic animal model experiments have demonstrated considerable tumor suppression, validating the system's efficacy in improving therapeutic outcomes. These results underscore the synergistic benefits of combining targeted drug delivery with ROS-mediated tumor cell eradication within a single platform. The potential of this system could be further enhanced through combination therapies, such as immunotherapy or radiotherapy. By modulating the tumor microenvironment and improving drug delivery, the BTO-CPT/FA system could serve as a cornerstone for integrated cancer treatments.

The versatility of the BTO-CPT/FA system extends beyond bladder cancer. Its foundational principles—self-driven electric triggering, tunneling nanotube formation, and ROS-mediated cytotoxicity—can be adapted for other malignancies requiring localized and precise drug delivery. Furthermore, the targeting capability provided by folic acid can be replaced or augmented with other ligands to address various cancer types or even non-cancerous diseases.

Despite these promising results, several challenges must be addressed to optimize the clinical application of the BTO-CPT/FA system. One key issue is the system's reliance on internal forces for activation, which could be addressed by using ultrasound in the absence of these forces. Additionally, the molecular mechanisms governing the formation and stabilization of tunnel nanotubes remain unclear. Future studies focusing on the specific genes and proteins involved will provide valuable insights into optimizing nanotube-mediated drug delivery. Another important consideration is the regulation of ROS. While ROS production can promote apoptosis in tumor cells, excessive ROS levels may damage surrounding healthy tissues and exacerbate inflammation. Fine-tuning the system to regulate ROS levels and balance therapeutic efficacy with safety is crucial.

The development of the self-driven electrical triggering system (BTO-CPT/FA) represents a noteworthy advancement in the treatment of bladder cancer. By leveraging intravesical pressure to generate electricity, this system enables rapid drug delivery through tunneling nanotubes and enhances CPT absorption by tumor cells. Additionally, the production of ROS under controlled voltage further amplifies its anti-tumor efficacy, effectively eliminating bladder cancer cells. Preclinical studies in an orthotopic animal model demonstrated the potential of this system to overcome the limitations of current intravesical chemotherapy, such as poor drug penetration and uncontrolled release. With its ability to improve drug delivery and therapeutic outcomes, this innovative approach holds great promise for transforming bladder cancer treatment and warrants further exploration in both research and clinical settings.

## Methods

All animal experiments were approved by the Ethical Review Committees of Sichuan Provincial People's Hospital and the University of Electronic Science and Technology of China (Institutional Animal Care and Use Committee (IACUC) number: 202514), following the standards set by the National Institutes of Health Guide for the Care and Use of Laboratory Animals. Mice were provided ad libitum access to food and water and housed in a specific pathogen-free (SPF) environment (a 12-hour light/12-hour dark cycle, temperatures of -22–25 °C with 40–60% humidity). All the maximal tumor size/burden in our experiments did not exceed the maximal tumor size/burden permitted (2000 mm$^3$). All cells, including MB49 cells (Mouse bladder carcinoma cells), MB49-Luc cells (Luciferase-expressing MB49), SV-HUC-1 (a type of bladder epithelial cell), and Vx2 (Rabbit squamous cell carcinoma cells), were purchased from the Cell Bank of the Chinese Academy of Sciences (Shanghai, China).

**Synthesis of tetragonal phase barium titanate (Tetragonal BTO)**
The preparation of barium titanate involved a two-step process: first, the cubic-phase barium titanate was synthesized via a hydrothermal reaction, followed by high-temperature calcination to obtain the tetragonal-phase barium titanate.

The synthesis of cubic-phase barium titanate (Cubic BTO) was carried out as follows: 0.01 mol of titanium n-butoxide was dissolved in 10 mL of anhydrous ethanol and placed in a Teflon-lined vessel, where it was thoroughly stirred. Subsequently, 4 mL of ammonia and 5 mL of deionized water were added, and the mixture was stirred for 15 minutes, resulting in the formation of a $Ti(OH)_4$ suspension. To prepare the $Ba(OH)_2$ solution, 0.015 mol of anhydrous $Ba(OH)_2$ was dissolved in deionized water by stirring at 70 °C until fully dissolved. This solution was then quickly added to the $Ti(OH)_4$ suspension under stirring, yielding a uniform, milky-white suspension. The reaction vessel was sealed and placed in a muffle furnace, where it was heated continuously for 24 hours. After cooling naturally to room temperature, the reaction products were collected, washed sequentially with acetic acid, ethanol, and deionized water, and then dried at 80 °C for 24 hours.

To obtain tetragonal-phase barium titanate (Tetragonal BTO) with piezoelectric properties, the dried cubic BTO was calcined at 1000 °C for 3 hours. After naturally cooling to room temperature, the barium titanate was thoroughly milled using a mortar. To enhance the surface hydrophilicity of the tetragonal BTO, 1.0 g of the material was dispersed in 250 mL of a 30 wt% hydrogen peroxide solution and reaction at 85 °C for 8 hours. The resulting product was collected by centrifugation at 13,000 g for 5 minutes, washed with deionized water, and dried at 60 °C for 12 hours. Unless otherwise specified in this paper, BTO refers to tetragonal-phase barium titanate (Tetragonal BTO).

**Synthesis of the BTO-CPT/FA**
CPT and FA-PEG-COOH were loaded onto the BTO surface via the Mitsunobu reaction. Briefly, 100 mg of BTO was dispersed in 100 mL of pre-cooled, ultra-dry acetone. Next, 0.03 mmol of triphenylphosphine and 0.01 mmol of CPT were sequentially added. Air was removed from the reaction system using a vacuum pump, and argon gas was introduced to maintain an inert environment.

Subsequently, 2 mL of acetone containing 0.03 mmol of DEAD was injected into the system, and the reaction was conducted in an ice-water bath for 6 hours in the dark. Then, acetone containing 0.01 mmol of FA-PEG$_{2000}$-COOH was added, and the reaction continued for 3 hours at 0 °C, followed by overnight incubation at 50 °C.

The reaction products were collected by centrifugation at 13,000 g and washed sequentially with acetone, anhydrous ethanol, and deionized water. The resulting product was denoted as BTO-CPT/FA. The synthesis of BTO-CPT followed a similar procedure, except that FA-PEG$_{2000}$-COOH was not added, with all other steps remaining unchanged.

The concentration optimization process of CPT in the BTO-CPT/FA nanoplatform was conducted based on the thermogravimetric analysis, specifically, the ratio of components can be referred to Supplementary Table 1.

In addition, to investigate whether the CPT binds to BTO through chemical coupling, (1) the ζ potential of BTO nanoparticles and CPT in acetone was examined by a nanoparticle size analyzer; (2) the UV-Vis absorption spectra (λ = 384 nm, referring to CPT) of the precipitate (BTO-CPT) were obtained after washing several times.

**Characterization of BTO-CPT/FA**
The TEM and SEM micrographs of BTO-CPT/FA were obtained using a Transmission Electron Microscope (JEM-F2000) and Scanning Electron Microscope (ZEISS-sigma300), respectively. The hydrodynamic size, polymer dispersity index (PDI), and ζ-potential were measured using a nanoparticle size analyzer (Malvern Zetasizer Nano S90).

UV-vis spectroscopy results were recorded with an Ultraviolet-Visible Spectrophotometer (UV-3600 Plus), while FT-IR spectra were acquired using a Fourier Transform Infrared Spectrometer (Thermo Fisher iS5). The contents of CPT and FA on BTO-CPT/FA were conducted on thermogravimetric analysis (HITACHI-STA 200), and XRD patterns were obtained using an X-ray Diffractometer (Rigaku Ultima IV). The real-time monitoring of the voltage generated by BTO-CPT/FA under external force was conducted using a Piezoelectric Force Microscope (MFP-3D).

**Preparation of PFM response measurement samples.** A solution of chitosan (CHT) in deionized water was prepared and spin-coated onto a silicon wafer (1 cm × 1 cm) at 15 g, followed by drying at 60 °C for 10 hours.

**Electrical properties measurements.** Piezoelectric responses and hysteresis loops were obtained using a ScanAsyst-enabled Dimension Icon Atomic Force Microscope (AFM, Bruker) equipped with a platinum-coated, silicon-based conductive AFM tip (Olympus OMCL-AC240TM) with a nominal spring constant of ~4.6 N/m and a free-air resonance frequency of 70 kHz. Output voltage and current measurements were performed using a Keithley 6514 high-resistance electrometer.

**Evaluation of degradation ability of BTO-CPT/FA for malachite green**
Polarized BTO-CPT/FA generates abundant reactive oxygen species (ROS) under ultrasound, which effectively degrades malachite green by breaking its C=C and C=N bonds. The absorbance at λ = 618 nm was measured to evaluate the ROS production capability of BTO-CPT/FA under ultrasound. The degradation of malachite green by BTO-CPT/FA under ultrasound was investigated in both deionized water and artificial urine.

**Degradation in deionized water.** For experiments in deionized water, 1 mg of BTO-CPT/FA was dispersed in 5 mL of deionized water containing 20 μM malachite green. The sonication conditions were set to 1.0 MHz, 1.0 W/cm², and a 50% duty cycle. Supernatant samples were collected at 0 min, 1.5 min, 3 min, 6 min, and 12 min, and their UV-vis spectra (500–750 nm) were recorded. As a control, 5 mL of deionized water containing 20 μM malachite green without BTO-CPT/FA was treated under identical ultrasound conditions.

Furthermore, to eliminate the potential influence of dissolved oxygen in the deionized water on the degradation of malachite green, the deoxygenated deionized water was selected to conduct the degradation experiments. The experimental conditions were maintained as mentioned above except for deionized water deoxygenation.

**Quantifying degradation efficiency.** To quantify the percentage degradation of malachite green by BTO-CPT/FA under ultrasound, the absorbance values of the supernatant at λ = 618 nm were recorded at different time intervals. The degradation efficiency was calculated using the following equations:

$$Degradation\ efficiency = \frac{OD_t}{OD_0} \times 100\%$$

Here, the $OD_t$ and $OD_0$ represent the absorbance values at t minutes (t = 0, 1.5, 3, 6, and 12) after ultrasound treatment and at 0 minutes (before ultrasound), respectively, measured at λ = 618 nm.

The procedure for malachite green degradation by BTO-CPT/FA in artificial urine under ultrasound followed the same steps as described above, except that deionized water was replaced with artificial urine. Additionally, to identify the primary reactive species responsible for malachite green degradation, specific scavengers were used. Tri-n-

butylamine (TBA), a scavenger for hydroxyl radicals (·OH); 1,4-Benzo-quinone (BQ), a scavenger for superoxide radicals ($·O_2^-$); and ethyle-nediaminetetraacetic acid disodium salt (EDTA-2Na), a scavenger for holes ($h^+$), were added individually to 5 mL of deionized water containing 1 mg of BTO-CPT/FA and 20 μM malachite green. The degradation rate of malachite green after 12 minutes of ultrasound treatment was calculated using the formula:

$$Degradation\ Rate = \frac{(OD_0 - OD)}{OD_0} \times 100\%$$

Here, $OD_0$ is the absorbance value of malachite green at λ = 618 nm at 0 minutes, and OD is the absorbance value at 12 minutes under ultrasound.

## Responsive drug release driven by a self-generated electrical triggering system

The controlled release of active drugs through a self-driven electrical triggering system is critical for advancing the application of nanoplatforms in in vivo studies.

**For a linear reciprocating motion.** 60 mg of BTO-CPT/FA was dispersed in 30 mL of artificial urine and then placed in a balloon to simulate the urine bladder. Under the force of a linear reciprocating motion device to simulate bladder contraction and expansion, and the release of the drug was monitored, specifically, polarized and non-polarized BTO nano-systems were established and performed a linear reciprocating motion for 30 min/day for seven days to monitor the cumulative release of CPT.

## Stability assessment of BTO-CPT/FA in artificial urine

To evaluate the stability of BTO-CPT/FA in artificial urine, the nano-platform was incubated in artificial urine at a final concentration of 100 μg/mL for 5 days at 37 °C. The hydrodynamic size, PDI and ζ potential were measured daily using a nanoparticle size analyzer to monitor any changes over time.

## The cellular uptake mediated by FA

**Synthesis of Cy5.5-labeled BTO-Cy5.5/FA and Cellular Uptake Studies.** The Cy5.5-labeled BTO-Cy5.5/FA was synthesized following these steps: 25 mg of BTO was dispersed in pre-cooled, ultra-dry chloroform. Sequentially, 0.01 mmol of triphenylphosphine and 0.001 mmol of Cy5.5-COOH were added. The reaction system was then evacuated of air using a vacuum pump, and argon gas was introduced to maintain an inert atmosphere. Following this, chloroform solution containing 0.01 mmol of DEAD was injected into the reaction system, which was allowed to react in an ice-water bath under dark conditions for 6 hours. Subsequently, chloroform solution containing 0.01 mmol of FA-PEG$_{2000}$-COOH was added, and the mixture was stirred for an additional 3 hours. The reaction continued for 8 hours at 60 °C. The products were collected by centrifugation and washed sequentially with chloroform and anhydrous ethanol. The synthesis of BTO-Cy5.5 was identical to that of BTO-Cy5.5/FA, except without the addition of FA-PEG$_{2000}$-COOH. The loading amount of Cy5.5 on BTO was determined using a standard curve based on the UV-vis absorption of Cy5.5 at λ = 650 nm.

**Cell culture.** MB49 cells, MB49-Luc cells, and SV-HUC-1 cells were cultured in Dulbecco's Modified Eagle Medium (DMEM, GIBCO, South America) supplemented with 10% (v/v) fetal bovine serum (FBS, HyClone, USA), 1% (v/v) penicillin, and 1% (v/v) streptomycin. The cells were maintained in a humidified incubator (Thermo Fisher Scientific Inc., USA) at 37 °C with 5% $CO_2$.

**Cellular endocytosis evaluation.** To assess the cellular uptake, MB49 cells were seeded into confocal petri dishes at a density of $1 \times 10^5$ cells per well and incubated for 36 hours at 37 °C in a humidified environment. Following this, serum-free DMEM medium containing Cy5.5, BTO-Cy5.5, and BTO-Cy5.5/FA (with the same concentration of BTO: 100 μg/mL) was added to the cells, and the cultures were incubated for 0.5, 1, and 2 hours, respectively. Afterward, the cells were fixed with 4% (v/v) paraformaldehyde for 20 minutes at room temperature and washed with PBS. To stain the cell membranes, Dil, a cell membrane dye, was added and incubated with the cells for 30 minutes at 37 °C. The nuclei were stained with DAPI for 25 minutes at 37 °C. The fluorescence images of Cy5.5 (Ex/Em = 673 nm/707 nm), Dil (Ex/Em = 549 nm/565 nm), and DAPI (Ex/Em = 340 nm/488 nm) were captured using confocal microscopy. The relative fluorescence intensity of Cy5.5 was analyzed using ImageJ software.

## Evaluation of anti-tumor effect of BTO-CPT/FA in vitro

The anti-tumor assay of BTO-CPT/FA in vitro includes live and dead cell staining, ROS detection, cytotoxicity analysis, and cell migration assessment.

**Live and dead cell staining assay.** MB49 cells were seeded into confocal petri dishes at a density of $1 \times 10^5$ cells per well and incubated for 48 hours at 37 °C. The cells were then divided into the following groups: control, BTO, BTO + US, CPT, BTO-CPT/FA, and BTO-CPT/FA + US. MB49 cells in the experimental groups were incubated with serum-free DMEM medium containing BTO, BTO-CPT/FA, and CPT (all at a concentration of 100 μg/mL BTO) for 4 hours. For the BTO + US and BTO-CPT/FA + US groups, ultrasound treatment was applied for 3 minutes (1.0 MHz, 1.0 W/cm², 50% duty cycle). After the ultrasound treatment, a mixed staining solution containing Calcein-AM and propidium iodide (PI) at a final concentration of 2 μM was added to the wells, and the cells were incubated at 37 °C in a cell culture incubator for 25 minutes. Fluorescence images were then captured using a laser confocal microscope, observing the Calcein-AM (Ex/Em = 490 nm/515 nm) and PI (Ex = 535 nm, Em = 617 nm) channels. The relative fluorescence intensity of dead cells (PI channel) was analyzed using ImageJ software.

**Reactive Oxygen Species (ROS) Assay.** Intracellular ROS levels were detected using Dichlorofluorescein diacetate (DCFH-DA). After MB49 cells were cultured in confocal dishes for 48 hours, they were divided into the following groups: control, BTO, BTO + US, CPT, BTO-CPT/FA, and BTO-CPT/FA + US (with a constant BTO concentration of 100 μg/mL). The cells were incubated with the respective treatments for an additional 6 hours. For the BTO + US and BTO-CPT/FA + US groups, ultrasound was applied for 3 minutes (1.0 MHz, 1.0 W/cm², 50% duty cycle). Afterward, PBS containing 20 μg/mL DCFH-DA was added, and cells were incubated at 37 °C for 20 minutes. Before confocal microscopy observation, MB49 cells were stained with Hoechst 33342 (5 μg/mL). Fluorescence images of the DCF (Ex/Em = 488 nm/525 nm) channel were captured using a laser confocal microscope.

**Cytotoxicity assay.** Cytotoxicity was assessed using a standard 3-(4,5-dimethylthiazol-2-yl)−2,5-diphenyltetrazolium bromide (MTT) assay. MB49 cells were seeded into 96-well plates at a density of $5 \times 10^3$ cells per well and allowed to adhere for 36 hours. The cells were then incubated with various concentrations of BTO, CPT, and BTO-CPT/FA, with ultrasound applied to the BTO + US and BTO-CPT/FA + US groups for 4 minutes (1.0 MHz, 1.0 W/cm², 50% duty cycle). After another 18 hours of incubation, cytotoxicity was assessed using the MTT method.

**Cell migration assay.** MB49 cells were seeded in six-well plates at a density of $2 \times 10^6$ cells per well and incubated for 12 hours. The cells

were then incubated with serum-free DMEM containing BTO and BTO-CPT/FA (both at a concentration of 200 μg/mL BTO) for 6 hours. For the BTO + US and BTO-CPT/FA + US groups, ultrasound was applied for 3 minutes (1.0 MHz, 0.5 W/cm², 50% duty cycle). After incubation, the medium was replaced with DMEM containing 1% (v/v) fetal bovine serum, and wound width was recorded at 0 h, 24 h, and 36 h. The colony area within the wound was quantified using ImageJ software at each time point.

**Biodistribution of BTO-Cy5.5/FA.** Female C57BL/6 mice were purchased from SPF Biotechnology (Beijing, China) (5-7 weeks old). The biodistribution of Cy5, BTO-Cy5.5, and BTO-Cy5.5/FA was evaluated in orthotopic bladder cancer mice, which were randomly divided into three groups (n = 3). Seven days post-inoculation of the bladder wall with $1 \times 10^6$ MB49-Luc cells, the bladder was infused with Cy5, BTO-Cy5.5, and BTO-Cy5.5/FA (10 mg/kg BTO). Fluorescence signals were monitored at various time points (0, 4, 8, 12, 24, and 36 hours) using the IVIS Lumina II imaging system (Xenogen, USA). After 36 hours, the mice were euthanized, and the bladder tumor tissues and major organs were harvested for imaging. Fluorescence intensity in the tumor tissues and major organs was recorded using the region of interest (ROI) tool. The tumor tissues were embedded in optimal cutting temperature (OCT) compound, sectioned at 7 μm thickness, and stained with DAPI to visualize cell nuclei. Confocal microscopy images of Cy5.5 (Ex/Em = 673 nm/707 nm) and DAPI (Ex/Em = 340 nm/488 nm) fluorescence were acquired. Panoramic images of the bladder were obtained post-staining, and the fluorescence intensity of Cy5.5 in the bladder tumor tissues was quantified using ImageJ software.

## Molecular docking of folic acid and folate receptor

The interaction between folic acid and the folate receptor was evaluated using the molecular docking method in AutoDock Vina. Briefly, the folate receptor (ID: P35846) was downloaded from UniProt, and the structure of folic acid was obtained from PubChem. The folate receptor was designated as the receptor, while folic acid served as the ligand. The receptor-ligand complex was prepared by using PyMOL to separate the original ligand and protein structures, remove organics, and dehydrate the molecules. Hydrogenation, charge checking, and calculations were performed using AutoDock Tools.

After the docking process was completed using AutoDock Vina, the binding scores for the protein-small molecule interaction were calculated. The interaction forces were analyzed and visualized in both 3D and 2D formats using PyMOL and Discovery Studio software. The atom types were assigned to AD4 types, and docking grid boxes were constructed for the protein structures. Finally, the docking combination scores between folic acid and the folate receptor were calculated, and the interaction forces were visualized using PyMOL and Discovery Studio software.

## Detection of ROS generation by BTO-CPT/FA driven by intravesical pressure

The intravesical pressure in both orthotopic bladder cancer mice and tumor-free mice in vivo, as well as in mice with different bladder volumes ex vivo, was investigated using a manometer. Briefly, orthotopic bladder cancer mice were inoculated with $1 \times 10^6$ MB49-Luc cells in the bladder wall and allowed to grow for 6 days, while healthy mice served as controls (n = 6). The intravesical pressure was monitored using a manometer placed 0.5 cm vertically from the lower abdominal area where the mouse bladder was located. For the analysis of intravesical pressure ex vivo, the pressure was measured using the same manometer at a 0.5 cm perpendicular distance from the isolated bladder. The intravesical pressure at this location was recorded. Next, the urine from the bladders of control mice was drained, and the bladders were infused with 50 μL, 80 μL, and 110 μL of artificial urine.

The intravesical pressure for each volume was measured using the same experimental method.

Subsequently, 7-week-old tumor-free female C57BL/6 mice were randomly divided into two groups (n = 3). The bladders were surgically exposed, and 30 μL of saline containing 1 mg/mL BTO-CPT/FA and 20 μM DCFH-DA was injected into the bladder wall. In the control group, 30 μL of saline containing only 20 μM DCFH-DA was injected. Following the injection, the bladders were infused with 50 μL of artificial urine, and the mice were kept in a water-deprived environment for 12 hours. After this period, the mice were euthanized, and the bladders were extracted. The bladder wall of the mice was rinsed with normal saline, and the fluorescence intensity of DCF in the bladder wall was quantitatively analyzed (Ex/Em = 488 nm/525 nm).

## Simulation of intravesical pressure for electricity production assay in vitro
**Fabrication of a piezoelectric nanogenerator.** To fabricate the piezoelectric nanogenerator, a silicone elastomer base and curing agent were mixed in a clean glass beaker at a weight ratio of 10:1. The PDMS solution and BTO crystal powder were then uniformly mixed using probe ultrasound for 5 minutes at concentrations of 10, 20, 30, and 40 wt%. The piezoelectric composite device was formed by drop-casting the PDMS solution and BTO into a 3D-printed mold (20 × 15 × 0.7 mm). The PDMS and BTO mixed solution was placed in a vacuum oven for 30 minutes to degas. After degassing, the mixture was solidified by curing at 80 °C for 3 hours in the oven. A second composite film was fabricated similarly, but with BTO replaced by BTO/CPT/FA crystal powder. Al/PET electrodes were placed over the PDMS and BTO composite for piezoelectric measurements. Finally, copper wires were connected to the conductive Al electrodes for device testing.

**For COMSOL multi-physics simulation for piezoelectric response and mechanical tests.** The design and optimization of the flexible piezoelectric nanogenerator were performed using AutoCAD 2024 software. The simulation utilized the Solid Mechanics, Electrostatics, and Electrical Circuits modules in COMSOL Multiphysics v.5.6. The Solid Mechanics module incorporated piezoelectric effects into the model. Device dimensions were measured using a digital vernier caliper, as shown in Fig. S3. For the calculations, material parameters of the piezoelectric BTO were obtained from the COMSOL Multiphysics® Material Library. The simulation focused on the piezoelectric voltage distribution of the fabricated BTO-based device under the compression and expansion of the bladder. A compressive pressure (1-5 KPa) was applied to the model to study the piezoelectric response and the corresponding stress.

## The mechanism of the system for high-speed drug delivery through the tunneling nanotube
**Cell-based SEM images.** The MB49 cells, at a density of $8 \times 10^3$, were seeded on 6-well plates preplated with circular coverslips (φ10) and incubated for 48 hours at 37 °C in a cell culture incubator. Subsequently, the cells were fixed with glutaraldehyde, and the SEM images were obtained in accordance with standard procedures.

**Brightfield observation.** The MB49 cells with a density of $8 \times 10^3$ were seeded onto confocal petri dishes and incubated for 48 hours at 37 °C in a cell culture incubator. The brightfield images of tunneling nanotubes were obtained via CLSM.

**The role of ultrasound on the entrance of BTO-CPT/FA into tunneling nanotubes.** MB49 cells, at a density of $8 \times 10^3$, were seeded onto confocal petri dishes and incubated for 48 hours at 37 °C in a cell culture incubator. Thereafter, serum-free DMEM medium containing

BTO-Cy5.5/FA (50 μg/mL) was added to the cells and incubated for an additional 1 hour. Experimental groupings including, BTO-Cy5.5/FA (control group) and BTO-Cy5.5/FA+ultrasound group. For BTO-Cy5.5/FA+ultrasound group, ultrasound was applied for 30 seconds (1.0 MHz, 0.5 W/cm², 50% duty cycle), whereas the control group received without other treatment. Subsequently, the cell membrane dye, Dil (10 μM), was added to the dishes and incubated for 30 minutes at 37 °C. The MB49 cells were then stained with Hoechst 33342 (5 μg/mL) at 37 °C for 20 minutes. Fluorescence images of Cy5.5 (Ex/Em = 673 nm/707 nm), Dil (Ex/Em = 549 nm/565 nm) and Hoechst 33342 (Ex/Em = 340 nm/488 nm) were captured using CLSM.

**The effect of Cytochalasin D on BTO-CPT/FA NPs transport through tunneling nanotubes.** To investigate the role of Cytochalasin D (a common inhibitor of tunneling nanotube formation) on tunneling nanotube formation, MB49 cells (at a density of $3 \times 10^4$) were incubated for 48 hours at 37 °C in a cell culture incubator. Afterward, Cytochalasin D (2 μg/mL) was added, and the cells were co-cultured for an additional 4 hours. The control group received no treatment. Brightfield images of the MB49 cells were captured using confocal laser scanning microscopy (CLSM).

To assess whether BTO-CPT/FA NPs, transported through tunneling nanotubes, induced cell death in surrounding cells (i.e., antitumor effects), MB49 cells treated with BTO-CPT/FA NPs were co-incubated with untreated MB49-Luc cells (labeled with luciferase). The intensity of bioluminescence was then measured. The experimental groups included:(1) MB49-Luc cells (i1); (2) MB49 cells + MB49-Luc cells (i2); (3) Cytochalasin D + MB49-Luc cells (i3); (4) MB49 cells (BTO-CPT/FA NPs) + MB49-Luc cells + ultrasound (i4); (5) MB49 cells (BTO-CPT/FA NPs) + MB49-Luc cells + Cytochalasin D + ultrasound (i5); (6) MB49-Luc cells (BTO-CPT/FA NPs) + ultrasound (i6). The experimental procedure for the group "MB49 cells (BTO-CPT/FA NPs) + MB49-Luc cells + Cytochalasin D + ultrasound" was as follows:

a)  MB49 cells (at a density of $1 \times 10^4$) were incubated for 24 hours at 37 °C in a cell culture incubator. BTO-CPT/FA NPs (200 μg/mL) were then added, and the cells were incubated for an additional 12 hours. The residual NPs were removed by washing with PBS.

b)  The MB49 cells (BTO-CPT/FA NPs) were then co-incubated with MB49-Luc cells (at a density of $2 \times 10^3$) for another 12 hours. Subsequently, Cytochalasin D (2 μg/mL) was added, and the cells were incubated for 4 hours. Ultrasound (1.0 MHz, 1.0 W/cm², 50% duty cycle, for 4 minutes) was then applied.

c)  After the incubation, the supernatant was replaced with fresh medium containing luciferin potassium salt (500 μg/mL) and incubated for 10 minutes at 37 °C.

d)  Bioluminescence intensity was measured using a microplate reader (SpectraMax iD3, USA) with an integration time of 20 seconds and a read height of 1 mm. The experimental conditions for the other groups followed the same procedure, with the necessary adjustments to the respective components.

**Real-time single particle tracking (SPT) of BTO-Cy5.5/FA NPs inside tunneling nanotubes.** The MB49 cells with a density of $3 \times 10^4$ were seeded onto confocal petri dishes and incubated for 48 hours at 37 °C in a cell culture incubator. In order to have a more comprehensive insight into the transport of BTO-Cy5.5/FA NPs among MB49 cells through tunneling nanotubes, free Cy5.5 or BTO-Cy5.5/FA NPs (50 μg/mL) were added and co-incubated with MB49 cells for 20 min, residuae was removed by PBS, subsequently ultrasound was applied (1.0 MHz, 0.5 W/cm², 50% duty cycle, for 30 seconds), and the process of BTO-Cy5.5/FA NPs movement along the tunneling nanotubes was monitored by real-time imaging of live cells (Tokai Hit, Japan). Meanwhile, the free Cy5.5 group and BTO-Cy5.5/FA NPs group without ultrasound were selected as control groups, the other experimental

condition kept the same with the BTO-Cy5.5/FA NPs + ultrasound group. Images were captured at 1-min intervals for 10 times, which Cy5.5 (Ex/Em = 673 nm/707 nm), Dil (Ex/Em = 549 nm/565 nm) and Hoechst 33342 (Ex/Em = 340 nm/488 nm).

**Fluorescence recovery after photobleaching (FRAP).** In this experiment, the free Cy5 group and BTO-Cy5/FA NPs group without ultrasound were selected as control groups, the BTO-Cy5/FA NPs + ultrasound group as the experimental group, the other experimental condition stays consistent. Specifically, the MB49 cells with a density of $5 \times 10^4$ were seeded onto confocal petri dishes and adhesion for 48 hours. Subsequently, free Cy5 or BTO-Cy5/FA NPs (50 μg/mL) were added and co-incubated with MB49 cells for 6 h, residue was removed by PBS, subsequently ultrasound was applied (1.0 MHz, 0.5 W/cm², 50% duty cycle, for 30 seconds), and the process of a region is randomly selected on a singular cell as the photobleaching region, and an equal-sized blank region is established surrounding it as a control, and the process of *FRAP* was monitored by real-time imaging of live cells (Tokai Hit, Japan). The laser intensity of bleaching was adjusted to 100% (640 nm), and the bleaching process was terminated when the fluorescence in the bleached area decreased to 30% of the initial value, which Cy5.5 (Ex/Em = 649 nm/670 nm), Hoechst 33342 (Ex/Em = 340 nm/488 nm).

**The selective targeting and biocompatibility evaluation of BTO-CPT/FA nano-system in vitro**

**Sensitivity of normal bladder epithelial cells (SV-HUC-1) and bladder cancer cells (MB49) to ROS.** The calcein-AM/PI kit was used to monitor the apoptosis of the MB49 cell and SV-HUC-1 cell. After MB49 cells and SV-HUC-1 cells with a density of $1 \times 10^5$ were cultured in confocal dishes for 48 hours, they were incubated with BTO-CPT/FA NPs (20 μg/mL) for another 4 h. For the BTO-CPT/FA+ ultrasound group, ultrasound was applied for 3 minutes (1.0 MHz, 1.0 W/cm², 50% duty cycle). After the ultrasound treatment, a mixed staining solution containing Calcein-AM and propidium iodide (PI) at a final concentration of 2 μM was added to the wells and incubated for 25 minutes. Fluorescence images were then captured using a laser confocal microscope, observing the Calcein-AM (Ex/Em = 490 nm/515 nm) and PI (Ex = 535 nm, Em = 617 nm) channels.

**Mitochondrial membrane potential.** MB49 cells and SV-HUC-1 cells with a density of $1 \times 10^5$ were cultured in confocal dishes for 48 hours. They were incubated with BTO-CPT/FA NPs (20 μg/mL) for another 4 h. For the BTO-CPT/FA+ ultrasound group, ultrasound was administered for 3 minutes (1.0 MHz, 1.0 W/cm², 50% duty cycle). After the ultrasound treatment, a mixed staining solution containing Annexin FITC and Mito-Tracker Red CMXRos was added to the wells and incubated for 20 minutes. Fluorescence images were then captured using a laser confocal microscope, observing the Mito-Tracker Red CMXRos ((Ex/Em:579 nm/599 nm)) and Annexin FITC (Ex = 492 nm, Em = 520 nm) channels.

**Calcium ion channels.** Calcium ion probe (Fluo-4 AM) are employed to evaluate whether calcium ion channels are activated under different conditions. MB49 cells and SV-HUC-1 cells with a density of $1 \times 10^5$ were cultured in confocal dishes for 48 hours. They were incubated with BTO-CPT/FA NPs (20 μg/mL) for another 4 h. For the BTO-CPT/FA+ ultrasound group, ultrasound was administered for 3 minutes (1.0 MHz, 1.0 W/cm², 50% duty cycle). After the ultrasound treatment, a mixed staining solution containing Fluo-4 AM and Hoechst 33342 with a final concentration of 10 μM was added to the wells and incubated for 20 minutes at 37 °C. Fluorescence images were then captured using a laser confocal microscope, observing the Fluo-4 AM ((Ex/Em:488 nm/520 nm)) and Hoechst 33342 (Ex/Em = 340 nm/488 nm) channels.

### The selective targeting and biocompatibility evaluation of BTO-CPT/FA nano-system in vivo

The control mice (healthy mice) and orthotopic bladder cancer female C57BL/6 mice (8 weeks old) were administered BTO-CPT/FA every two days ($n = 3$, 15 mg/kg BTO) via bladder perfusion, and the process continued for 3 times. Post-administration, blood was collected via the eyeball. Subsequently, the mice were sacrificed and dissected, and bladders were collected from each group, too. In order to evaluate the selective targeting and biocompatibility of BTO-CPT/FA to healthy and bladder cancer mice, blood-related parameters were analyzed, and paraffin-embedded sections of the bladder were sliced to a thickness of 7 μm and stained with H&E.

### Investigation of the endocytosis mechanism

In order to compare systematically our delivery system with classical delivery pathways (e.g., passive diffusion, clathrin-mediated endocytosis, and caveolin-dependent endocytosis). Firstly, we synthesized mesoporous silica (MS NPs) and gold nanoparticles (Au NPs), which are two types of nanosized carriers that rely on clathrin-mediated endocytosis and caveolin-mediated endocytosis.

**Preparation of nanomaterials.** In short, the synthesis procedures for Cy5.5-modified mesoporous silica (MS-Cy5.5 NPs) about 40 nm were as follows: 16 ml of CTAB solution (Wt: 25%) and 90 mg of triethanolamine (TEA) were adder to 20 ml of deionized water, and stirring at 60 °C for 1 h, then 1.5 ml of TEOS and 1 ml of cyclohexane were added slowly below the liquid surface, and continued stirring at 60 °C for 18 h, then centrifuged at 13,000 $g$ to collect the precipitate. Subsequently, to conjugate Cy5.5, 100 mg of dried mesoporous silica was dispersed in toluene, followed by the addition of 4 ml of amino silane coupling agent. The system was then refluxed at 110 °C for 48 h to obtain amino-modified mesoporous silica. Finally, 100 mg of the amino mesoporous silica was dispersed in anhydrous ethanol, 5 mg Cy5.5 was added, and the system was reacted overnight at 50 °C. In addition, to prepare Cy5.5-modified gold nanoparticles (Au-Cy5.5 NPs) with a size of 33 nm, initially, 10 mg of HAlCl$_4$ was dissolved in 100 ml of deionized water, and stirred at 130 °C for 10 min, then 3 ml of sodium citrate solution (Wt: 1%) was added, and the system continued for another 30 min, followed by cooling to r.t. Subsequently, 1 mg of Cy5.5 was added and allowed to stir for 12 h at r.t.

### Investigation of endocytosis mechanism

**For MS NPs.** MB49 cells (density: $1\times10^5$) were cultured in confocal dishes for 48 hours, subsequently, chlorpromazine (CPZ) with a final concentration of 2 mM was incubated with MB49 for another 30 min. The control group was treated without other treatment. And, cells were incubated with MS-Cy5.5 NPs (50 μg/mL) for another 2 h, then cells were washed with PBS three times, and the cells were stained with Hoechst 33342 for 20 min before being imaged by CLSM.

**For Au NPs.** MB49 cells with a density of $1\times10^5$ were cultured in confocal dishes for 48 hours, then methyl-β-cyclodextrin (MCD, 200 nM) incubated with MB49 for another 30 min at 37 °C, meanwhile, the control group without other treatment. Then, these MB49 cells were incubated with Au-Cy5.5 NPs (50 μg/mL) for another 2 h, subsequently, the cells were stained with Hoechst 33342 for 20 min before being captured by CLSM.

**For BTO-FA NPs.**
1. *Effect of inhibitors on endocytosis efficiency:* MB49 cells with a density of $1 \times 10^5$ were allowed to adhere in confocal dishes for 48 hours, cells were pretreated with CPZ (2 mM), MCD (200 nM) for another 30 min, respectively. Subsequently, cells were incubated with BTO-Cy5.5/FA NPs (50 μg/mL) for 4 h, finally, the Hoechst 33342 was added to stain for 20 min before imaged by CLSM.
2. *Effect of temperature on endocytosis efficiency:* MB49 cells (density: $1 \times 10^5$) were allowed to incubate in confocal dishes for 48 hours, subsequently, cells were treated with BTO-Cy5.5/FA NPs (50 μg/mL) for 4 h at 37 °C, while, the incubation temperature of 4 °C was selected as control group. After incubation, cells were stained with Hoechst 33342 for another 20 min.

### Evaluation of anticancer Effect of BTO-CPT/FA in vivo

The orthotopic bladder cancer female C57BL/6 mice (5-7 weeks old) were randomly assigned to control, CPT, BTO, BTO-CPT, and BTO-CPT/FA groups. The mice were administered PBS, CPT, BTO, BTO-CPT, or BTO-CPT/FA every two days ($n = 5$, 15 mg/kg BTO) via bladder perfusion, starting on the third day after the inoculation of $4 \times 10^5$ MB49-Luc cells into the bladder wall. The bioluminescence intensity of MB49-Luc cells was recorded every five days using IVIS Lumina II and continued for 25 days, with body weight measurements taken daily.

At 25 days post-administration, the mice were sacrificed and dissected. Bladder tumor tissues and major organs, including the heart, liver, spleen, kidney, and lung, were collected from each group. To assess biosafety, paraffin-embedded sections of major organs were sliced to a thickness of 7 μm and stained with H&E.

**For therapeutic evaluation.** Bladder tumor tissues were sectioned (5 μm thick), stained with H&E, and panoramic images of the tumor tissue were obtained. To examine the level of ROS in bladder tumor tissues after treatment, frozen sections were stained with DCFH-DA (10 μM) at 37 °C for 30 minutes, followed by DAPI staining. Fluorescence images of DCF (Ex/Em=488 nm/525 nm) were captured using CLSM.

**For DNA fragmentation detection.** Terminal deoxynucleotidyl transferase-mediated dUTP nick-end labeling (TUNEL) staining was performed following the manufacturer's instructions (Beyotime Biotechnology, China).

**Evaluation of the tumor proliferation.** Tissue sections were fixed with 4% paraformaldehyde for 10 minutes at room temperature, permeabilized with 0.2% Triton-X 100 for 5 minutes, and blocked with 2% BSA for 1 hour. The sections were then incubated with rabbit anti-mouse Ki67 monoclonal antibody (1:100 dilution) for 2 hours at room temperature, followed by staining with FITC-labeled sheep anti-rabbit antibody (1:500 dilution) for 1 hour at 37 °C. Finally, the sections were stained with DAPI. Fluorescence images of FITC (Ex/Em=488 nm/520 nm) were obtained using CLSM.

### Sequencing of the eukaryotic transcriptome of the bladder tumor tissues

Total RNA was extracted from bladder tumor tissues of the control and BTO-CPT/FA groups using TRIzol reagent. Complementary DNA was synthesized following the standard reverse transcription procedure (EASYspin Plus, hxbio, China). Three replicate samples were prepared for each group, and RNA sequencing was performed by Berry Genomics (Beijing, China).

### Evaluation of anti-tumor effects of BTO-CPT/FA in vitro from the clinical bladder cancer patients

Normal bladder and bladder cancer clinical samples were provided by doctors from the Department of Oncology, Sichuan Provincial People's Hospital, which obtained ethical review and approval (Institutional Animal Care and Use Committee (IACUC) number: 202514). Participants were not specifically recruited for this study; healthy and tumor tissue from patients' samples were provided by Department of Oncology, Sichuan Provincial People's Hospital. Patient sample

information is detailed in Supplementary Table 2. To observe the histopathological structure of normal bladder tissue and bladder tumors, the clinical samples were sectioned into 7 μm paraffin slices and stained with hematoxylin/eosin using a standard staining kit. Additionally, primary bladder tumor cells were extracted from tumor tissues by trypsin digestion. The primary bladder tumor cells were seeded into confocal petri dishes at a density of $1 \times 10^5$ cells per well and incubated for 48 hours at 37 °C in a humidified incubator.

The cells were then divided into six groups: control, BTO, BTO + US, CPT, BTO-CPT/FA, and BTO-CPT/FA + US, and incubated with serum-free DMEM medium containing BTO, BTO-CPT/FA, and CPT (200 μg/mL) for 4 hours. Ultrasound was applied for 3 minutes (1.0 MHz, 1.0 W/cm², 50% duty cycle) to the BTO + US and BTO-CPT/FA + US groups. After incubation, the cells were stained with Calcein-AM and PI at 37 °C for 30 minutes. Fluorescence images of the Calcein-AM (Ex/Em = 490 nm/515 nm) and PI (Ex = 535 nm, Em = 617 nm) channels were captured using CLSM. The ratio of live to dead cells was analyzed using ImageJ software.

### Evaluation of the anti-tumor effect of BTO-CPT/FA in orthotopic bladder cancer rabbit model

Female New Zealand rabbits, with an average weight of 2.5 kg, were employed to evaluate the anti-tumor effect of BTO-CPT/FA. Prior to inoculation of Vx2 cells in the bladder wall, the rabbits underwent abdominal skin preparation, and water intake was withheld. The following day, the rabbits were anesthetized by intravenous injection of 30 mg/kg sodium pentobarbital via the ear vein. The orthotopic bladder cancer rabbit model was established by injecting 200 μL of serum-free DMEM containing $5 \times 10^6$ Vx2 cells into the bladder wall. To prevent wound infections, daily intramuscular injections of 10 mg/kg potassium penicillin were administered for one week. The model was then randomly divided into control and BTO-CPT/FA groups ($n = 3$). Bladder infusions of either 10 ml saline or 10 ml saline containing BTO-CPT/FA (15 mg/kg BTO) were administered one day apart for a period of 40 days.

At the end of the treatment, blood biochemical samples were collected by drawing 2 ml of blood from the ear vein. To euthanize the rabbits, a 20 wt% KCl solution was injected into the marginal ear vein. The heart, liver, spleen, lung, and kidney were isolated, sectioned into 7 μm slices, and stained with hematoxylin/eosin. Additionally, 5 μm paraffin sections of the isolated bladder (tumor) tissues were prepared and stained with hematoxylin/eosin to evaluate the therapeutic effects of BTO-CPT/FA.

### Evaluation of permeability in C57BL/6 mouse and minipig bladders to assess self-generated electrical triggering system

**Mouse bladder permeability experiments.** 6 week-old female C57BL/6J mice were used as study subjects. The bladders were perfused with 50 μl of BTO-Cy5.5 (unpolarized) and 50 μl of BTO-Cy5.5 (polarized) (400 μg BTO). The mice were allowed to survive for 12 hours without access to water. Afterward, the isolated bladder tissues were prepared into 7 μm frozen sections and stained with anti-CD31 antibody (FITC) and DAPI. Fluorescence images of Cy5.5 (Ex/Em = 673 nm/707 nm), FITC (Ex/Em = 488 nm/520 nm), and DAPI (Ex/Em = 340 nm/488 nm) were captured using confocal microscopy. The Cy5.5 fluorescence intensity in the bladder tissues was then quantitatively analyzed using ImageJ software.

**Mini pig bladder permeability experiment.** Female minipigs with an average weight of 2.0 kg were selected for the bladder permeability study. The bladders were infused with 15 ml of BTO-Cy5.5 (unpolarized) and 15 ml of BTO-Cy5.5 (polarized) (20 mg BTO) for 12 hours before the pigs were euthanized. The pigs were euthanized by intracardiac injection of 5 ml of KCl (20 wt%). Isolated bladder tissues were

sectioned into 7 μm slices and stained with DAPI. Fluorescence images of Cy5.5 (Ex/Em = 673 nm/707 nm) and DAPI (Ex/Em = 340 nm/488 nm) were obtained using CLSM.

### Statistical Analysis

All data are presented as means ± S.D. Statistical differences between the two groups were assessed using Student's t-test. For comparisons among multiple groups, one-way analysis of variance (ANOVA) followed by Sidak's multiple comparisons test was used. Two-way ANOVA with Tukey's multiple comparisons test was applied for comparisons across serial measurements or multiple conditions within different groups. All statistical analyses were two-sided, and p-values < 0.05 were considered statistically significant. Statistical calculations were performed using GraphPad Prism version 8.0 (GraphPad Software).

### Reporting summary

Further information on research design is available in the Nature Portfolio Reporting Summary linked to this article.

## Data availability

The main data supporting the results in this study are available within the paper and its Supplementary Information. Source data are provided with this paper. All data generated in this study are available from the corresponding authors on reasonable request. Source data are provided with this paper.

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

## Acknowledgements

This work was partially supported by the National Natural Science Foundation of China (No. 32401173, 82271120, 82571261, 82121003, 82502533), CAMS Innovation Fund for Medical Sciences (2019-I2M-5-032), Sichuan Academy of Medical Sciences & Sichuan Provincial People's Hospital (No. 30420220004), Sichuan Science and Technology Program (24GJHZ0072), Sichuan Province Science and Technology Activities Funding for Returned Overseas Scholars, the 15th Batch of Basic Research Funds for Central Universities and Colleges Frontier Cultivation Project (ZYGX2024J026), Shenzhen Science and Technology Program (20240808155422003), TOP Young Talents Program of Sichuan (No. DQ202407 to Y.S.), Chengdu Science and Technology program (2025-YF09-00047-SN), Department of Science and Technology of Sichuan Province, China (2024ZHYS0018 to Y.S.) and Jiangsu Hansoh Pharmaceutical Group Co., Ltd. All authors have accepted responsibility for the entire content of this manuscript and approved its submission. We would like to express our gratitude to Dr. Siyuan Wang from Sichuan Cancer Hospital for generously providing clinical bladder cancer samples.

## Author contributions

Z.-J. L. and Y.-B. M. conceived and designed the study. Y., S. and Z.-H. L. provided key insights for the design of experiments; Y.-B. M. supervised the project. Z.-J. L., R., J., Z.-G.Z., F.-L. L., Y., G., M.-Y. S., X.-X. L., T., J., L., Z. and S.-Y. W. performed the experiments. Z.-J. L., R., J., and Z.-G.Z. interpreted the data. Z.-J. L., Y.-B. M., R., J. and Z.-H. L. wrote the manuscript. All authors approved the final version of the manuscript.

## Competing interests

The authors declare no competing interests.

## Additional information

[1]Department of Haematology, Sichuan Academy of Medical Sciences & Sichuan Provincial People's Hospital, School of Medicine of University of Electronic Science and Technology of China, Chengdu, China. [2]Department of Urology, South China Hospital, Medical School, Shenzhen University, Shenzhen, China. [3]Guangdong Key Laboratory for Biomedical Measurements and Ultrasound Imaging, National-Regional Key Technology Engineering Laboratory for Medical Ultrasound, School of Biomedical Engineering, Shenzhen University Medical school, Shenzhen, China. [4]Department of Biomedical Engineering, National Taiwan University, Taipei, Taiwan. [5]Department of Power Mechanical Engineering, National Tsing Hua University, Hsinchu, Taiwan. [6]Department of Surgery & Cancer, Faculty of Medicine, Imperial College London, London, UK. [7]Sichuan Provincial Key Laboratory for Human Disease Gene Study and the Center for Medical Genetics, Department of Laboratory Medicine, Sichuan Academy of Medical Sciences and Sichuan Provincial People's Hospital, University of Electronic Science and Technology of China, Chengdu, China. [8]Research Unit for Blindness Prevention of Chinese Academy of Medical Sciences (2019RU026), Sichuan Academy of Medical Sciences and Sichuan Provincial People's Hospital, Chengdu, China. [9]School of Food and Biological Engineering, Chengdu University, Chengdu, China. [10]Urology (Department), Sichuan Clinical Research Center for Cancer, Sichuan. Cancer Hospital & Institute, Sichuan Cancer Center, Affiliated Cancer Hospital of University of Electronic Science and Technology of China, Chengdu, China. [11]Sichuan-Chongqing Joint Key Laboratory of Pathology and Laboratory Medicine, Jinfeng Laboratory, Chongqing, China. ✉e-mail: shiyi1614@126.com; zhlin@ntu.edu.tw; miaoyangbao@uestc.edu.cn

