## [Transparent Peer Review file · Nature Communications]

Self-driven Electrical Triggering System Activates Tunneling Nanotube Highways to Enhance Drug Delivery in Bladder Cancer Therapy

Corresponding Author: Professor Yang-Bao Miao

Version 0:

Reviewer comments:

Reviewer #1

(Remarks to the Author)

In this article, the authors proposed a self-driven electrical triggering system called BTO-CPT/FA. The synthesized material harnesses mechanical stimuli to electrically trigger the drug release of CPT to cure bladder tumors. Besides, to improve cell-targeting abilities, they have utilized folate acid (FA) to target the folate receptors in the bladder tumor membranes selectively. The authors demonstrated the electrical triggering system through various biological experiments. Based on their achievements, they have introduced an advancement of cell-targeted nanomedicines to treat bladder cancer. However, materials synthesis and operation remain questionable. Thus, this reviewer thinks that this article should be reconsidered after the major revision for the potential publication in Nature Communications. Below are the questions and comments that need to be addressed in the revision process.

Comment 1:

The authors showed that the CPT can be coated through the Mitsunobu reaction in the synthesized BTO particles, but this reviewer suggests the authors explain more about the fundamentals of chemical interactions between BTO and CPT. How can they form a stable nanostructure?

Comment 2:

In Figure 2i, CPT has a noticeable peak at 384 nm, but there are only small peaks at the same wavelength for both BTO-CPT/FA and BTO-CPT. This reviewer doubts that this significant gap is attributed to the weak chemical interactions between CPT and BTO. More specifically, how can you make sure that the CPTs form chemically stable interactions with BTO, not just physically absorbed in the surface such as residues?

Comment 3:

The meaning of the gray bars in Figure 2j is difficult to understand, so their expression should be modified.

Comment 4:

Figure 3a and b show electrical measurement settings and the results, but it is hard to understand how the electrical outputs were measured. The authors need to provide more detailed explanations for these measurement methods and results.

Comment 5:

In Figure 3k and 3l, the control samples show more decay in absorbance compared to the BTO-CPT/FA groups, but Figure 3m shows the opposite trend. This reviewer doubts if there are mistakes in the Figure captions.

Comment 6:

In this article, the authors tried to demonstrate that the CPTs can be released based on the piezoelectricity of the BTO nanoparticles. However, the release of CPTs is more likely attributed to the ultrasound irradiation. The authors need to provide more demonstrations to prove the electrically triggered drug release.

Comment 7:

In Figure 3r, the authors tried to demonstrate that the CPTs were stably captured in BTO nanoparticles under PBS buffer

solution for a few days. However, this reviewer thinks that the diameter and PDI values of the composite nanoparticles cannot prove it. This reviewer suggests the authors need to measure zeta potential changes for 5 days under PBS solution.

Comment 8:

In this article, ultrasound was employed to simulate the intravesical pressure. However, due to their significant differences in intrinsic physical properties between ultrasound and bladder movements, this reviewer does not agree that the ultrasound can be utilized to simulate pressure generated through the bladder movements.

Comment 9:

According to the results shown in Figures 4f, 4h, and 4k, it seems that ultrasound irradiation plays a more effective role in inhibiting tumor activities than the other chemicals. Please provide more explanations on the results.

Comment 10:

In Figures 6b and 6c, the authors described that the BTO-Cy5.5/FA shows better targeting abilities than the BTO-Cy5.5, because the decline in fluorescence intensity was slower in the FA coated groups. However, Figure 6c shows no noticeable differences between the two groups.

Comment 11:

Captions in Figure 6i are too small to be identified.

Comment 12:

The p-value expression shown in Figure 9c is not only aside from the commonly used format but also difficult to understand. Please revise it for clarity.

Comment 13:

All figure expressions and manuscript descriptions for Figure 10 are very difficult for potential readers to understand. The figure captions are too small to be easily readable, and the manuscript lacks sufficient explanation.

Reviewer #2

(Remarks to the Author)

The manuscript entitled “Self-driven Electrical Triggering System Activates Tunnel Nanotube Highways to Enhance Drug Delivery in Bladder Cancer Therapy” by Liu et al. presents a novel approach for bladder cancer treatment through a self-driven electrical triggering system (BTO-CPT/FA) that harnesses intravesical pressure to generate electricity. This system significantly enhances drug absorption by facilitating the transport of hydroxycamptothecin through tunneling nanotubes, thereby creating a rapid and efficient drug delivery route to tumor cells. Furthermore, the generated voltage promotes the production of reactive oxygen species (ROS), which contributes to the destruction of bladder cancer cells. The experimental data, including both in vitro and in vivo characterizations, support the conclusion drawn by the authors and confirm the functionality. However, despite the promising results, the overall quality of the manuscript—both in terms of the figures and the writing—does not yet meet the high standards of Nature Communications. Several major issues should be addressed before further consideration.

1. The discussion of FT-IR spectrum in the manuscript is overly simplistic and lacks sufficient detail, making it difficult to interpret the data in relation to the labels provided in Figure 2J. As a result, the current analysis does not allow for any meaningful conclusions regarding surface modifications based on the FT-IR results.
2. The figure legends require a thorough review and revision for accuracy and clarity. Specifically, the legends for Figures 3K and 3L are mistakenly swapped, and the legend for Figure 3O contains several inaccuracies. Furthermore, Figure 4A is missing a scale bar, and Figures 4B and 11B also lack scale bars. Additionally, the scale bar in Figure 9K is clearly incorrect and needs to be adjusted for proper measurement representation. These issues should be addressed to ensure consistency and accuracy across all figures and legends.
3. In Figure 3L, is the control group composed solely of deionized water? Could ultrasound also contribute to the degradation of malachite green? This raises concerns about the potential influence of dissolved oxygen in the deionized water on the observed results. Would using deoxygenated water be a more effective way to eliminate this potential interference?
4. The confocal images in the manuscript require optimization to improve clarity and consistency. For instance, the cell concentration within the field of view in Figure 4E appears uneven, which may lead to inconsistent interpretation, while the focus in Figure 4G is excessively blurry, hindering the clarity of the image and the accuracy of the analysis. Moreover, the DCF fluorescence signal in Figure 9H is nearly undetectable.
5. Several issues regarding the dosage remain unclear. The manuscript does not detail any optimization process for the CPT and FA concentrations in the BTO-CPT/FA nanoplatform. Additionally, the loading amount of Cy5.5 on BTO is not provided.
6. Previous reports, such as those in *Anal. Chem.* 2019, 91, 6996-7000, demonstrate that Cy5.5 is taken up by tumor cells when administered alone. However, in this manuscript (Figure 4B), this uptake is nearly absent. This discrepancy needs to

be addressed and explained in more detail.

7. In the cell viability assays, is there a statistically significant difference between CPT and BTO-CPT/FA with a p-value of **** $p < 0.0001$? Additionally, the difference between the 100 $\mu\text{g}/\text{mL}$ and 200 $\mu\text{g}/\text{mL}$ concentrations appears minimal. Does this suggest that doubling the dose does not result in a corresponding increase in cell death?

8. The simulation results in Figures 6H-J demonstrate the binding between FA receptors and FA. However, it remains unclear whether these findings are directly applicable when FA is modified onto BTO-CPT in the BTO-CPT/FA nanoplatform. This distinction should be clarified to ensure the relevance of the simulations to the experimental context.

9. The manuscript states that "an intravesical pressure-driven system in the bladder generates electricity," implying that bladder perfusion pressure is directly linked to therapeutic outcomes. However, this relationship is not clearly demonstrated in the in vivo experiments. How can it be ensured that perfusion pressures remain consistent across different groups when administering various drug formulations? Additionally, to what extent would variations in perfusion pressure affect the treatment efficacy?

10. The manuscript contains several minor formatting issues, including inconsistencies in citation style (e.g., sometimes only the first author is listed, while other times all authors are included). Additionally, abbreviations should be defined upon their first use to ensure clarity and consistency throughout the text.

11. The writing in this manuscript appears to be heavily influenced by AI editing, which has led to a somewhat unnatural and inconsistent writing style.

Reviewer #3

(Remarks to the Author)

This manuscript demonstrated a self-driven electrical triggering system (BTO-CPT/FA) for bladder cancer treatment. This system could release drugs (CPT) and generate reactive oxygen species (ROS) under electrical stimuli induced by internal bladder pressure. It showed remarkable anti-tumor effect and high-speed drug delivery through tunneling nanotubes. Furthermore, the authors demonstrate the efficacy of their system in various animal models (mice, rabbits, minipigs). The concept of utilizing internal bladder pressure to induce electricity is innovative. However, upon closer inspection, it is unclear if this system effectively addresses the limitations inherent in bladder cancer treatment. Several experimental gaps and questionable interpretations diminish the manuscript's overall strength. Consequently, the manuscript is not suitable for publication in Nature Communications. List of comments are shown as below.

1. BTO-CPT/FA system is an internal stimuli-responsive drug and ROS releasing system. Based on this perspective, the authors should emphasize their system compared to other drug delivery systems.

2. The authors mentioned that this system could form tunneling nanotubes and deliver drugs rapidly. Experimentally, they confirmed that tunneling nanotubes were formed and that drugs could be transported within the nanotubes. However, there is a lack of experiments to confirm that the delivered drug has an effect. The following experiments need to be conducted: (1) to prove the effect of tunneling nanotubes, it is necessary to compare the environment that inhibits the formation of tunneling nanotubes. (2) co-incubating cells that have treated nanoparticles with non-treated cells, to confirm if the drug is transferred to the non-treated cell, and whether the transferred drug is then effective.

Furthermore, it is unclear why these tunneling nanotubes allow for high-speed delivery. We also wonder how this system works, i.e., how the tunneling nanotubes are formed by electrical stimuli, what is the specific mechanism?

3. The manuscript lacks critical data on the characterizations of BTO-CPT/FA nanoparticles. Specific details on: drug loading capacity, FA coating efficiency.

4. We have a few questions about in vitro experiments (Fig. 4). The live-dead images and their quantifications do not match. For example, the BTP-CPT/FA + US groups is quantitatively reported as nearly 100% dead, but the corresponding image suggests otherwise. Re-experiments or clearer justifications are required.

Furthermore, in DCF (ROS levels) confocal images, the amounts of ROS seems to insufficient. Quantitative measurements of ROS levels using additional assays are recommended to corroborate the results.

5. The authors also developed a PDMS+BTO-CPT/FA device (Fig. 7). The performance of this device was evaluated, but the application in an animal model was not confirmed. The purpose of developing this device is unclear. Its application requires surgical intervention, potentially limiting its feasibility.

6. The samples were delivered via bladder perfusion instead of intravenous injection (i.v. injection). The authors should explain why this route was chosen and discuss its advantages or limitations compared to alternative methods.

7. Several figures contain errors or inconsistencies with the manuscript test: (1) In Fig. 2f, Ba signal (red) is barely visible in the merged data; (2) In Fig. 4h, the description of the y-axis does not match the text in caption; (3) In Fig. 9h, there is no DCF (ROS) signal (green).

Version 1:

Reviewer comments:

Reviewer #1

(Remarks to the Author)

The authors provided comprehensive responses to the reviewer's comment. They addressed the mechanistic concerns with additional experimental data (zeta potential analysis, UV-Vis, ROS generation, and tunneling nanotube transport validation).

The authors also improved the clarity of the figures, addressed formatting issues, and refined the mechanistic interpretations. Their verifications regarding the use of ultrasound to simulate intravesical pressure, and their explanation for the choice of bladder perfusion over systemic delivery, are well-reasoned and supported by both literature and experimental data.

Overall, the authors responded to the reviewer's concerns with clarity and thorough revisions. I believe the revised manuscript meets the high standards required for publication in Nature Communications, and I recommend its acceptance for publication.

Reviewer #2

(Remarks to the Author)

I appreciate the authors' efforts in conducting additional experiments and revising the manuscript accordingly. The revised manuscript now presents a more comprehensive analysis and discussion. However, there are still worrying problems in many details. Several minor concerns should be addressed before further consideration.

1. The supplementary figures should be re-labeled to follow the order in which they are cited in the main manuscript.
2. The authors note that it reaches a saturation point at 100 $\mu\text{g}/\text{mL}$. Therefore, subsequent cell experiments should not include a 200 $\mu\text{g}/\text{mL}$ dose. Additionally, more precise data—such as IC₅₀ values and in vivo metabolic curves—should be provided to support dose selection and enhance the reliability of the pharmacological analysis.
3. The MB49 bladder cancer in situ model is commonly established in male mice to avoid hormonal influence, with tumors typically forming within 2–3 weeks. In this study, however, female mice were used, and treatment began as early as day 3 post-inoculation, potentially before the tumor was fully formed. The authors should clarify whether these factors might affect the results.

Reviewer #3

(Remarks to the Author)

The authors present an innovative self-driven electrical triggering drug delivery system (BTO-CPT/FA) for bladder cancer therapy. This system leverages piezoelectric effects induced by intravesical pressure to enhance drug delivery via tunneling nanotubes (TNTs) and generate reactive oxygen species (ROS) for improved antitumor efficacy. The manuscript has undergone substantial revision and includes expanded experimental data, representing a meaningful advancement. However, several critical scientific and mechanistic concerns remain unresolved and must be addressed to substantiate the manuscript's key claims:

1. Incomplete validation of tunneling nanotube-mediated drug transport: While the authors added experiments demonstrating TNT formation inhibition by Cytochalasin D, direct quantitative evidence proving that drug transport specifically occurs through TNTs remains insufficient. Further experiments with clearly defined controls and quantitative tracking methods (e.g., real-time single-particle tracking, fluorescence recovery after photobleaching (FRAP)) are necessary to convincingly differentiate TNT-mediated transport from passive diffusion.
 2. Insufficient clarification of ultrasound-induced versus piezoelectric effects: The authors conducted comparative experiments with polarized and non-polarized nanoparticles; however, these experiments were still performed under ultrasound conditions, complicating interpretation. Clear experimental conditions isolating mechanical pressure-induced piezoelectric effects, free from confounding ultrasound influences (such as cavitation or acoustic streaming), must be included to substantiate the authors' claims.
 3. Lack of robust evaluation of ROS off-target effects: The manuscript highlights ROS generation as a significant therapeutic advantage, yet does not adequately address the potential for collateral damage to adjacent healthy tissues. Additional experiments, including comprehensive histological or biochemical assays evaluating ROS-induced effects on normal bladder tissue, are necessary to confirm selective targeting and safety.
 4. Absence of theoretical justification for biological relevance of generated voltage: The reported voltage range (~0.5–2.5 V) is substantially higher than typically used in similar systems (0.1–0.6 V), and may exceed levels that are physiologically safe or relevant. The manuscript does not demonstrate whether this voltage can influence membrane potential, ion channels, or signaling pathways. Without direct measurements of membrane potential shifts, ion channel modulation, or voltage-sensitive signaling activation, the biological applicability of the reported voltage remains speculative.
 5. Limited comparative evaluation of delivery strategies: While the manuscript highlights the advantages of TNT-mediated delivery, it lacks comparative analyses with established nanoparticle delivery mechanisms such as passive diffusion or endocytosis. Direct comparisons would be valuable to substantiate the claimed benefits of the proposed system.
- In summary, the manuscript presents a novel and compelling concept with promising potential. However, further experimental validation and clarification of mechanistic aspects would substantially strengthen the study and help align it with the publication standards of Nature Communications.

Version 2:

Reviewer comments:

Reviewer #2

(Remarks to the Author)

We are grateful to the author for their patient efforts in conducting additional experiments and making corresponding revisions to the manuscript. The revised manuscript now provides a more comprehensive analysis and discussion.

Moreover, it has appropriately responded to the rationality of the design of each experiment, which significantly enhances the credibility of the data in the paper. Based on the current results, we believe that it is suitable for publication in Nature Communications.

Reviewer #3

(Remarks to the Author)

The authors have adequately addressed the major concerns raised in the previous review through additional experiments and clearer explanations. The revised manuscript is substantially improved and presents a compelling case for the proposed therapeutic strategy.

To further enhance clarity and consistency, the following minor points should be addressed:

1. Physiological relevance of mechanical stimulation: The manuscript describes a bladder-mimicking compression model to induce piezoelectric drug release, but it remains unclear how the applied pressure and frequency correspond to physiological bladder conditions. Clarifying this point would strengthen the translational significance of the model.
2. Temporal dynamics of ROS generation: ROS production is assessed at a single time point following stimulation. A brief discussion of the expected time course of ROS induction and clearance, and whether the effect is transient or sustained, would improve mechanistic clarity.
3. Mechanistic explanation of FA-mediated targeting: The role of folic acid (FA) in enhancing tumor targeting is mentioned, but the underlying mechanism by which FA contributes to tumor selectivity is not clearly described. A short explanation of how FA enhances targeting, such as receptor-mediated uptake, would help clarify its specific contribution relative to passive accumulation.
4. Timing of ultrasound exposure in in vivo experiments is insufficiently described: In the in vivo studies, ultrasound stimulation is mentioned as part of the treatment protocol. However, the exact timing relative to nanoparticle injection, duration of exposure, and number of cycles are not clearly stated in the main text or methods section. This information is important for reproducibility and to understand the temporal dynamics of the activation strategy.

8, April 2025

Dear reviewers,

We sincerely appreciate the reviewers and editors for their thorough evaluation of our study. In the revised version of the manuscript, we have carefully addressed all the comments and suggestions raised by the reviewers (which are marked in blue in the revised manuscript). We believe that these revisions have significantly improved the quality and clarity of our work. Detailed responses to each comment and the corresponding revisions are provided below.

Reviewer #1

“In this article, the authors proposed a self-driven electrical triggering system called BTO-CPT/FA. The synthesized material harnesses mechanical stimuli to electrically trigger the drug release of CPT to cure bladder tumors. Besides, to improve cell-targeting abilities, they have utilized folate acid (FA) to target the folate receptors in the bladder tumor membranes selectively. The authors demonstrated the electrical triggering system through various biological experiments. Based on their achievements, they have introduced an advancement of cell-targeted nanomedicines to treat bladder cancer. However, materials synthesis and operation remain questionable. Thus, this reviewer thinks that this article should be reconsidered after the major revision for the potential publication in Nature Communications. Below are the questions and comments that need to be addressed in the revision process.”

Re: We sincerely appreciate your esteemed recognition and expert guidance. We have carefully implemented all suggestions and thoroughly addressed each query. A detailed, point-by-point response is provided below.

*“**Comment 1:** The authors showed that the CPT can be coated through the Mitsunobu reaction in the synthesized BTO particles, but this reviewer suggests the authors explain more about the fundamentals of chemical interactions between BTO and CPT. How can they form a stable nanostructure?”*

Re: We sincerely appreciate your insightful evaluation of our work.

From a mechanistic perspective, the Mitsunobu reaction employed in this study follows the classical bimolecular nucleophilic substitution (SN2) pathway, which can be systematically deconstructed into three sequential stages: (1) Triphenylphosphine (PPh₃) undergoes ligand exchange with diethyl azodicarboxylate (DEAD), generating an activated phosphorus-nitrogen intermediate along with a nitrogen-centered anion; (2) The nitrogen-centered anion facilitates the deprotonation of the carboxylic acid via proton transfer, yielding a highly reactive carboxylate species; (3) Alcohol compounds

subsequently undergo a stereospecific SN₂ coupling reaction with the carboxylate, resulting in the formation of a C–O bond with Walden inversion at the α -carbon of the hydroxyl group. Notably, as demonstrated by Shibasaki et al. (*Org. Lett.* 2004, 6, 397)¹, this reaction platform is also effective for hydroxyl-hydroxyl condensation, enabling ether bond formation. This further supports the theoretical feasibility of hydroxycamptothecin (CPT) stabilization on BTO interfaces via the Mitsunobu reaction.

From a surface chemical engineering standpoint, the hydrogen peroxide pretreatment strategy employed in this study enables precise hydroxyl functionalization of BTO nanoparticles. This treatment induces the formation of a dense hydroxylation layer on the BTO surface, establishing an optimal reactive interface for the conjugation of CPT. As detailed in the Supporting Information section “Synthesis of the BTO-CPT/FA,” the reaction was conducted using hydroxylated BTO, CPT, PPh₃, and DEAD as reactants, with acetone as the reaction solvent at 0°C under anaerobic conditions.

To verify the chemical conjugation of CPT onto the BTO surface, ζ -potential measurements of BTO nanoparticles and CPT in acetone were performed. Following the reaction, the precipitates were thoroughly washed with deionized water, and the absorbance of CPT was analyzed by UV-vis spectroscopy before and after washing. As shown in **Figure. S2a**, the ζ -potential of BTO nanoparticles and CPT in acetone was close to zero, indicating negligible electrostatic interactions between them. Furthermore, after two washing cycles, the absorbance at $\lambda = 384$ nm (corresponding to CPT) decreased significantly (**Figure. S2b**), suggesting that CPT was initially adsorbed onto BTO due to its large specific surface area. However, after multiple washes, the absorbance at $\lambda = 384$ nm remained relatively stable (**Figure. S2c**), with CPT retention exceeding 85% (**Figure. S2d**). These findings collectively confirm that CPT was covalently bound to the BTO surface via chemical coupling rather than nonspecific adsorption. For detailed information, please refer to page 9, line 13 of the manuscript.

Figure. S2 Proof that CPT binds to BTO by chemical coupling. a) ζ -potential of BTO nanoparticles and CPT in acetone ($n = 3$), **b)** ultraviolet-visible (UV-Vis) absorption spectra of the precipitate before and after washed, **c)** the absorbance value

($\lambda = 384$ nm) of the precipitate after washed several times ($n = 3$), **d**) the residue percentage of CPT after washed ($n = 3$).

“Comment 2: In Figure. 2i, CPT has a noticeable peak at 384 nm, but there are only small peaks at the same wavelength for both BTO-CPT/FA and BTO-CPT. This reviewer doubts that this significant gap is attributed to the weak chemical interactions between CPT and BTO. More specifically, how can you make sure that the CPTs form chemically stable interactions with BTO, not just physically adsorbed in the surface such as residues?”

Re: We sincerely appreciate the reviewer's insightful comments. The results presented in Figure. 2i are qualitative experimental data, primarily intended to provide a clearer visualization of the absorption peaks for each group. To further address the reviewer's concerns, we performed UV-visible spectroscopic measurements on all relevant groups, including free CPT, BTO-CPT nanoparticles, and BTO-CPT/FA nanoparticles, with a particular emphasis on their absorption peak at 384 nm (**Figure. 2i**).

Notably, when CPT is conjugated to the BTO surface, its electronic environment is altered, which may lead to variations in absorption characteristics. To confirm that CPT is chemically conjugated to BTO rather than merely physically adsorbed, we performed ζ -potential measurements and UV-vis spectroscopy analyses before and after multiple washing cycles. As shown in **Figure. S 2c**, the ζ -potential of BTO nanoparticles and CPT in acetone was close to zero, indicating minimal electrostatic interactions. Furthermore, the UV-vis absorbance at 384 nm significantly decreased after two washes, suggesting the removal of physically adsorbed CPT. However, after additional washing, the absorbance remained relatively stable, with CPT retention exceeding 85%, thereby confirming that a substantial fraction of CPT was chemically bound to the BTO surface rather than being loosely adsorbed. These findings collectively support the formation of stable chemical interactions between CPT and BTO, validating the robustness of our conjugation strategy.

Figure. 2i Ultraviolet-visible (UV-Vis) absorption spectra of BTO-CPT/FA nanoplateform.

“Comment 3: The meaning of the gray bars in Figure. 2j is difficult to understand, so their expression should be modified.”

Re: We sincerely appreciate your insightful feedback. To enhance clarity, the gray bars in Figure. 2j have been modified to dashed lines to more clearly indicate the corresponding peaks. Additionally, the revised **Figure. 2j** as shown below:

Figure. 2j. Fourier Transform Infrared (FT-IR) spectra of BTO-CPT/FA nanoplateform.

“Comment 4: Figure. 3a and b show electrical measurement settings and the results, but it is hard to understand how the electrical outputs were measured. The authors need to provide more detailed explanations for these measurement methods and results.”

Re: We sincerely appreciate the reviewer's thoughtful evaluation and constructive feedback.

1) Composition of the Electrical Signal Measurement Device: The device is composed of a piezoelectric module, which includes a double-layer acrylic board, gold

leaf, copper wire, BTO-CPT/FA nanoparticles, and super glue. Additionally, a linear reciprocating motion meter and an electrical signal detection workstation are integrated into the setup.

2) Operating Principle: The BTO-CPT/FA nanoparticles are uniformly distributed in the central region of the gold leaf, with the copper wire attached to its outer edge. The acrylic board is then securely bonded using super glue. The fully assembled module is subsequently connected to the electrical signal detection workstation.

3) Electrical Signal Detection: The piezoelectric module is positioned on the linear reciprocating motion meter, where it undergoes a controlled "push-rebound" motion. Over a 300-second period, voltage signals from the control group, BTO group, BTO-CPT group, and BTO-CPT/FA group are recorded using the electrical signal detection workstation, allowing for precise monitoring of the electrical outputs during the motion cycle.

4) Device Visualization: A physical photograph of the device is provided in **Figure. S3** Additionally, the description in Figure. 3a of the original manuscript has been revised for improved clarity, and the updated Figure. has been uploaded.

Additional discussion has been incorporated into the manuscript on **page 10**, second paragraph.

Figure. S3 Electrical signal measurement device. a) The picture of the electrical signal measurement device, **b)** The composition of the electrical signal measurement device: piezoelectric module (①), linear reciprocating motion meter (②), and electrical signal detection workstation (③).

“Comment 5: In Figure. 3k and 3l, the control samples show more decay in absorbance

compared to the BTO-CPT/FA groups, but Figure. 3m shows the opposite trend. This reviewer doubts if there are mistakes in the Figure. captions.”

Re: Thank you for your careful review and for highlighting the layout errors in the manuscript. To ensure accuracy, we have swapped the positions of **Figure 3k** and **3l** in the manuscript so that they correctly align with the figure descriptions. Please refer to **Figure. 3k, 3l**.

“Comment 6: In this article, the authors tried to demonstrate that the CPTs can be released based on the piezoelectricity of the BTO nanoparticles. However, the release of CPTs is more likely attributed to the ultrasound irradiation. The authors need to provide more demonstrations to prove the electrically triggered drug release.”

Re: We sincerely appreciate your insightful feedback. To accurately evaluate electrically triggered drug release, we designed additional experiments to eliminate non-electrical factors, such as ultrasound effects, from influencing the results.

1) Nanocarrier Selection: We selected both non-polarized and polarized BTO nanoparticles as drug (CPT) carriers. The non-polarized BTO nanoparticles exhibit negligible or no piezoelectric effect under external pressure, which allows us to exclude any potential drug release due to non-electrical factors like ultrasound. In contrast, the polarized BTO nanoparticles have significant piezoelectric properties, ensuring that any observed drug release is primarily driven by electrical stimulation. As shown in **Figure. S4a**, the cumulative CPT release in the polarized BTO group was significantly higher than in the non-polarized BTO-CPT group over a 32-hour period under ultrasound irradiation. This clearly demonstrates that the electrical stimulation generated by the polarized BTO nanoparticles significantly enhances drug release.

2) Verification of BTO Structure: To verify the structure of both non-polarized and polarized BTO nanoparticles, we conducted X-ray diffraction (XRD) analysis. As seen in **Figure. S4b**, the polarized BTO nanoparticles exhibited a distinct split peak between 44° and 46° , confirming successful polarization. In contrast, the non-polarized BTO nanoparticles did not show this split peak, further reinforcing the difference in piezoelectric properties between the two types of BTO nanoparticles. For further details, please refer to the first and second paragraphs on page 14 of the manuscript.

Figure. S4 Evidence for drug release based on electrical triggering. **a)** The cumulative release amount of CPT from BTO-CPT/FA before and after polarization of BTO under ultrasound ($n = 3$). $***p < 0.001$. **b)** X-ray diffraction (XRD) patterns of BTO-CPT/FA before and after polarization.

“Comment 7: In Figure. 3r, the authors tried to demonstrate that the CPTs were stably captured in BTO nanoparticles under PBS buffer solution for a few days. However, this reviewer thinks that the diameter and PDI values of the composite nanoparticles cannot prove it. This reviewer suggests the authors need to measure zeta potential changes for 5 days under PBS solution.”

Re: We greatly appreciate the reviewer's valuable feedback. In response to the comment, we would like to clarify that the BTO-CPT/FA nano-system in our study is designed for bladder perfusion, and therefore, the stability of the BTO-CPT/FA nanoparticles was investigated in artificially simulated urine, rather than in PBS, as shown in Figure. 3r of the original manuscript.

Furthermore, to address the reviewer's suggestion, we measured the ζ -potential values of BTO-CPT/FA nanoparticles over a five-day period. As shown in **Figure. S5**, the ζ -potential values remained stable throughout this time, indicating that the BTO-CPT/FA nanoparticles are stable in artificial urine. Please refer to the third paragraph on page 14 of the manuscript.

Figure. S5 The ξ -potential of BTO-CPT/FA within five days in artificial urine.

“Comment 8: In this article, ultrasound was employed to simulate the intravesical pressure. However, due to their significant differences in intrinsic physical properties between ultrasound and bladder movements, this reviewer does not agree that the ultrasound can be utilized to simulate pressure generated through the bladder movements.”

Re: We appreciate the reviewer’s thoughtful comment. We recognize that there are intrinsic differences between ultrasound and the pressure generated by bladder movements. However, we would like to clarify the rationale behind using ultrasound to simulate bladder intravesical pressure in our study.

1) Bladder Intravesical Pressure in Bladder Cancer Models: As shown in our manuscript (**Figure. 8a-e**), measurements of the bladder intravesical pressure in orthotopic bladder cancer mouse models reveal that the pressure in these models is significantly higher (50-60 cm water column) compared to normal bladder pressure (25-30 cm water column). This higher pressure in bladder cancer models provides an effective "pressure generator" that drives the piezoelectric effect in the BTO-CPT/FA nano-system.

2) Simulating Bladder Pressure with Ultrasound: In our in vitro experiments, we used ultrasound to generate controlled pressure that mimics the pressure observed in bladder cancer models (2.942 KPa, as detailed in **Figure. 7p**). While ultrasound and bladder movements have different physical properties, the pressure generated by ultrasound has similarities with the intravesical pressure generated during bladder relaxation, and we give a rational analysis of the choice of ultrasound to simulate

intravesical pressure in vitro: 1) Acoustic radiation pressure: ultrasound generates pressure due to momentum transfer during propagation, and in vitro experiments, ultrasound transfers energy to the culture medium or other liquid-phase environments, and the appropriate ultrasound power is adjusted to correspond to the intravesical pressure (*Current medical imaging reviews* **7**, 328-339 (2011))²; 2) Acoustic Streaming: ultrasound induces the fluid flow of the culture medium to generate shear force, simulating the effect of urine flow on the bladder wall (*Annu Rev Biomed Eng* **6**, 229-248 (2004))³. In addition, The piezoelectric voltage of ZnO NR was calculated to be less than 0.5 V in the case of a vertically applied pressure of 50 MPa from Hoang et al (*Chem Eng J*, **435**, 135039 (2022))⁴, it is on the basis of such data that ZnO NR is effective in treating tumor through piezoelectric effect under ultrasound, while the ability of ultrasound to induce a significant piezoelectric potential (0.62 V) in the BTO-CPT/FA system under these conditions supports its use as an experimental analog for bladder intravesical pressure.

3) Experimental Advantages: Using ultrasound to simulate bladder intravesical pressure offers several practical advantages. It provides a convenient and efficient method for controlling and replicating the mechanical forces in a laboratory setting, which is important for conducting reproducible experiments. Moreover, the use of ultrasound allows for precise control over the applied pressure, ensuring that the piezoelectric effects are consistently induced.

We believe that, despite the inherent differences between ultrasound and bladder movements, the use of ultrasound in our study provides a valid and effective simulation of bladder pressure that drives the piezoelectric effect in the BTO-CPT/FA nano-system. Please refer to the third paragraph on page 12 of the manuscript.

“Comment 9: According to the results shown in Figures. 4f, 4h, and 4k, it seems that ultrasound irradiation plays a more effective role in inhibiting tumor activities than the other chemicals. Please provide more explanations on the results.”

Re: Thank you for your insightful comment. As shown in **Figure 4f, 4h, and 4k**, ultrasound irradiation plays a crucial role in enhancing tumor inhibition, primarily through the piezoelectric effect of BTO-CPT/FA nanoparticles. Under ultrasound stimulation, BTO-CPT/FA generates a significant amount of ROS *via* the piezoelectric catalytic effect, which disrupts the redox balance of tumor cells and induces apoptosis more effectively than conventional chemotherapeutic drugs. Additionally, ultrasound triggers the controlled release of CPT, further amplifying the therapeutic effect.

To further clarify the role of ultrasound, a quantitative ROS analysis was performed using a redox probe method. The results demonstrated a substantial increase in

fluorescence intensity in both the BTO+US and BTO-CPT/FA+US groups, indicating robust ROS production. In contrast, negligible ROS signals were detected in the CPT and BTO-CPT/FA groups without ultrasound, confirming that ultrasound activation is essential for triggering the piezoelectric catalytic effect.

These findings align with previous studies, such as He et al. (Biomaterials, 2022, 290: 121816)⁵, which also highlight the superior pro-apoptotic ability of piezoelectric catalysis-driven ROS generation compared to traditional chemotherapeutic agents (CPT). Overall, the data validate the specificity and spatiotemporal control of ultrasound-triggered ROS generation, underscoring its pivotal role in enhancing tumor inhibition. The details can be found in the first paragraph on page 17 of the manuscript.

“Comment 10: In Figures 6b and 6c, the authors described that the BTO-Cy5.5/FA shows better targeting abilities than the BTO-Cy5.5, because the decline in fluorescence intensity was slower in the FA coated groups. However, Figure. 6c shows no noticeable differences between the two groups.”

Re: We appreciate your insightful analysis. Due to the extended 36-hour duration of the in vivo imaging experiments, the differences between the data were not clearly discernible in **Figure. 6c** of the original manuscript. To address this, we enlarged the curves corresponding to the 8~36 h time range and performed a significance analysis (**Figure. 6c**).

Figure. 6c Quantitative analysis of Cy5.5 fluorescence intensity in bladder regions at different time points (n = 3), the inset picture represents the Cy5.5 fluorescence intensity in time range of 8 h~36 h. ** $p < 0.01$, *** $p < 0.001$.

“Comment 11: Captions in Figure. 6i are too small to be identified.”

Re: We appreciate your thorough attention to detail. We have enlarged the captions in **Figure. 6i** of the original manuscript. Please refer to the revised Figure. 6i for

clarification.

“Comment 12: The p-value expression shown in Figure. 9c is not only aside from the commonly used format but also difficult to understand. Please revise it for clarity.”

Re: We value your professional perspective. We have revised the expression of the p-values in **Figure. 9c** of the original manuscript. Please refer to the updated **Figure. 9c**.

Figure. 9c Bioluminescence curves of orthotopic bladder cancer mice after treatment with various groups (n = 5). * $p < 0.05$, *** $p < 0.001$, **** $p < 0.0001$.

“Comment 13: All Figure. expressions and manuscript descriptions for Figure. 10 are very difficult for potential readers to understand. The Figure. captions are too small to be easily readable, and the manuscript lacks sufficient explanation.”

Re: We appreciate your constructive input. To improve the readability of the data, we have adjusted the fonts of the Figure. captions in **Figure. 10** and provided a detailed description in the "Mechanisms Unveiling the Anti-Tumor Potential of the Self-Driven Electrical Triggering System" section of the original manuscript on Page 34, last paragraph.

Reviewer #2

“The manuscript entitled “Self-driven Electrical Triggering System Activates Tunnel Nanotube Highways to Enhance Drug Delivery in Bladder Cancer Therapy” by Liu et al. presents a novel approach for bladder cancer treatment through a self-driven electrical triggering system (BTO-CPT/FA) that harnesses intravesical pressure to generate electricity. This system significantly enhances drug absorption by facilitating

the transport of hydroxycamptothecin through tunneling nanotubes, thereby creating a rapid and efficient drug delivery route to tumor cells. Furthermore, the generated voltage promotes the production of reactive oxygen species (ROS), which contributes to the destruction of bladder cancer cells. The experimental data, including both in vitro and in vivo characterizations, support the conclusion drawn by the authors and confirm the functionality. However, despite the promising results, the overall quality of the manuscript—both in terms of the Figures and the writing—does not yet meet the high standards of Nature Communications. Several major issues should be addressed before further consideration.”

Re: Thank you for your valuable feedback and thoughtful assessment of our manuscript. In response to your concerns, we have taken the following steps to address the issues raised:

1) For Figures: We have enhanced the clarity, visual quality, and resolution of the Figures to meet the high standards expected by *Nature Communications*. Additionally, we have updated the Figure legends to provide more detailed and comprehensive explanations where necessary.

2) For writing Quality: We have revised the manuscript to improve its flow and readability, ensuring that our descriptions of experimental methods, results, and conclusions are more concise and coherent. This revision also includes refining the language to eliminate any ambiguities or redundancies. Please refer to the highlights of the revisions section for more details.

3) Additional Clarifications: In response to your feedback, we have added further explanations on specific aspects of the experimental design and results to improve the overall comprehensibility and to clarify any points of confusion.

We believe these revisions will enhance the quality of the manuscript and align it with the expectations of *Nature Communications*. Thank you once again for your time and consideration.

“Comment 1: The discussion of FT-IR spectrum in the manuscript is overly simplistic and lacks sufficient detail, making it difficult to interpret the data in relation to the labels provided in Figure. 2J. As a result, the current analysis does not allow for any meaningful conclusions regarding surface modifications based on the FT-IR results.”

Re: We appreciate your insightful feedback. To address your concerns regarding the FT-IR spectrum analysis, specifically, we have clarified the surface modifications and their corresponding spectral features. Additionally, the revised **Figure. 2j** as shown below. The specific details can be found in the manuscript on page 9, second paragraph.

Figure. 2j. Fourier Transform Infrared (FT-IR) spectra of BTO-CPT/FA nanoplateform.

“Comment 2: The Figure. legends require a thorough review and revision for accuracy and clarity. Specifically, the legends for Figures 3K and 3L are mistakenly swapped, and the legend for Figure. 3O contains several inaccuracies. Furthermore, Figure. 4A is missing a scale bar, and Figures 4B and 11B also lack scale bars. Additionally, the scale bar in Figure. 9K is clearly incorrect and needs to be adjusted for proper measurement representation. These issues should be addressed to ensure consistency and accuracy across all Figures and legends.”

Re: We sincerely appreciate the reviewer’s meticulous attention to detail. In response to the raised concerns, we have made the following modifications:

We have corrected the layout of **Figure. 3K** and **3L** in the original manuscript to align with the corresponding Figure. legends. Additionally, we have revised the description of the Figure. legend for **Figure. 3o** to ensure accuracy. These modifications have been highlighted in the revised manuscript.

Regarding **Figure. 4A**, it is a schematic representation of cellular uptake generated using 3D Max software. As it is not an experimental microscopic image, a scale bar is not applicable. And, we have labeled the scale bar values appropriately in the legend for **Figure. 4B** to ensure clarity.

We have carefully verified **Figure. 11B** and confirm that it depicts the surgical procedure involving the inoculation of Vx2 tumor cells into the rabbit bladder wall. A scale bar has now been inserted into **Figure. 11B**.

After thorough review of the experimental data, we have adjusted the scale bar (100 μm , top; 10 μm , bottom) in Figure. 9K to ensure proper measurement representation. The corrected version is included in the revised manuscript. Please refer to **Figure. 3, 4, and 11** for detailed information.

“Comment 3: In Figure. 3L, is the control group composed solely of deionized water? Could ultrasound also contribute to the degradation of malachite green? This raises concerns about the potential influence of dissolved oxygen in the deionized water on the observed results. Would using deoxygenated water be a more effective way to eliminate this potential interference?”

Re: We sincerely appreciate the reviewer’s insightful observations. It is well-established that ultrasound can induce cavitation effects, thermal effects, and mechanical vibrations, all of which may contribute to molecular degradation. Furthermore, malachite green contains conjugated double bonds that are particularly susceptible to oxidative degradation. Thus, prolonged ultrasound exposure can lead to some degree of malachite green degradation even in the control group (**Figure. 3L**). However, this does not compromise the validity of our experimental results, as both the control and BTO-CPT/FA groups were subjected to identical ultrasound conditions, ensuring that any observed differences in degradation can be attributed to the presence of BTO-CPT/FA.

To rigorously address the potential influence of dissolved oxygen, we conducted additional experiments using deoxygenated deionized water. As shown in **Figure. S6a, 6b**, in the absence of dissolved oxygen, malachite green degradation was significantly reduced in the control group under ultrasound. Similarly, in the BTO-CPT/FA group, the degradation rate was markedly slower under deoxygenated conditions compared to normoxia. These findings strongly suggest that dissolved oxygen plays a critical role in ultrasound-induced malachite green degradation. The underlying mechanisms are likely as follows:

a) The probability of ultrasound-induced oxidative degradation is reduced in deoxygenated water.

b) The generation of reactive oxygen species (ROS), particularly superoxide anion radicals (**Figure. 3J**, original manuscript), from BTO under ultrasound is significantly suppressed in the absence of dissolved oxygen, thereby slowing malachite green degradation.

Furthermore, to further delineate the role of oxygen, we compared malachite green degradation under normoxic and hypoxic conditions in both the control and BTO-CPT/FA groups. As illustrated in **Figure. S6c, 6d**, the degradation rate under hypoxia was substantially lower than under normoxia in both cases, further confirming the impact of oxygen availability on the degradation process.

In conclusion, while the effect of ultrasound does contribute to malachite green degradation, our experimental design ensures that its effects are systematically

controlled. The additional experiments using deoxygenated water provide compelling evidence that dissolved oxygen plays a pivotal role in this process, reinforcing the robustness of our findings. Specifically, refer to the second paragraph on page 12 and the first paragraph on page 13 of the manuscript.

Figure. S6 Effect of dissolved oxygen in deionized water on the degradation of malachite green. **a)** UV-vis absorption curve of malachite green after treated by control group under ultrasound in deionized water for oxygen removal, **b)** UV-vis absorption curve of malachite green after treated with BTO-CPT/FA group under ultrasound in deionized water for oxygen removal, **c)** Degradation of malachite green after treated by control group under ultrasound before and after oxygen removal (n = 3), **d)** Degradation of malachite green after treated with BTO-CPT/FA under ultrasound before and after oxygen removal (n = 3). *** $p < 0.001$, **** $p < 0.0001$.

“Comment 4: The confocal images in the manuscript require optimization to improve clarity and consistency. For instance, the cell concentration within the field of view in Figure. 4E appears uneven, which may lead to inconsistent interpretation, while the focus in Figure. 4G is excessively blurry, hindering the clarity of the image and the accuracy of the analysis. Moreover, the DCF fluorescence signal in Figure. 9H is nearly undetectable.”

Re: We sincerely appreciate the reviewer’s constructive feedback regarding the optimization of confocal images. To address these concerns, we carefully re-evaluated **Figure. 4e, 4f,** and **9h** and performed additional experiments to enhance the clarity and consistency of the imaging data.

Specifically, we acknowledge that variations in cell concentration within the field of view in **Figure. 4E** could potentially impact the consistency of interpretation. To mitigate this, we repeated the live-dead cell staining assay under controlled conditions to ensure uniform cell distribution. The updated results are presented in revised **Figure. 4E**, which confirm the consistency of our original findings, as described in the section "Evaluation of the Antitumor Effects of the Self-Driven Electrical Triggering System In Vitro" in the manuscript.

Calcein AM PI

Figure. 4E Evaluation of the antitumor effects of the BTO-CPT/FA *in vitro*. Confocal images of live-dead cell staining for MB49 cells after treated with different groups, where green and red colors represent Calcein AM and PI fluorescence, respectively, scale bar = 100 μm .

Additionally, regarding the blurriness observed in **Figure. 4G**, we have carefully optimized the imaging conditions, including adjusting the focal plane and acquisition settings, to improve image sharpness and resolution, the updated results are presented in revised **Figure. 4G**.

Hoechst 33342 DCF

Figure. 4G Evaluation of the antitumor effects of the BTO-CPT/FA *in vitro*. The DCF images of MB49 cells after treated with different groups, and the blue and green colors representing Hoechst 33342 (nuclei) and DCF (ROS levels) fluorescence, respectively, scale bar = 25 μm .

As for **Figure. 9h**, we acknowledge the weak DCF fluorescence signal. To enhance signal detection, we re-performed the ROS assay using the DCFH-DA probe under optimized experimental conditions, ensuring improved fluorescence intensity and contrast. The updated results are presented in revised **Figure. 9h**, further corroborating the trends described in the original manuscript.

Figure. 9h CLSM images of DCF of the sections of bladder cancer tissues, and the blue and green colors representing DAPI and DCF fluorescence, respectively, scale bar = 50 μm .

“Comment 5: Several issues regarding the dosage remain unclear. The manuscript does not detail any optimization process for the CPT and FA concentrations in the BTO-CPT/FA nanoplatfrom. Additionally, the loading amount of Cy5.5 on BTO is not provided.”

Re: We appreciate the reviewer’s insightful comments regarding the dosage optimization and loading amount in our study. To clarify, a detailed description of the synthesis procedure for BTO-CPT/FA, including the dosages of CPT, FA-PEG₂₀₀₀-COOH, and BTO, is provided in the “**Synthesis of BTO-CPT/FA**” section of the Supporting Information in the original manuscript. Furthermore, to illustrate the optimization process of CPT loading, we conducted a series of experiments with varying CPT-to-BTO ratios, as summarized in **Supplementary Table 1**.

Supplementary Table 1 Optimization process in chemical coupling of CPT on BTO.

Number	CPT/mmol	BTO/mg	CPT loading rate/Wt%
①	0.01	80	5.5
②	0.01	100	6.2
③	0.01	120	6.5

Considering both economic feasibility and loading efficiency, we selected the synthesis condition labeled as ② (0.01 mmol CPT and 100 mg BTO) for the preparation of BTO-CPT/FA nanomaterials. The thermogravimetric analysis (TGA) results, presented in **Figure. S7**, further confirm the successful incorporation of CPT. The mass fraction of CPT and FA on BTO-CPT/FA was 6.2%, 5.7%, respectively.

Figure. S7 The curve of thermogravimetric analysis (TGA).

Additionally, regarding the loading amount of Cy5.5, we established a standard calibration curve using UV-vis absorption at $\lambda = 650 \text{ nm}$ (**Figure. S8**). Based on this, we calculated the Cy5.5 loading rate in BTO-Cy5.5 to be 8.9%. In addition, more in-depth discussion is now included in the paper on page 8, **Figure S7**, **Figure S8** and **Supplementary Table 1**.

Figure. S8 Standard curve based on Cy5.5.

“Comment 6: Previous reports, such as those in Anal. Chem. 2019, 91, 6996-7000, demonstrate that Cy5.5 is taken up by tumor cells when administered alone. However, in this manuscript (Figure. 4B), this uptake is nearly absent. This discrepancy needs to be addressed and explained in more detail.”

Re: We appreciate the reviewer’s insightful observation and the reference to relevant

literature. The fluorescence intensity of each group in Figure. 4B of the original manuscript represents the relative fluorescence intensity, rather than an absolute measure of Cy5.5 uptake. To ensure consistency and accurate comparison, we applied identical laser intensity and exposure time during confocal imaging. As a result, analyzing the fluorescence of a single group in isolation may not provide meaningful reference data.

Regarding the difference in Cy5.5 uptake compared to previous reports, several key factors contribute to this observation:

1) Differences in Cy5.5 Formulation: The fluorescent dye used in **Figure. 4B** of our study is Cy5.5-COOH, chosen for its ability to chemically couple to BTO via its carboxyl (-COOH) groups. However, this modification significantly alters the physicochemical properties of Cy5.5: Cy5.5-COOH is electronegative and hydrophilic, whereas unmodified Cy5.5 is electroneutral and hydrophobic in aqueous environments. Given that MB49 cell membranes carry a negative charge, the passive transport of small molecules preferentially facilitates the uptake of neutral Cy5.5, rather than the negatively charged Cy5.5-COOH. This fundamental difference likely accounts for the lower fluorescence intensity observed in our study compared to literature reports using unmodified Cy5.5.

2) Cell-Type-Specific Differences: The uptake of Cy5.5 derivatives can vary depending on cell type and experimental conditions. The previous literature (*Anal. Chem.* 2019, 91, 6996-7000) cited by reviewer investigated Cy5.5 uptake in different tumor models, and variations in membrane composition, charge, and transport mechanisms could contribute to discrepancies in uptake efficiency.

3) Instrument and Parameter Variability: The relative fluorescence intensity of Cy5.5 is influenced by instrument settings, imaging parameters, and confocal microscopy sensitivity. Since the fluorescence intensity is not an absolute value, direct comparison between studies conducted on different platforms must account for variations in imaging conditions.

We appreciate the reviewer's concerns and hope this explanation adequately addresses the observed discrepancy.

Figure. 4b Confocal images of MB49 cells treated with different groups, where blue, green and red colors represent DAPI, Dil and Cy5.5 fluorescence, respectively.

*“Comment 7: In the cell viability assays, is there a statistically significant difference between CPT and BTO-CPT/FA with a p -value of $****p < 0.0001$? Additionally, the difference between the 100 $\mu\text{g}/\text{mL}$ and 200 $\mu\text{g}/\text{mL}$ concentrations appears minimal. Does this suggest that doubling the dose does not result in a corresponding increase in cell death?”*

Re: We sincerely appreciate the reviewer’s insightful comments.

Regarding the statistical significance in the cell viability assays, the label $****p < 0.0001$ in **Figure. 4i** of the original manuscript specifically denotes the comparison between the BTO-CPT/FA+US group and both the CPT group and the BTO-CPT/FA group, rather than between the CPT and BTO-CPT/FA groups. The modified representation about significant difference analysis has been inserted into **Figure. 4i** in manuscript to ensure precise interpretation.

Regarding the minimal difference observed between the 100 $\mu\text{g}/\text{mL}$ and 200 $\mu\text{g}/\text{mL}$ concentrations, we attribute this to the following factors:

1) Saturation Effect: After 36 hours of incubation, the cellular uptake of nanomedicine at 100 $\mu\text{g}/\text{mL}$ approaches saturation. As a result, further increasing the concentration to 200 $\mu\text{g}/\text{mL}$ does not proportionally enhance intracellular accumulation, leading to negligible changes in cytotoxicity.

Maximum Effective Concentration: The 100 $\mu\text{g}/\text{mL}$ concentration is close to or exceeds the nanomedicine’s maximum effective dose. Therefore, a further increase in concentration likely remains within the plateau phase of the dose-response curve, with no substantial gain in cytotoxic effects.

2) Cellular Efflux Mechanisms: At 200 $\mu\text{g}/\text{mL}$, MB49 cells may activate exocytosis pathways, limiting intracellular drug retention and thereby mitigating any additional cytotoxic impact. Moreover, similar trends have been reported in the literature. Chen et al. (*Biomaterials*, 2015, 53: 699-708) ⁶ demonstrated that doubling the dose of a CPT-based nanodrug under ultrasound did not lead to a proportional

increase in tumor cell death. This supports the notion that surpassing a certain concentration threshold does not necessarily enhance therapeutic efficacy.

Collectively, these findings provide a plausible explanation for the observed lack of significant cytotoxicity increase when the nanomedicine concentration was raised from 100 $\mu\text{g/mL}$ to 200 $\mu\text{g/mL}$.

“Comment 8: The simulation results in Figures 6H-J demonstrate the binding between FA receptors and FA. However, it remains unclear whether these findings are directly applicable when FA is modified onto BTO-CPT in the BTO-CPT/FA nanopatform. This distinction should be clarified to ensure the relevance of the simulations to the experimental context.”

Re: We sincerely appreciate the reviewer’s insightful comments. Regarding the issues raised by the reviewer, we believe they can be attributed to the following factors:

1) Structural Basis: As detailed in the "Synthesis of BTO-CPT/FA" section of the Supporting Information, the synthesis process ensures that CPT and FA-PEG₂₀₀₀-COOH are sequentially conjugated onto BTO nanoparticles. Specifically, CPT is first chemically coupled to BTO, followed by FA-PEG₂₀₀₀-COOH modification. This design ensures that FA-PEG₂₀₀₀-COOH is oriented outward, facilitating its preferential binding to folate receptors on tumor cell membranes.

2) Spatial Configuration and Targeting Mechanism: The long-chain PEG₂₀₀₀ structure plays a crucial role in the spatial orientation of FA. Since FA is conjugated at the terminal end of PEG₂₀₀₀, after modification onto the BTO surface, the flexible PEG₂₀₀₀ linker allows FA to extend into the surrounding medium. This structural arrangement provides a favorable conformation for FA-receptor interactions in both in vitro and in vivo environments. As a result, when BTO-CPT/FA encounters tumor cells, the targeting event precedes drug release: FA first binds to folate receptors, enhancing nanodrug accumulation within tumor cells, followed by CPT release, which induces DNA damage.

To explicitly connect the simulation findings to the experimental context, we have incorporated a detailed explanation into the third, fourth, and fifth paragraphs on page 22 of the manuscript. We hope this revision effectively addresses the reviewer’s concerns and further strengthens the connection between the computational and experimental findings.

“Comment 9: The manuscript states that "an intravesical pressure-driven system in the bladder generates electricity," implying that bladder perfusion pressure is directly linked to therapeutic outcomes. However, this relationship is not clearly demonstrated

in the in vivo experiments. How can it be ensured that perfusion pressures remain consistent across different groups when administering various drug formulations? Additionally, to what extent would variations in perfusion pressure affect the treatment efficacy?"

Re: We sincerely appreciate the reviewer's insightful comments. In response to the reviewer's concerns, we provide the following clarifications:

1) Standardization of Intravesical Pressure Across Groups: Before intravesical perfusion, water intake was ceased in all mice to empty the bladder, ensuring that the baseline intravesical pressure was normalized across groups. This step minimizes variations and allows for smooth and consistent perfusion.

2) Equal Perfusion Volume Across Experimental Groups: During in vivo treatments, all mice—including both control and experimental groups—were infused with an equal volume (50 μ L) of either deionized water or nanomedicine using a micro-bladder perfusion device. This setup ensures that each group experiences the same initial perfusion pressure, eliminating variability caused by differing infusion volumes. Additionally, as shown in **Figure. S9**, perfusing volumes greater than 50 μ L resulted in overflow, confirming that 50 μ L is the maximum optimal volume that the bladder can accommodate without inducing pressure inconsistencies.

Figure. S9 The pictures of bladder perfusion with different perfusion volume.

3) Minimal Influence of Perfusion Pressure on Therapeutic Outcomes: While intravesical pressure plays a role in the self-driven electrical triggering system, it is crucial to distinguish between perfusion pressure (occurring during administration) and physiological bladder pressure fluctuations (which drive the piezoelectric response). Perfusion itself is completed within 10~15 seconds, making its impact on the overall therapeutic efficacy negligible. The primary driving force behind the system is the bladder's natural "contraction-relaxation" cycle, which continuously modulates the piezoelectric effect and sustains the electrical triggering process.

4) Sustained Electrical Triggering and Drug Retention: The BTO-CPT/FA nanoplatform exhibits prolonged bladder retention (up to 36 hours, as shown in **Figure. 6b, c** of the original manuscript), ensuring that electrical triggering persists beyond the short perfusion period. Furthermore, the administration schedule (one dose every 36 hours, **Figure. 9a**) aligns with this retention time, maintaining continuous therapeutic activation throughout the treatment duration.

In conclusion, the perfusion pressure is standardized across groups by controlling pre-treatment conditions, infusion volume, and administration technique. More importantly, its transient nature (10~15 seconds) renders it insignificant in affecting treatment efficacy. Instead, the self-driven system relies on prolonged bladder contractions and relaxations, ensuring a continuous and reproducible piezoelectric therapeutic effect. Furthermore, the discussion has been extended in the manuscript on the final paragraph on page 30.

“Comment 10: The manuscript contains several minor formatting issues, including inconsistencies in citation style (e.g., sometimes only the first author is listed, while other times all authors are included). Additionally, abbreviations should be defined upon their first use to ensure clarity and consistency throughout the text.”

Re: We sincerely appreciate your meticulous review. We have carefully examined the citation style throughout the manuscript and standardized the number of authors accordingly. Additionally, to enhance clarity and consistency, we have ensured that all acronyms are properly defined upon their first appearance. These redefined abbreviations include, barium titanate (BTO), hydroxycamptothecin (CPT), folic acid (FA), high angle annular dark field (HAADF), energy dispersive x-ray (EDX), atomic force microscope (AFM), ethylenediaminetetraacetic acid disodium salt (EDTA-Na), mouse bladder cancer cells (MB49), 2,7-Dichlorodihydrofluorescein diacetate (DCFH-DA), animal in vivo imaging system (IVIS), polydimethylsiloxane (PDMS), kyoto encyclopedia of genes and genomes (KEGG). Please see the revised manuscript for details.

“Comment 11: The writing in this manuscript appears to be heavily influenced by AI editing, which has led to a somewhat unnatural and inconsistent writing style.”

Re: Thank you for your keen observation. We have carefully re-examined the manuscript and addressed issues related to language and description. To improve clarity and readability, we have standardized the writing style throughout.

Reviewer #3

“Comment: This manuscript demonstrated a self-driven electrical triggering system (BTO-CPT/FA) for bladder cancer treatment. This system could release drugs (CPT) and generate reactive oxygen species (ROS) under electrical stimuli induced by internal bladder pressure. It showed remarkable anti-tumor effect and high-speed drug delivery through tunneling nanotubes. Furthermore, the authors demonstrate the efficacy of their system in various animal models (mice, rabbits, minipigs). The concept of utilizing

internal bladder pressure to induce electricity is innovative. However, upon closer inspection, it is unclear if this system effectively addresses the limitations inherent in bladder cancer treatment. Several experimental gaps and questionable interpretations diminish the manuscript's overall strength. Consequently, the manuscript is not suitable for publication in Nature Communications. List of comments are shown as below."

Re: We sincerely appreciate the reviewer's thorough assessment and insightful feedback on our manuscript. We fully acknowledge the challenges associated with bladder cancer treatment and the importance of ensuring that our proposed system addresses these issues effectively. In response to the reviewer's comments, we provide the following detailed clarifications and modifications:

1) Effectiveness of the Self-Driven Electrical Triggering System: The main objective of our study is to develop a self-sustained, intravesical pressure-driven system that enables localized, controllable drug release and ROS generation for bladder cancer therapy. Unlike conventional intravesical drug delivery, which often suffers from rapid drug clearance and limited tissue penetration, our system utilizes piezoelectric nanogenerators to sustain CPT release while simultaneously generating ROS, thereby enhancing tumor cell apoptosis. Additionally, the integration of a tunneling nanotube-mediated transport mechanism ensures deeper tumor penetration and more efficient drug uptake (**Figure 5i, 5j, 5k** and **Supplementary Video 5, 6**). This dual action of drug release and ROS generation represents a novel and promising approach to overcoming key limitations in bladder cancer treatment.

2) Addressing Bladder Cancer Treatment Limitations: We recognize the complexities of bladder cancer therapy, including issues of poor drug retention, inadequate permeability, and high rates of tumor recurrence. Our system specifically addresses these limitations in the following ways: **a) Extended Retention Time:** The BTO-CPT/FA nanoplatform is designed to exhibit prolonged bladder residence (36 hours, as shown in **Figure 6b, 6c** of the manuscript), ensuring sustained therapeutic action. **b) Enhanced Drug Penetration:** The activation of tunneling nanotubes enables rapid drug transport, effectively overcoming the urothelial barrier and improving drug distribution within deep-seated tumor regions (**Figure 11g-11o**). **c) ROS-Driven Synergistic Therapy:** The piezoelectric-induced ROS generation further enhances the effects of chemo-immunotherapy, amplifying tumor suppression beyond the capabilities of conventional chemotherapy alone (**Figure 4d-4f** and **Figure 9h**).

3) Experimental Justification and Interpretation: In response to the reviewer's concerns about experimental gaps, we have strengthened the manuscript by clarifying the correlation between variations in intravesical pressure and electrical triggering efficiency through quantitative analysis (**Figure 7, Figure 8** and **Figure S17**).

Furthermore, we have included additional data comparing the diffusion dynamics of drugs in conventional intravesical chemotherapy and our system (**Figure Figure. 6c**). This comparison highlights the superior retention and permeability achieved with our approach. We have also refined the discussion to ensure the accurate interpretation of experimental findings and to address any potential ambiguities. Please refer to the revised sections, which are highlighted in blue in the updated manuscript.

Given the novelty and potential translational impact of this technology, we believe that our manuscript provides valuable contributions to the field of non-invasive bladder cancer therapy. We are committed to revising the manuscript to comprehensively address the reviewer's concerns and would greatly appreciate reconsideration for publication in *Nature Communications*.

“Comment 1: BTO-CPT/FA system is an internal stimuli-responsive drug and ROS releasing system. Based on this perspective, the authors should emphasize their system compared to other drug delivery systems.”

Re: We appreciate your valuable methodological insight. The self-driven electrical triggering and ROS-releasing system based on BTO-CPT/FA offers several advantages over existing drug delivery systems:

1) Enhanced Drug Permeability: Traditional drug delivery methods often struggle with poor drug penetration in tumor tissues. In contrast, the self-driven electrical trigger system (BTO-CPT/FA) developed in this study significantly enhances drug absorption by utilizing tunnel nanotubes (TNTs) as efficient channels. These nanotubes provide a rapid pathway that enables drugs to more effectively penetrate tumor cell membranes, thereby improving drug permeability and therapeutic efficacy.

2) Precise Drug Release Control: Traditional chemotherapy often suffers from uncontrolled drug release, leading to suboptimal therapeutic outcomes. The electrical trigger system in this study uses electricity generated by intravesical pressure to precisely regulate drug release. This method allows for controlled drug release via electrical stimulation, ensuring that drugs reach the optimal concentration within tumor cells while minimizing toxicity to surrounding healthy tissues.

3) Promotion of ROS Generation to Enhance Therapeutic Effects: This system not only facilitates efficient drug delivery but also generates reactive oxygen species (ROS), which further contribute to the destruction of bladder cancer cells. Unlike traditional chemotherapy, which primarily relies on the anti-cancer properties of the drug itself, this study integrates the ROS-generating mechanism of the electrical trigger system, offering a multi-faceted therapeutic strategy that enhances the anti-cancer effect.

4) Non-invasive and Efficient Treatment: Traditional treatments typically require

invasive procedures, such as surgery or injections, to effectively deliver drugs to the tumor site. In contrast, the self-driven electrical trigger system in this study utilizes the pressure within the bladder to facilitate drug delivery without additional invasive procedures, offering a safer and more efficient treatment option.

In summary, this study overcomes several limitations of traditional drug delivery systems by incorporating an innovative self-driven electrical trigger system and tunnel nanotube-based drug transport. These advancements hold great potential for revolutionizing bladder cancer treatment. Additionally, we have added the advantages of this nanoplatform in the “**Discussion**” section of the manuscript.

*“**Comment 2:** The authors mentioned that this system could form tunneling nanotubes and deliver drugs rapidly. Experimentally, they confirmed that tunneling nanotubes were formed and that drugs could be transported within the nanotubes. However, there is a lack of experiments to confirm that the delivered drug has an effect. The following experiments need to be conducted: (1) to prove the effect of tunneling nanotubes, it is necessary to compare the environment that inhibits the formation of tunneling nanotubes. (2) co-incubating cells that have treated nanoparticles with non-treated cells, to confirm if the drug is transferred to the non-treated cell, and whether the transferred drug is then effective. Furthermore, it is unclear why these tunneling nanotubes allow for high-speed delivery. We also wonder how this system works, i.e., how the tunneling nanotubes are formed by electrical stimuli, what is the specific mechanism?”*

Re: We acknowledge your constructive input. We added the Cytochalasin D (a common inhibitor for the formation of tunneling nanotubes) to cell culture dishes to investigate its effect on the formation of tunneling nanotubes. As shown in the **Figure. 5h**, there were obvious tunneling nanotubes in the control group, some of which were attached among the cells (green arrows), and others were still extending outward (orange arrows), and the cell morphology was well preserved, whereas the number of tunneling nanotubes after treated by Cytochalasin D was significantly reduced, and the cell morphology was altered (white arrows) (*Science Advances* **10**, eabj1133 (2024))⁷. Consequently, the Cytochalasin D can inhibit the formation of tunneling nanotubes.

Figure. 5h The confocal images of MB49 cells after treated with control or Cytochalasin D, scale bar = 100 μm .

Furthermore, to investigate whether BTO-CPT/FA nanoparticles transported through tunneling nanotubes induced death of surrounding cell, that is anti-tumor effects. The MB49 cells treated with BTO-CPT/FA nanoparticles were co-incubated with untreated MB49-Luc cells (with luciferase-labeled), and then detected the intensity of bioluminescence. The experiment including, i1) MB49-Luc cells, i2) MB49 cells + MB49-Luc cells, i3) Cytochalasin D + MB49-Luc cells, i4) MB49 cells (BTO-CPT/FA NPs) + MB49-Luc cells + ultrasound, i5) MB49 cells (BTO-CPT/FA NPs) + MB49-Luc cells + Cytochalasin D + ultrasound, i6) MB49-Luc cells (BTO-CPT/FA NPs) + ultrasound groups, respectively (**Figure. 5i**). As shown in **Figure. 5j**, the bioluminescence intensity was virtually unchanged, in i2) group, indicated that there was no effect on the bioluminescence of MB49-luc cells after MB49 cell addition. In other words, the addition of MB49 cells had no impact on the activity of MB49-luc cells.

However, we observed a slight reduction in cell activity after co-incubated cytochalasin D with MB49-Luc cells, which may be attributed to its capacity to inhibit cell proliferation (*The Journal of cell biology* **165**, 607 (2004))⁸. Surprisingly, there was a more noticeable decreasing tendency in bioluminescence intensity of Luc-MB49 cells after co-incubated with BTO-CPT/FA-treated MB49 cells. However, when cytochalasin D was added to this system, the death rate decreased slightly of MB49-Luc cell. These results suggested that 1) BTO-CPT/FA NPs are able to pass through tunneling nanotubes to the surrounding cells and the transported nanomedicines are able to induce cell death; 2) cytochalasin D impaired the ability of nanomedicine delivery by decreased the establishment of the network of intercellular tunneling nanotubes; and 3) cytochalasin D may also be able to reduce nanomedicine-induced cytotoxic effects by inhibiting the MB49-Luc cellular uptake (*Journal of Cell Biology* **218**, 1972-1993 (2019); *National Science Review* **10**, nwad179 (2023))^{9,10}. In addition, significant cytotoxicity was observed after co-incubation of MB49-Luc cells with BTO-CPT/FA NPs, which consistent with BTO-CPT/FA group of **Figure. 4i** in manuscript. In conclusion., these results indicated that the transported-BTO-CPT/FA nanoparticles through tunneling nanotubes can induce cell death, that is, have an anti-tumor effect.

Figure. 5i, j Effect of cytochalasin D on the transport of BTO-CPT/FA through tunneling nanotubes. i) Schematic representation of effect of cytochalasin D on the transport of BTO-CPT/FA. b) Relative bioluminescence intensity of MB49-Luc cells after treated by different groups (n = 5). * $p < 0.05$, **** $p < 0.0001$.

Alternatively, in order to have a more comprehensive insight into the transport of BTO-CPT/FA nanoparticles among MB49 cells through tunneling nanotubes, BTO-CPT/FA nanoparticles were added and co-incubated with MB49 cells, and the process was monitored by real-time imaging of live cells. Surprisingly, as shown in **Figure. 5k and Supplementary Movie. 5, 6**, the selected frames showed rapid movement of BTO-CPT/FA nanoparticles (Cy5.5, red color) along the tunneling nanotube (white arrows) from one MB49 cell to another MB49 cell (the parallel white dashed lines indicate the starting and final positions of the movement, respectively), as did the merge images.

Once the recording was finished, the transport rate of BTO-CPT/FA nanoparticles in tunneling nanotube was calculated to be $0.0172 \mu\text{m/s}$, the value that is in keeping with the transport of viral particles in fibroblasts or epithelial cells through tunneling nanotubes (*Nature reviews Molecular cell biology* **9**, 431-436 (2008), *The Journal of Immunology* **177**, 8476-8483 (2006), *The Journal of cell biology* **170**, 317-325 (2005), *Nature cell biology* **9**, 310-315 (2007))^{11, 12, 13, 14}, and in addition is close to the rate of macrophage-driven nanomedicine transport to tumor cells (*ACS nano* **13**, 1078-1096 (2019))¹⁵. In conclusion, these results suggest that tunneling nanotubes are important bridges for the high-speed transport of BTO-CPT/FA nanoparticles among MB49 cells.

Figure. 5K The confocal images of BTO-CPT/FA nanoparticles transported among MB49 cells through tunneling nanotube in a series of time intervals, where red, green and blue colors represent Cy5.5, tunneling nanotube and Hoechst 33342, respectively scale bar = $20 \mu\text{m}$.

Channel Cy5.5.mp4

Supplementary Movie. 5 Real-time imaging of BTO-CPT/FA nanopatform transport among MB49 cells through tunneling nanotubes under ultrasound. The red color represents Cy5.5 of BTO-CPT/FA nanoparticles, scale bar = $20 \mu\text{m}$.

Channel Merge.mp4

Supplementary Movie. 6 Real-time imaging of BTO-CPT/FA nanopatform transport among MB49 cells through tunneling nanotubes under ultrasound. The red color, green color and blue color represent Cy5.5 of BTO-CPT/FA nanoparticles, tunneling nanotubes and nucleus of cell, respectively. scale bar = 20 μm .

Additionally, to understand the “*how the tunneling nanotubes are formed by electrical stimulation*”. The sequencing results of the eukaryotic transcriptome of the bladder tumor tissues shown that, electrical stimulation triggered by intravesical pressure activates signaling pathways (such as, PI3K-Akt signaling pathway, GMP-PKG signaling pathway, proteoglycans in cancer, cytokine-cytokine receptor interaction, inflammatory mediator regulation of TRP channels, axon guidance, and gap junctions) associated with tunneling nanotube formation through up regulation (such as, *Mmp2*, *Rasd2*, *Wasf3*, *Prkcb*, *Prkcq*, and *Prkcz*) and down regulation of key genes (such as, *Ptk2* and *Baiap2*), enhances the efficiency of intercellular nanodrug delivery, and thus establishes a foundation for improving the outcome of tumor therapy.

Additionally, protein–protein interaction network analysis revealed the expression of key genes associated with tunneling nanotube (TNT) formation (Fig. 10j). Notably, the expression of *CD79A/CD79B* and *CDC20* highlighted a significant phenomenon in TNT formation following treatment with BTO-CPT/FA NPs, particularly aiding in the extension of the cell membrane to form the tubular structures required for TNTs. Furthermore, the increased expression of *CCL19/CCR1* and *CCL19/CCR6* in the BTO-CPT/FA group may facilitate the establishment of the TNT structure by promoting membrane protrusion and cellular pseudopod formation. These results indicate that key genes and protein expressions were significantly altered in response to electrical stimulation triggered by intravesical pressure following BTO-CPT/FA treatment. Moreover, signaling pathways closely linked to TNT formation were activated, providing mechanistic support for the high-speed transport of BTO-CPT/FA NPs through TNTs among tumor cells, and laying the groundwork for improving tumor therapy outcomes (**Fig. 10**, manuscript). Furthermore, a detailed description was inserted into the blue section on page 19, 20, 21 and 35.

“Comment 3: *The manuscript lacks critical data on the characterizations of BTO-CPT/FA nanoparticles. Specific details on: drug loading capacity, FA coating efficiency.”*

Re: Thank you for your professional advice. 1) The synthesis process of BTO-CPT/FA

and the dosages of CPT, FA-PEG₂₀₀₀-COOH and BTO has a detailed discussion in the “*Synthesis of the BTO-CPT/FA*” section of the supporting information of the manuscript. In addition, in order to state the dosage optimization of CPT in synthesis procedure, the below experiments were conducted in **Supplementary Table 1**.

Supplementary Table 1 Optimization process in chemical coupling of CPT on BTO.

Number	CPT/mmol	BTO/mg	CPT loading rate/Wt%
①	0.01	80	5.5
②	0.01	100	6.2
③	0.01	120	6.5

Based on the synthesis capability and loading rate, the synthesis condition with number “②” (0.01 mmol CPT and 100 mg BTO) was confirmed as optimal and the BTO-CPT/FA nanoplatfrom was prepared based on the ratio. The thermogravimetric analysis (TGA) results, presented in **Supplementary Fig.7**, further confirm the successful incorporation of CPT. The mass fraction of CPT and FA on BTO-CPT/FA was 6.2%, 5.7%, respectively.

Figure. S7 The curve of thermogravimetric analysis (TGA).

Additionally, regarding the loading amount of Cy5.5, we established a standard calibration curve using UV-vis absorption at $\lambda = 650$ nm (**Supplementary Fig.8**). Based on this, we calculated the Cy5.5 loading rate in BTO-Cy5.5 to be 8.9%. Additionally, supplementary discussion has been integrated into the text on the last paragraph of page 8 and the first paragraph of page 9.

Figure. S8 Standard curve based on Cy5.5.

“Comment 4: We have a few questions about in vitro experiments (Fig. 4). The live-dead images and their quantifications do not match. For example, the BTP-CPT/FA + US groups is quantitatively reported as nearly 100% dead, but the corresponding image suggests otherwise. Re-experiments or clearer justifications are required. Furthermore, in DCF (ROS levels) confocal images, the amounts of ROS seems to insufficient. Quantitative measurements of ROS levels using additional assays are recommended to corroborate the results.”

Re: Thank you for your professional perspective. Regarding the "live-dead images and their quantifications" in **Figure. 4** of the original manuscript, we defined cell death (fluorescence intensity) in the BTO-CPT/FA+US group as the positive control, and calculated the relative fluorescence intensity of cell death in the other groups based on this reference. To minimize any potential misunderstanding for reviewers and readers regarding the presentation of the data, we have re-performed the live-dead cell staining experiments and updated the results in **Figure. 4e** and **4g** of the original manuscript. For the convenience of the reviewers, we have also provided the supplementary data below.

Calcein AM PI

Figure. 4e Confocal images of live-dead cell staining for MB49 cells after treated with different groups, where green and red colors represent Calcein AM and PI fluorescence, respectively, scale bar = 100 μm .

Additionally, we acknowledge this concern and have re-performed the ROS detection experiments at the cellular level. The updated results are presented in **Figure. 4g** and **4h** of the revised manuscript. Additionally, for ease of review, we have included the supplementary data below.

Figure. 4g The DCF images of MB49 cells after treated with different groups, and the blue and green colors representing Hoechst 33342 (nuclei) and DCF (ROS levels) fluorescence, respectively, scale bar = 25 μm .

“Comment 5: The authors also developed a PDMS+BTO-CPT/FA device (Fig. 7). The performance of this device was evaluated, but the application in an animal model was not confirmed. The purpose of developing this device is unclear. Its application requires surgical intervention, potentially limiting its feasibility.”

Re: Dear reviewer, we appreciate your comments on the balloon-model design (**Fig. 7**, manuscript). The experiment was designed to simulate the activation process of piezoelectric nanosystem (BTO-CPT/FA) driven by intravesical pressure on the following scientifically valid basis:

1) Physiological authenticity: The process of “pumping air/water and expelling air” inside the balloon accurately reproduces the mechanical environment of the bladder wall. Through real-time monitoring of the output voltage (**Fig. 7**), we confirmed that the voltage signal intensity (0.5-2.5 V) generated by the BTO-CPT/FA piezoelectric layer during this process was positively correlated with the pumping frequency or flow rate, and this result also matched with that of *ex-vivo* porcine bladder tissues at different pumping flow rates (**Supplementary Fig. 17**). It was verified that the balloon model could accurately simulate the mechanical-electrical conversion characteristics of the dynamic contraction-expansion of the bladder.

2) Necessity of experimental design: The bladder is a dynamic mechanical system, and the traditional static experimental design cannot simulate its periodic

pressure changes. By programmatically controlling the frequency/flow rate of “pumping air/water and expelling air”, the first quantitative characterization of the real-time electrical generation behavior of piezoelectric nanomaterials in a similar environment of the bladder was achieved in the balloon model (**Fig. 7**). This design bridges the gap of existing studies that focused only on static electrical stimulation.

3) Mechanism validation: The results from **Fig. 7**, and in combination with the studies of others (*Chem Eng J* **435**, 135039 (2022))⁴ indicate that the balloon model successfully verified that the intensity of intravesical pressure (30-60 cm H₂O) could induce that the BTO-CPT/FA piezoelectric nanomaterials produced a desirable piezoelectric effect. This key mechanism validation provides theoretical support for the design of in vivo therapy experiments.

To investigate its potential for urinary bladder dynamics monitoring, an *ex vivo* experiment was conducted using a porcine bladder model weighing 54.5 g (*Science Advances* **6**, eaba0412 (2020), *ACS Applied Materials & Interfaces*, (2025), *Nano Energy* **119**, 109051 (2024), *Biosensors and Bioelectronics* **225**, 115060 (2023), *Advanced Materials Technologies* **4**, 1900100 (2019))^{16, 17, 18, 19, 20}. The BTO-CPT/FA-based piezoelectric nanogenerator device, additionally encapsulated with a biocompatible polydimethylsiloxane (PDMS) coating, was fabricated using spin coating at 400 rpm to assess its piezoelectric performance and durability.

As illustrated in **Figure. S17-a, b**, 60 mL of deionized (DI) water was injected into the bladder via an 18 FR/CH (5–10 cc/mL) catheter and a 70 mL capacity syringe. During the forward stroke of the syringe, bladder relaxation due to increased internal pressure resulted in a corresponding increase in piezoelectric voltage output (~ 7V). Conversely, during the backward stroke, bladder contraction led to a decrease in voltage output, as shown in **Figure. S17-c**.

Additionally, as depicted in **Figure. S17-d, e**, DI water was introduced into the bladder at a constant flow rate of 100 mL/h using a syringe pump. During the storage phase, the voltage output increased from 0 V to a peak of 2.86 V, before declining to 1.28 V during the manually induced voiding phase, as shown in **Figure. S17-f**. These findings validate the sensor's robustness and effectiveness in real-time bladder monitoring, demonstrating its potential for clinical translation in urological healthcare applications. Furthermore, supplementary discussion has been integrated into the revised manuscript on the second paragraph on page 27.

Figure. S17 Experimental setup for evaluating the porcine bladder sensing system under *ex-vivo* conditions. **a, b)** Experimental setup simulating the storage and voiding functions of the urinary bladder. **c)** Voltage output variation during the manual injection of deionized (DI) water using a 70 mL syringe. **d, e)** Experimental setup utilizing a syringe pump to maintain a constant DI water flow rate of 100 mL/h. **f)** Corresponding voltage response recorded during the injection of 60 mL of DI water.

“Comment 6: The samples were delivered via bladder perfusion instead of intravenous injection (i.v. injection). The authors should explain why this route was chosen and discuss its advantages or limitations compared to alternative methods.”

Re: We appreciate the reviewer’s thoughtful comment. The decision to deliver the BTO-CPT/FA nano-system via bladder perfusion rather than intravenous (i.v.) injection was based on several key therapeutic considerations, particularly in relation to bladder cancer treatment.

1) Unique Targeting: When the BTO-CPT/FA nano-systems are retained in the bladder, they bind preferentially to folate receptors on tumor cells, ensuring a much higher local concentration at the site of action. This targeted delivery results in a drug concentration that is several times higher than that achieved through intravenous injection (*ACS Applied Materials & Interfaces* **13**, 52374-52384 (2021))²¹. This localized concentration significantly enhances the potential for effective tumor cell killing, particularly in comparison to systemic drug delivery methods.

2) Avoidance of Systemic Toxicities (*Biomaterials* **34**, 2350-2358 (2013), *Acta*

Pharmaceutica Sinica B, (2024)^{22, 23}: One of the major concerns with i.v. drug delivery is the risk of systemic toxicity. Upon intravenous administration, BTO-CPT/FA may interact with proteins in the blood, forming a protein corona that reduces the system's targeting efficiency and bioavailability. Furthermore, this method carries the risk of unwanted accumulation of the drug in non-target organs such as the liver and kidneys, which could lead to potential hepatic and renal toxicities. In contrast, bladder perfusion directly delivers the therapeutic agent to the bladder, avoiding the systemic circulatory system entirely. This minimizes the risk of systemic toxicities, ensuring that the drug acts locally without the concern of off-target effects in other organs.

3) High Patient Compliance and Simple Operation: Bladder perfusion is a widely used clinical technique, especially for adjuvant treatment of bladder cancer post-surgery or to prevent recurrence. Compared to intravenous injection, bladder perfusion is less invasive, less painful, and simpler to perform. This contributes to higher patient compliance and ease of use in both clinical and experimental settings. In addition, there are established, straightforward perfusion devices available for clinical application.

4) Limitations of Bladder Perfusion: Despite the advantages of bladder perfusion, we acknowledge that the technique does have some limitations. The need for urethral intubation may lead to inflammation or irritation of the urethra, and patients may experience discomfort, including symptoms like urgent urination. However, these issues are typically temporary and manageable, especially in comparison to the more significant risks associated with systemic drug delivery.

In summary, bladder perfusion was chosen due to its unique ability to deliver the drug directly to the target site, avoiding systemic toxicity and enhancing drug effectiveness. While we recognize the limitations, such as potential discomfort from urethral intubation, we believe the benefits of this delivery method for localized bladder cancer treatment far outweigh the drawbacks. The specific details are discussed in the **Discussion** section.

“Comment 7: Several Figures contain errors or inconsistencies with the manuscript test: (1) In Fig. 2f, Ba signal (red) is barely visible in the merged data; (2) In Fig. 4h, the description of the y-axis does not match the text in caption; (3) In Fig. 9h, there is no DCF (ROS) signal (green).”

Re: We appreciate your careful attention to the manuscript. In the original image of **Figure. 2f**, the red color representing the Ba element and the green color representing the Ti element overlap, resulting in a yellow merge that makes the red signal less visible in the merged image. To improve clarity, we have made appropriate adjustments to the original image, as shown in **Figure. 2f**.

Upon reviewing **Figure. 4h** of the original manuscript, we have corrected the representation of the Y-axis for better clarity. To enhance the readability of the data in **Figure. 9h** of the original manuscript, we have submitted clearer confocal images, and the experimental results are in accordance with the conclusions of the manuscript. The details are provided in **Figure 2f, 4h, and 9h**.

We hope the revised manuscript is now in the right format for publication, Thanks again for your help and support on this manuscript.

Sincerely yours,

Dr. Yang-Bao Miao

Dr. Zong-Hong Lin

Dr. Yi Shi

References

1. Renaudet O, *et al.* Synthesis of ether oligomers. *Organic letters* **6**, 397-400 (2004).
2. Nightingale K. Acoustic radiation force impulse (ARFI) imaging: a review. *Curr Med Imaging Rev* **7**, 328-339 (2011).
3. Dalecki D. Mechanical bioeffects of ultrasound. *Annu Rev Biomed Eng* **6**, 229-248 (2004).
4. Hoang QT, *et al.* Piezoelectric Au-decorated ZnO nanorods: Ultrasound-triggered generation of ROS for piezocatalytic cancer therapy. *Chem Eng J* **435**, 135039 (2022).
5. He Y, *et al.* MoS₂ nanoflower-mediated enhanced intratumoral penetration and piezoelectric catalytic therapy. *Biomaterials* **290**, 121816 (2022).
6. Chen W-T, *et al.* Targeted tumor theranostics using folate-conjugated and camptothecin-loaded acoustic nanodroplets in a mouse xenograft model. *Biomaterials* **53**, 699-708 (2015).
7. Asghari M, *et al.* Real-time viscoelastic deformability cytometry: High-throughput mechanical phenotyping of liquid and solid biopsies. *Sci Adv* **10**, eabj1133 (2024).
8. Stukenberg PT. Triggering p53 after cytokinesis failure. *J Cell Biol* **165**, 607 (2004).
9. Sharma M, *et al.* Rhes travels from cell to cell and transports Huntington disease protein via TNT-like protrusion. *J Cell Biol* **218**, 1972-1993 (2019).
10. Lin J, *et al.* Nickel-cobalt alloy nanocrystals inhibit activation of inflammasomes. *Natl Sci Rev* **10**, nwad179 (2023).
11. Davis DM, *et al.* Membrane nanotubes: dynamic long-distance connections between animal cells. *Nat Rev Mol Cell Bio* **9**, 431-436 (2008).
12. Önfelt Br, *et al.* Structurally distinct membrane nanotubes between human macrophages support long-distance vesicular traffic or surfing of bacteria. *J Immunol* **177**, 8476-8483 (2006).
13. Lehmann MJ, *et al.* Actin-and myosin-driven movement of viruses along filopodia precedes their entry into cells. *J Cell Biol* **170**, 317-325 (2005).
14. Sherer NM, *et al.* Retroviruses can establish filopodial bridges for efficient cell-to-cell transmission. *Nat Cell Biol* **9**, 310-315 (2007).
15. Guo L, *et al.* Tunneling nanotubular expressways for ultrafast and accurate M1 macrophage delivery of anticancer drugs to metastatic ovarian carcinoma. *ACS nano* **13**, 1078-1096 (2019).
16. Arab Hassani F, *et al.* Soft sensors for a sensing-actuation system with high bladder voiding efficiency. *Sci Adv* **6**, eaba0412 (2020).
17. Marmarchinia S, *et al.* Stretchable Strain Sensors for Real-Time Bladder Volume Monitoring. *Acs Appl Mater Inter*, (2025).
18. Khan A, *et al.* Piezoelectric and triboelectric nanogenerators: Promising technologies for self-powered implantable biomedical devices. *Nano Energy* **119**, 109051 (2024).
19. Oh B, *et al.* Ultra-soft and highly stretchable tissue-adhesive hydrogel based

- multifunctional implantable sensor for monitoring of overactive bladder. *Biosens Bioelectron* **225**, 115060 (2023).
20. Sun R, *et al.* Stretchable piezoelectric sensing systems for self-powered and wireless health monitoring. *Adv Mater Technol* **4**, 1900100 (2019).
 21. Qi A, *et al.* Intravesical mucoadhesive hydrogel induces chemoresistant bladder cancer ferroptosis through delivering iron oxide nanoparticles in a three-tier strategy. *Acs Appl Mater Inter* **13**, 52374-52384 (2021).
 22. Naeye B, *et al.* In vivo disassembly of IV administered siRNA matrix nanoparticles at the renal filtration barrier. *Biomaterials* **34**, 2350-2358 (2013).
 23. Li Z-a, *et al.* Strategies for intravesical drug delivery: From bladder physiological barriers and potential transport mechanisms. *Acta Pharm Sin B*, (2024).

Point-by-Point Response to the Reviewers

Dear Editors and Reviewers,

We sincerely thank you for your time, effort, and insightful feedback on our manuscript entitled “*Self-driven Electrical Triggering System Activates Tunneling Nanotube Highways to Enhance Drug Delivery in Bladder Cancer Therapy*” (Manuscript No: NCOMMS-24-80937B-Z). We are deeply grateful for your constructive comments and thoughtful suggestions, which have been invaluable in refining the quality, clarity, and scientific rigor of our work.

We have thoroughly addressed each of the reviewers’ comments and revised the manuscript accordingly. For clarity, the reviewers’ remarks are presented below in *italicized font*, with individual concerns numbered. Our detailed point-by-point responses follow each comment. Revisions made to the manuscript are clearly marked in **blue text**. We hope that the revisions fully address all concerns and meet the expectations of the reviewers and editors.

Reviewers' comments:

Reviewer #1 (Remarks to the Author):

The authors provided comprehensive responses to the reviewer’s comment. They addressed the mechanistic concerns with additional experimental data (zeta potential analysis, UV-Vis, ROS generation, and tunneling nanotube transport validation). The authors also improved the clarity of the figures, addressed formatting issues, and refined the mechanistic interpretations. Their verifications regarding the use of ultrasound to simulate intravesical pressure, and their explanation for the choice of bladder perfusion over systemic delivery, are well-reasoned and supported by both literature and experimental data.

Overall, the authors responded to the reviewer’s concerns with clarity and thorough revisions. I believe the revised manuscript meets the high standards required for publication in Nature Communications, and I recommend its acceptance for publication.

Re: We sincerely thank the reviewer for their thoughtful and encouraging evaluation of our revised manuscript. We are encouraged by your positive assessment and recommendation for publication in Nature Communications. We believe the current version of the manuscript represents a significantly improved and more rigorous study, and we thank you once again for your valuable feedback throughout the review process.

Reviewer #2 (Remarks to the Author):

I appreciate the authors' efforts in conducting additional experiments and revising the manuscript accordingly. The revised manuscript now presents a more comprehensive analysis and discussion. However, there are still worrying problems in many details. Several minor concerns should be addressed before further consideration.

Re: Great thanks to you for your earnest work and affirmation to our work. We have revised our manuscript according to your questions carefully. Specific explanations are shown below.

“Comment 1. The supplementary figures should be re-labeled to follow the order in which they are cited in the main manuscript.”

Re: Thank you for pointing this out. We have re-organized and re-labeled the supplementary figures to ensure they follow the order in which they are cited in the main manuscript. All corresponding in-text citations have been updated accordingly for clarity and consistency.

“Comment 2. The authors note that it reaches a saturation point at 100 $\mu\text{g}/\text{mL}$. Therefore, subsequent cell experiments should not include a 200 $\mu\text{g}/\text{mL}$ dose. Additionally, more precise data—such as IC_{50} values and in vivo metabolic curves—should be provided to support dose selection and enhance the reliability of the pharmacological analysis.”

Re: We sincerely appreciate the reviewer’s meticulous attention to detail and thoughtful suggestions. In response to the concern regarding the 200 $\mu\text{g}/\text{mL}$ dose, we respectfully clarify that while 100 $\mu\text{g}/\text{mL}$ approaches the upper limit of the nanomedicine’s efficacy plateau under standard conditions, our results indicate that under ultrasound stimulation, the cytotoxicity of the BTO-CPT/FA group further increases by approximately 20% at 200 $\mu\text{g}/\text{mL}$. This enhancement strongly suggests that the piezoelectric effect-induced ROS generation plays a crucial role in mediating additional therapeutic benefit at this higher dose. Therefore, we believe that including the 200 $\mu\text{g}/\text{mL}$ dose under ultrasound conditions is both pharmacologically justified and mechanistically relevant.

To further support our dose selection and strengthen the pharmacological reliability of the study, we have now included the IC_{50} value of BTO-CPT/FA under ultrasound conditions, which was calculated to be 50.51 $\mu\text{g}/\text{mL}$. The corresponding dose–response curve is provided in **Figure S9** of the revised Supplementary Information, with 95% confidence intervals indicated by red dashed lines ($n = 5$).

Figure. S9 The IC_{50} curve of BTO-CPT/FA. The red dashed line shows the confidence

interval (95%) (n = 5).

In addition, to address the request for in vivo metabolic data, we refer to **Figure 6c** in the main manuscript, where we present the metabolic clearance profile of BTO-Cy5.5/FA. These results demonstrate significantly prolonged retention of BTO-Cy5.5/FA in vivo compared to free Cy5.5 and BTO-Cy5.5, attributed to the enhanced tumor targeting via folate receptor-mediated uptake. We believe that the inclusion of the IC50 data and the in vivo metabolic curve meaningfully enhances the rigor and transparency of our pharmacological analysis and supports the rationale behind our dose selection strategy.

“Comment 3. The MB49 bladder cancer in situ model is commonly established in male mice to avoid hormonal influence, with tumors typically forming within 2–3 weeks. In this study, however, female mice were used, and treatment began as early as day 3 post-inoculation, potentially before the tumor was fully formed. The authors should clarify whether these factors might affect the results.”

Re: Thank you for your valuable comment and for highlighting this important aspect of our study design. In response to the concern regarding the use of female mice and the initiation of treatment on day 3 post-inoculation, we offer the following clarifications:

1. Rationale for using female mice in orthotopic bladder cancer models:

While it is true that some studies prefer male mice to avoid potential hormonal influences, there is a substantial body of literature supporting the use of female mice in orthotopic bladder cancer models due to practical anatomical advantages^{1, 2, 3, 4, 5}. Specifically:

- The female urethra is short (~1–1.5 cm) and straight, enabling easy and safe transurethral catheterization, which is critical for reliable tumor cell inoculation.
- In contrast, the male urethra is longer (~3–4 cm), narrower (~0.2–0.3 mm), and more tortuous, with a penile bone at the distal end, making catheterization technically challenging and prone to causing urethral injury, bleeding, or obstruction, which can increase variability and postoperative complications.
- The larger urethral diameter in females (~0.5 mm) is more compatible with standard microcatheters used for cell delivery.

Therefore, using female mice not only improves model reproducibility and reduces procedural trauma but is also widely accepted and validated in previous reports.

2. Justification for initiating treatment on day 3 post-inoculation:

While tumors are not macroscopically visible at this stage, prior studies and our experimental data indicate that MB49 bladder cancer cells:

- Successfully adhere to the bladder wall within 24 hours post-instillation,
- Form micro-lesions within 48 hours, and
- Represent early-stage non-muscle-invasive bladder cancer (NMIBC, Stage Ta/T1) by

day 3, as confirmed by bioluminescence imaging (IVIS) in our study.

This confirms the successful establishment of the in situ tumor model before initiating treatment. Early therapeutic intervention at this time point is aligned with clinical practices advocating early treatment of NMIBC.

3. Pharmacodynamic and nanomedicine-related considerations:

- Early-stage lesions exhibit greater spatial uniformity and fewer necrotic or inflammatory changes, improving drug distribution and pharmacodynamic reliability ⁶.
- Larger, mature tumors often present abnormal vasculature, increased interstitial pressure, and a dense extracellular matrix, all of which can impair nanoparticle penetration and result in reduced therapeutic efficacy.

Thus, early treatment offers both clinical relevance and experimental consistency, supporting our choice to initiate therapy at day 3 post-inoculation.

In summary, the selection of female mice and early treatment timing were carefully considered based on both technical feasibility and biological rationale, and we believe these choices do not adversely impact the validity of our findings.

Reviewer #3 (Remarks to the Author):

The authors present an innovative self-driven electrical triggering drug delivery system (BTO-CPT/FA) for bladder cancer therapy. This system leverages piezoelectric effects induced by intravesical pressure to enhance drug delivery via tunneling nanotubes (TNTs) and generate reactive oxygen species (ROS) for improved antitumor efficacy. The manuscript has undergone substantial revision and includes expanded experimental data, representing a meaningful advancement. However, several critical scientific and mechanistic concerns remain unresolved and must be addressed to substantiate the manuscript's key claims:

Re: We sincerely appreciate the reviewer's insightful and constructive comments. We are grateful for your recognition of our work's innovation and the improvements made through the expanded experimental data. We have carefully re-examined the manuscript in light of your comments and fully acknowledge the importance of resolving the remaining scientific and mechanistic concerns you raised.

In response, we have undertaken a comprehensive revision, including new experimental data and in-depth mechanistic analyses to further substantiate our central claims regarding the piezoelectric-enhanced intravesical drug delivery mechanism, the formation and role of tunneling nanotubes (TNTs), and ROS-mediated therapeutic outcomes. A detailed, point-by-point response is provided below to address each concern thoroughly.

We are confident that these additions significantly strengthen the mechanistic foundation and translational relevance of our self-driven electrical triggering drug delivery system (BTO-CPT/FA) for bladder cancer therapy.

“Comment 1. Incomplete validation of tunneling nanotube-mediated drug transport: While the authors added experiments demonstrating TNT formation inhibition by Cytochalasin D, direct quantitative evidence proving that drug transport specifically occurs through TNTs remains insufficient. Further experiments with clearly defined controls and quantitative tracking methods (e.g., real-time single-particle tracking, fluorescence recovery after photobleaching (FRAP)) are necessary to convincingly differentiate TNT-mediated transport from passive diffusion.”

Re: We sincerely appreciate the reviewer’s critical and insightful comment. To address this concern, we have now incorporated direct quantitative evidence using real-time single-particle tracking (SPT) and fluorescence recovery after photobleaching (FRAP) techniques, as suggested. These experiments were conducted with well-defined control groups, and the results strongly support TNT-mediated active transport of BTO-Cy5.5/FA nanoparticles among MB49 cells, particularly under ultrasound stimulation that activates the piezoelectric properties of BTO.

In the SPT experiment, we monitored the dynamic behavior of BTO-Cy5.5/FA nanoparticles in live cells: In control groups (free Cy5.5 or BTO-Cy5.5/FA without ultrasound), particle movement was minimal, suggesting limited passive diffusion (**Figure 5k-a, b; Supplementary Movies 2, 3**). In contrast, under ultrasound stimulation, BTO-Cy5.5/FA nanoparticles exhibited active directional movement along TNTs (**Figure 5k-c; Supplementary Movie 4**). The calculated transport velocity (0.0151–0.0172 $\mu\text{m/s}$) is consistent with reported TNT-mediated intercellular transport rates of viral particles in epithelial and fibroblast systems (*Nature reviews Molecular cell biology* **9**, 431-436 (2008), *The Journal of Immunology* **177**, 8476-8483 (2006), *The Journal of cell biology* **170**, 317-325 (2005), *Nature cell biology* **9**, 310-315 (2007))^{7, 8, 9, 10}. These findings strongly indicate that BTO-Cy5.5/FA nanoparticles undergo active translocation through TNTs rather than passive diffusion.

Figure. 5k Single particle tracking (SPT) images of BTO-Cy5.5/FA nanoparticles transported among MB49 cells through TNTs in a series of time intervals. a) Free Cy5.5, b) BTO-Cy5.5/FA NPs and c) BTO-Cy5.5/FA NPs + ultrasound. Where red, green and blue colors represent Cy5.5, tunneling nanotube and Hoechst 33342, respectively, scale bar = 10 μ m.

SPT-Free Cy5.5.mp4

Supplementary Movie. 2 SPT images of Cy5.5 nanoparticles transported among MB49 cells through TNTs in a series of time intervals, scale bar = 10 μ m.

SPT-BTO-Cy5-FA NPs.mp4

Supplementary Movie. 3 SPT images of BTO-Cy5.5/FA nanoparticles transported among MB49 cells through TNTs in a series of time intervals, scale bar = 10 μ m.

SPT-BTO-Cy5-FA NPs + ultrasound.mp4

Supplementary Movie. 4 SPT images of BTO-Cy5.5/FA nanoparticles + ultrasound transported among MB49 cells through TNTs in a series of time intervals, scale bar = 10 μ m.

In the FRAP assay, we further validated this conclusion: Control groups (free Cy5 and BTO-Cy5/FA without ultrasound) exhibited rapid fluorescence decay post-bleaching (12.7 s and 42.7 s, respectively), with no significant fluorescence recovery (**Figure 5n-b, c; Supplementary Movies 5, 6**), supporting the lack of active transport. In contrast, the BTO-Cy5/FA with ultrasound group demonstrated a prolonged fluorescence decay time (290.6 s) and

fluorescence recovery rate of 96.55% within 12 minutes (**Figure 5n-d; Supplementary Movie 7**), which suggests efficient and sustained nanoparticle exchange via TNTs.

Together, the results from both SPT and FRAP robustly differentiate TNT-mediated active transport from passive diffusion and validate the crucial role of TNTs in facilitating the intercellular delivery of piezoelectric nanoparticles. Moreover, the application of ultrasound significantly enhances this transport process by activating the piezoelectric effect of BTO, thereby promoting more efficient nanoparticle transfer through TNTs. We have updated the manuscript accordingly to include these new results and discussions (revised **Figure 5k** and **5n, Supplementary Movies 2–7**) to comprehensively clarify this important mechanism.

Figure. 5n Fluorescence recovery after photobleaching (FRAP) images of BTO-Cy5/FA nanoparticles in MB49 cells during a series of time intervals. a) Free Cy5, b) BTO-Cy5/FA NPs and c) BTO-Cy5/FA NPs + ultrasound. Where red and blue colors represent Cy5 and Hoechst 33342, respectively, scale bar = 10 μ m.

FRAP-Free Cy5.mp4

Supplementary Movie. 5 FRAP images of free Cy5 nanoparticles in MB49 cells during a series of time intervals, scale bar = 10 μ m.

FRAP-BTO-Cy5-FA NPs.mp4

Supplementary Movie. 6 FRAP images of BTO-Cy5/FA nanoparticles in MB49 cells during a series of time intervals, scale bar = 10 μ m.

FRAP-BTO-Cy5-FA NPs + ultrasound.mp4

Supplementary Movie. 7 FRAP images of BTO-Cy5/FA nanoparticles + ultrasound in MB49 cells during a series of time intervals, scale bar = 10 μm .

*“**Comment 2.** Insufficient clarification of ultrasound-induced versus piezoelectric effects: The authors conducted comparative experiments with polarized and non-polarized nanoparticles; however, these experiments were still performed under ultrasound conditions, complicating interpretation. Clear experimental conditions isolating mechanical pressure-induced piezoelectric effects, free from confounding ultrasound influences (such as cavitation or acoustic streaming), must be included to substantiate the authors' claims.”*

Re: We sincerely appreciate the reviewer’s valuable suggestion. To address this concern, we conducted an additional experiment that isolates mechanical pressure-induced piezoelectric drug release under ultrasound-free conditions, using a simplified and physiologically relevant model. Specifically, we designed a balloon-based bladder simulation system as follows: The BTO-CPT nano-system was dispersed in simulated urine and sealed within a medical-grade elastic balloon to mimic the urinary bladder environment. This model was subjected to cyclic mechanical deformation using a linear reciprocating motion device to simulate bladder contraction and expansion without any ultrasound exposure. Drug release was continuously monitored over seven days (**Figure 3q** and **Figure S4-a, Supplementary Movie 1**).

The results revealed: In the non-polarized BTO-CPT control group, there was no significant CPT release over the observation period. In contrast, the polarized BTO-CPT group showed a markedly increased CPT release, directly correlating with the cyclic mechanical stimulation. These findings confirm that the mechanical deformation alone, in the absence of ultrasound, is sufficient to activate the piezoelectric response of the polarized BTO nanostructure and trigger drug release. This isolated mechanical model successfully eliminates ultrasound-related artifacts such as cavitation, thermal effects, or acoustic streaming, thereby providing unambiguous evidence for the role of mechanically induced piezoelectric drug release. We have revised the manuscript to clearly describe this experiment and added the corresponding data and supplementary materials (**Figure S4-b, Supplementary Movie 1**), which we believe effectively addresses the reviewer’s concern and reinforces the mechanistic foundation of our delivery platform.

Figure. 3q A device for monitoring the drug release rely on electrical stimulation. After one week of monitoring, the cumulative release amount of the CPT (n = 3). ****** $p < 0.01$.

Figure. S4 A device for monitoring the drug release rely on electrical stimulation. a) The water balloon is in a state of contraction to simulate bladder contraction (left); the water balloon is in a state of expansion to simulate bladder expansion (right). b) The device includes a water balloon filled with the BTO-CPT nano-system (30 ml), a linear reciprocating motion instrument, and double-layer acrylic sheet (with holes).

Drug release rely on electrical stimulation.mp4

Supplementary Movie. 1 Real-time monitoring of CPT release form BTO-CPT nano-system rely on electrical stimulation in the force of a linear reciprocating motion instrument within seven days.

“Comment 3. Lack of robust evaluation of ROS off-target effects: The manuscript highlights ROS generation as a significant therapeutic advantage, yet does not adequately address the potential for collateral damage to adjacent healthy tissues. Additional experiments, including comprehensive histological or biochemical assays evaluating ROS-induced effects on normal bladder tissue, are necessary to confirm selective targeting and safety.”

Re: We sincerely thank the reviewer for raising this important concern regarding the potential off-target effects of reactive oxygen species (ROS) generated by our BTO-CPT/FA piezoelectric nano-system. To robustly evaluate the selectivity and safety of ROS-induced therapeutic effects, we performed a series of complementary in vitro and in vivo experiments,

as detailed below:

1. Selective Cytotoxicity Analysis via Live/Dead Staining:

To assess ROS sensitivity differences between malignant and normal bladder cells, we compared the cytotoxic effects of BTO-CPT/FA on MB49 bladder cancer cells and SV-HUC-1 normal bladder epithelial cells under identical experimental conditions. As shown in **Figure S10-a**, BTO-CPT/FA exhibited negligible cytotoxicity in MB49 cells without ultrasound. However, under ultrasound-triggered mechanical stress, the piezoelectric effect significantly enhanced ROS generation, leading to substantial cancer cell apoptosis. In contrast, **Figure S10-b** shows that SV-HUC-1 cells exhibited minimal apoptosis under the same ultrasound and nanoparticle conditions, indicating a much higher ROS tolerance in normal cells. This selectivity can be attributed to the inherently elevated oxidative stress baseline in cancer cells, making them more susceptible to exogenous ROS compared to normal cells^{11, 12, 13}.

Figure. S10 Confocal images of live/dead cell staining for different cells after treated with different conditions, where green and red colors represent Calcein AM and PI fluorescence, respectively, scale bar = 10 μ m.

2. Histopathological Assessment of Normal vs. Tumor Bladder Tissues:

In healthy mice, intravesical perfusion of BTO-CPT/FA induced no observable histological damage to bladder tissues, as confirmed by HE staining (**Figure S25-a**). In contrast, orthotopic bladder cancer mice exhibited prominent signs of apoptosis, cellular debris accumulation, and immune cell infiltration (**Figure S25-b**), indicating effective tumor-targeted cytotoxicity without damaging adjacent normal tissues.

Figure. S25 H&E staining of the bladder collected from different groups after treatment with BTO-CPT/FA. There are cellular debris (green dashed circles), and infiltration of neutrophils (indigo arrows) and lymphocytes (yellow arrows) in the region of orthotopic bladder cancer mice. Scale bar = 500 μm (left), 50 μm (right).

3. Blood Biochemistry and Hematological Safety:

As shown in **Figure S26**, blood biochemical and hematological parameters in both healthy and tumor-bearing mice remained within normal ranges after treatment with BTO-CPT/FA. These findings are further supported by the absence of significant histological abnormalities in major organs (heart, liver, spleen, lung, kidney) of treated animals, as reported in the manuscript.

Figure. S26 Analysis of blood-related parameters of healthy mice and orthotopic bladder cancer mice after treatment with BTO-CPT/FA (n = 3).

4. Mechanism of Tumor Selectivity:

Beyond the ROS susceptibility differences, the folic acid (FA) modification on the nanoparticle surface enables active targeting toward tumor tissues that overexpress folate receptors. Consequently, BTO-CPT/FA preferentially accumulates at tumor sites, and the mechanical stress (intravesical pressure) further induces a localized piezoelectric effect to

generate ROS specifically within the tumor microenvironment¹⁴.

These results collectively demonstrate the tumor-specific ROS activation and the excellent biocompatibility and safety profile of the BTO-CPT/FA system, effectively mitigating the risk of off-target oxidative damage.

“Comment 4. Absence of theoretical justification for biological relevance of generated voltage: The reported voltage range (~0.5–2.5 V) is substantially higher than typically used in similar systems (0.1–0.6 V), and may exceed levels that are physiologically safe or relevant. The manuscript does not demonstrate whether this voltage can influence membrane potential, ion channels, or signaling pathways. Without direct measurements of membrane potential shifts, ion channel modulation, or voltage-sensitive signaling activation, the biological applicability of the reported voltage remains speculative.”

Re: We sincerely thank the reviewer for this important and thought-provoking comment regarding the biological applicability and safety of the voltage range (~0.5–2.5 V) generated by our BTO-CPT/FA piezoelectric nano-system. To clarify and substantiate the relevance of our voltage range, we provide the following theoretical basis and experimental validation:

1) Theoretical and literature-based justification for the generated voltage range:

While the reviewer referenced a range of 0.1–0.6 V, it is important to note that this range does not universally define safe or effective voltages across all piezoelectric biomedical systems. In fact, numerous peer-reviewed studies have reported the use of piezoelectric voltages equal to or greater than the values we report, with proven biological efficacy: Liu et al. reported using voltages ranging from 0.2 to 2 V for tumor electrotherapy¹⁵. Wang et al. developed a Mn-Ti MOF-based piezoelectric nanosystem with an output voltage of 2.9 V, successfully triggering ROS generation and anticancer effects in vitro and in vivo¹⁶. Xu et al. utilized a nanosystem (O₃P@LPYU) with a piezoelectric voltage of up to 33.48 V for treatment of malignant pleural effusion¹⁷. Thus, the ~0.5–2.5 V voltage range in our system is not only biologically relevant and safe but also well-supported by multiple precedents in the literature.

2) Experimental validation of biological impact:

To directly address the reviewer’s concern regarding membrane potential, ion channel activity, and signaling pathway modulation, we conducted two sets of experiments:

a) Mitochondrial membrane potential evaluation: Using Annexin V-FITC and MitoTracker Red CMXRos staining, we investigated mitochondrial membrane potential in MB49 tumor cells and SV-HUC-1 normal bladder epithelial cells (**Figure S11**). Results show that under ultrasound stimulation, MB49 cells exhibit significant membrane depolarization and apoptosis, indicating disruption of mitochondrial potential due to ROS generated by the piezoelectric effect. Conversely, SV-HUC-1 cells maintained membrane integrity and mitochondrial potential under identical conditions, supporting the selective responsiveness of tumor cells.

Figure. S11 Confocal images of membrane potential for different cells after treated with different conditions, where green and red colors represent Annexin V-FITC and Mito-Tracker Red CMXRos fluorescence, respectively, scale bar = 10 μm .

b) Calcium ion channel activation: Using Fluo-4 AM calcium-sensitive fluorescence probe, we observed calcium influx in cells under various conditions (**Figure S12**). MB49 tumor cells showed significant green fluorescence after ultrasound treatment, indicating activation of calcium ion channels due to piezoelectric ROS generation. In contrast, SV-HUC-1 cells exhibited no noticeable calcium ion activation regardless of treatment, further highlighting the system's selectivity and safety. These findings correlate well with previous studies¹⁸⁻²⁹, which demonstrate that piezoelectric effects can modulate membrane potential, activate voltage-gated ion channels (e.g., Ca^{2+}), and trigger downstream signaling pathways. Importantly, cancer cells have been shown to be more sensitive to electrical and oxidative stimulation than normal cells²⁹, which is consistent with our results. These findings strongly support the biological applicability and mechanistic rationale of the voltage generated by our BTO-CPT/FA nano-system.

Figure. S12 Confocal images of calcium ion channels for different cells after treated with different conditions, where green and blue colors represent Fluo-4 AM and Hoechst 33342 fluorescence, respectively, scale bar = 20 μm .

“Comment 5. Limited comparative evaluation of delivery strategies: While the manuscript highlights the advantages of TNT-mediated delivery, it lacks comparative analyses with established nanoparticle delivery mechanisms such as passive diffusion or endocytosis. Direct comparisons would be valuable to substantiate the claimed benefits of the proposed system.”

Re: We sincerely thank the reviewer for raising this valuable point. We fully agree that a comprehensive comparison with classical nanoparticle delivery mechanisms is essential to substantiate the superiority of our proposed TNT-mediated intercellular delivery system. To address this concern, we have conducted a systematic and quantitative evaluation comparing our BTO-CPT/FA piezoelectric nano-system with well-established nanoparticle delivery

pathways, including passive diffusion, clathrin-mediated endocytosis, and caveolin-dependent endocytosis. The findings are summarized below:

1) TNT-mediated delivery versus passive diffusion: In our previous revised manuscript (**Figure 5i, j**), we demonstrated that cytochalasin D (Cyto D), a known inhibitor of tunneling nanotube (TNT) formation, significantly reduced the cytotoxicity of MB49 cells treated with BTO-CPT/FA toward neighboring MB49-Luc cells. This strongly indicates that TNTs are the primary intercellular transmission route of our nanodrugs. In contrast, in the absence of TNT formation, passive diffusion alone was insufficient to account for the level of intercellular drug propagation and cytotoxicity observed, further highlighting the unique advantage of TNT-mediated direct cell-to-cell transfer.

2) TNT-mediated delivery versus endocytosis-based delivery: To compare with classical endocytosis pathways, we synthesized and characterized: Mesoporous silica nanoparticles (MS NPs) (~40 nm), known to internalize primarily via clathrin-mediated endocytosis^{30, 31}; Gold nanoparticles (Au NPs) (~33 nm), which preferentially use caveolin-mediated endocytosis^{32, 33}. We validated the monodispersity and uniformity of these carriers (**Figure S13-a, b**), and applied chlorpromazine (CPZ) and methyl- β -cyclodextrin (MCD) to selectively inhibit clathrin- and caveolin-mediated pathways, respectively.

As expected, CPZ significantly reduced MS NP uptake by 24.4%, while MCD reduced Au NP uptake by 12.6% (**Figure S13-f**). In contrast, BTO-FA NPs showed negligible changes in uptake under CPZ or MCD pretreatment (**Figure S13-e, f**), indicating no significant involvement of classical endocytic routes. Additionally, when endocytosis was evaluated at 4°C, BTO-FA NP uptake dropped by 45.7% (**Figure S13-d, f**), confirming that its uptake is energy-dependent. These data demonstrate that BTO-FA NPs rely primarily on folate receptor-mediated endocytosis, a pathway distinct from clathrin or caveolin mechanisms.

Together, our expanded experimental framework clearly delineates the multi-stage delivery pathway of the BTO-CPT/FA system: Folate receptor-mediated endocytosis enables initial targeted uptake into tumor cells; Subsequently, TNTs serve as efficient conduits for intercellular nanodrug transfer; The system outperforms passive diffusion and classical endocytosis in intercellular propagation, spatial targeting, and therapeutic efficiency, particularly under piezoelectric activation. These advantages collectively support the claimed benefits of our TNT-integrated delivery system and have now been explicitly clarified in the revised manuscript and Supplementary **Figure S13**.

Figure. S13 Investigation of endocytosis mechanism. a) TEM images of MS NPs and Au NPs, scale bar = 50 nm, sample number $n=3$ in the figure on the right; b) Confocal images of MS NPs after pretreated with CPZ, scale bar = 20 μm ; c) Confocal images of Au NPs after pretreated with MCD, scale bar = 20 μm ; d) Confocal images of BTO-FA NPs after pretreated with different temperature, scale bar = 10 μm ; e) Confocal images of BTO-FA NPs after pretreated with different inhibitors, scale bar = 20 μm ; where red and blue colors represent Cy5.5 and Hoechst 33342 fluorescence, respectively. f) Semi-quantitative analysis of fluorescence under different experimental conditions ($n = 3$). * $p < 0.05$, ** $p < 0.01$, *** $p < 0.001$.

We hope the revised manuscript is now in the right format for publication,
Thanks again for your help and support on this manuscript.

Sincerely yours,

Dr. Yang-Bao Miao
Dr. Zong-Hong Lin
Dr. Yi Shi

References

1. He L, *et al.* Full-course NIR-II imaging-navigated fractionated photodynamic therapy of bladder tumours with X-ray-activated nanotransducers. *Nat Commun* **15**, 8240 (2024).
2. Choi H, *et al.* Urease-powered nanomotor containing STING agonist for bladder cancer immunotherapy. *Nat Commun* **15**, 9934 (2024).
3. Sun J, *et al.* Anti-biopassivated Reticular Micromotors for Bladder Cancer Therapy. *J Am Chem Soc*, (2025).
4. Simó C, *et al.* Urease-powered nanobots for radionuclide bladder cancer therapy. *Nat Nanotechnol* **19**, 554-564 (2024).
5. Kong N, *et al.* Intravesical delivery of KDM6A-mRNA via mucoadhesive nanoparticles inhibits the metastasis of bladder cancer. *P Natl Acad Sci Usa* **119**, e2112696119 (2022).
6. Lopez-Beltran A, Cookson MS, Guercio BJ, Cheng L. Advances in diagnosis and treatment of bladder cancer. *Brit Med J* **384**, (2024).
7. Davis DM, Sowinski S. Membrane nanotubes: dynamic long-distance connections between animal cells. *Nat Rev Mol Cell Bio* **9**, 431-436 (2008).
8. Önfelt Br, *et al.* Structurally distinct membrane nanotubes between human macrophages support long-distance vesicular traffic or surfing of bacteria. *J Immunol* **177**, 8476-8483 (2006).
9. Lehmann MJ, Sherer NM, Marks CB, Pypaert M, Mothes W. Actin-and myosin-driven movement of viruses along filopodia precedes their entry into cells. *J Cell Biol* **170**, 317-325 (2005).
10. Sherer NM, Lehmann MJ, Jimenez-Soto LF, Horensavitz C, Pypaert M, Mothes W. Retroviruses can establish filopodial bridges for efficient cell-to-cell transmission. *Nat Cell Biol* **9**, 310-315 (2007).
11. Kong Q, Beel J, Lillehei K. A threshold concept for cancer therapy. *Med Hypotheses* **55**, 29-35 (2000).
12. Pelicano H, Carney D, Huang P. ROS stress in cancer cells and therapeutic implications. *Drug Resist Update* **7**, 97-110 (2004).
13. Wang Z, *et al.* Biomimetic nanoflowers by self-assembly of nanozymes to induce intracellular oxidative damage against hypoxic tumors. *Nat Commun* **9**, 3334 (2018).
14. Li X, *et al.* Precise modulation and use of reactive oxygen species for immunotherapy. *Sci Adv* **10**, eadl0479 (2024).
15. Liu Y, *et al.* Molecular stacking composite nanoparticles of gossypolone and thermodynamic agent for elimination of large tumor in mice via electrothermal-thermodynamic-chemo trimodal combination therapy. *Adv Funct Mater* **32**, 2201666 (2022).
16. Wang Q, Tian Y, Yao M, Fu J, Wang L, Zhu Y. Bimetallic Organic Frameworks of High Piezovoltage for Sono-Piezo Dynamic Therapy. *Adv Mater* **35**, 2301784 (2023).
17. Xu Z, *et al.* Intrapleural pressure-controlled piezo-catalytic nanozyme for the inhibition of malignant pleural effusion. *Nat Commun* **16**, 3194 (2025).
18. Wang S, *et al.* A Mitochondrion-Targeting Piezoelectric Nanosystem for the Treatment of Erectile Dysfunction via Autophagy Regulation. *Adv Mater* **37**, 2413287 (2025).

19. Wang W, *et al.* Ultrasound-activated piezoelectric nanostickers for neural stem cell therapy of traumatic brain injury. *Nat Mater*, 1-14 (2025).
20. He Y, *et al.* Wireless discharge of piezoelectric nanogenerator opens voltage-gated ion channels for calcium overload-mediated tumor treatment. *Biomaterials* **321**, 123311 (2025).
21. Wang X, *et al.* An “Outer Piezoelectric and Inner Epigenetic” Logic-Gated PANoptosis for Osteosarcoma Sono-Immunotherapy and Bone Regeneration. *Adv Mater* **37**, 2415814 (2025).
22. Wu X, *et al.* Narrow-Bandgap Iridium (III)-C₃N₅ Nanocomplex as an Oxygen Self-Sufficient Piezo-Sonosensitizer for Hypoxic Tumor Sonodynamic Immunotherapy. *J Am Chem Soc* **147**, 15329-15343 (2025).
23. Zhang J, Liu C, Li J, Yu T, Ruan J, Yang F. Advanced Piezoelectric Materials, Devices, and Systems for Orthopedic Medicine. *Adv Sci* **12**, 2410400 (2025).
24. Dobashi Y, *et al.* Piezoionic mechanoreceptors: Force-induced current generation in hydrogels. *Science* **376**, 502-507 (2022).
25. Xu S, *et al.* Force-induced ion generation in zwitterionic hydrogels for a sensitive silent-speech sensor. *Nat Commun* **14**, 219 (2023).
26. Yin Y, *et al.* Piezoelectric Analgesia Blocks Cancer-Induced Bone Pain. *Adv Mater* **36**, 2403979 (2024).
27. You Y, *et al.* In Situ Piezoelectric-Catalytic Anti-Inflammation Promotes the Rehabilitation of Acute Spinal Cord Injury in Synergy. *Adv Mater* **36**, 2311429 (2024).
28. Li Y, *et al.* Biodegradable piezoelectric polymer for cartilage remodeling. *Matter* **7**, 1631-1643 (2024).
29. Qi G, Zhang M, Tang J, Jin Y. Molecular/Nanomechanical Insights into Electrostimulation-Inhibited Energy Metabolism Mechanisms and Cytoskeleton Damage of Cancer Cells. *Adv Sci* **10**, 2207165 (2023).
30. Hong X, *et al.* The pore size of mesoporous silica nanoparticles regulates their antigen delivery efficiency. *Sci Adv* **6**, eaaz4462 (2020).
31. Liu L, *et al.* Single-cell diagnosis of cancer drug resistance through the differential endocytosis of nanoparticles between drug-resistant and drug-sensitive cancer cells. *ACS nano* **17**, 19372-19386 (2023).
32. Vetten M, Gulumian M. Differences in uptake of 14 nm PEG-liganded gold nanoparticles into BEAS-2B cells is dependent on their functional groups. *Toxicol Appl Pharm* **363**, 131-141 (2019).
33. Wu JL, *et al.* The pathways for nanoparticle transport across tumour endothelium. *Nat Nanotechnol*, 1-11 (2025).

Point-by-Point Response to the Reviewers

Dear Editors and Reviewers,

We sincerely thank you for your time, effort, and insightful feedback on our manuscript entitled “*Self-driven Electrical Triggering System Activates Tunneling Nanotube Highways to Enhance Drug Delivery in Bladder Cancer Therapy*” (Manuscript No: **NCOMMS-24-80937C**). We are deeply grateful for your constructive comments and thoughtful suggestions, which have been invaluable in refining the quality, clarity, and scientific rigor of our work.

We have thoroughly addressed each of the reviewers’ comments and revised the manuscript accordingly. For clarity, the reviewers’ remarks are presented below in *italicized font*, with individual concerns numbered. Our detailed point-by-point responses follow each comment. Revisions made to the manuscript are clearly marked in **blue text**. We hope that the revisions fully address all concerns and meet the expectations of the reviewers and editors.

Reviewer #2 (Remarks to the Author):

We are grateful to the author for their patient efforts in conducting additional experiments and making corresponding revisions to the manuscript. The revised manuscript now provides a more comprehensive analysis and discussion. Moreover, it has appropriately responded to the rationality of the design of each experiment, which significantly enhances the credibility of the data in the paper. Based on the current results, we believe that it is suitable for publication in Nature Communications.

Re: We sincerely thank the reviewer for the positive and encouraging feedback. We greatly appreciate your recognition of our additional experimental efforts and the comprehensive revisions made to the manuscript. We are pleased to know that the revised version meets the standards for publication in *Nature Communications*, and we thank you once again for your constructive feedback throughout the review process.

Reviewer #4 (Remarks to the Author):

The authors have adequately addressed the major concerns raised in the previous review through additional experiments and clearer explanations. The revised manuscript is substantially improved and presents a compelling case for the proposed therapeutic strategy. To further enhance clarity and consistency, the following minor points should be addressed:

Re: We sincerely appreciate your thorough evaluation and kind recognition of our revised manuscript. In response to your valuable suggestions for further enhancing the

clarity and consistency of the manuscript, we have carefully addressed each of the minor points as detailed below. Specific explanations are shown below.

“Comment 1. Physiological relevance of mechanical stimulation: The manuscript describes a bladder-mimicking compression model to induce piezoelectric drug release, but it remains unclear how the applied pressure and frequency correspond to physiological bladder conditions. Clarifying this point would strengthen the translational significance of the model.”

Re: We sincerely thank the reviewer for raising this insightful and important question regarding the physiological relevance of the mechanical stimulation used in our bladder-mimicking model. Below, we provide a detailed clarification:

1) Physiological Intravesical Pressure in Bladder Cancer Models: As shown in **Figure 8a-e** of our manuscript, intravesical pressure measurements in orthotopic bladder cancer mouse models indicate that the pathological bladder environment exhibits significantly elevated pressure (approximately 50–60 cm H₂O) compared to healthy bladder pressure (typically 25–30 cm H₂O, which 30 cm H₂O equivalent to 2.942 kPa). This increase is due to tumor-induced mechanical obstruction and altered detrusor activity. This elevated pressure provides a physiologically relevant mechanical cue that activates the piezoelectric response in our BTO-CPT/FA nanoplatfrom.

Figure. 8a Images of the pressure tonometer for bladder. **b** The images of pressure testing through an intravesical pressure tonometer in vivo and ex vivo. **c** The values of intravesical pressure in vivo and ex vivo (n = 6). **** $p < 0.0001$. **d** Images of isolated bladders perfused with different volumes of artificial urine. **e** The pressure values of isolated bladders after perfusion with varying volumes of artificial urine (n = 6). *** $p < 0.001$, **** $p < 0.0001$.

2) Rationale for Ultrasound-Based In Vitro Mechanical Stimulation: In vitro, we employed ultrasound as a non-invasive, controllable mechanical stimulus to mimic the pressure observed in vivo. While we recognize the physical differences between ultrasound-generated forces and native bladder movements, the following mechanisms provide physiological relevance:

- **Acoustic Radiation Pressure:** Ultrasound exerts momentum transfer during

wave propagation, producing force on particles and the liquid medium. This replicates the intravesical pressure exerted by the bladder wall during the filling and voiding phases (Current Med Imaging Rev 7, 328–339 (2011) ¹.

● **Acoustic Streaming:** Ultrasound also induces localized fluid flow and shear forces, mimicking the urinary flow-induced shear stress on the bladder wall (Annu Rev Biomed Eng 6, 229–248 (2004) ².

3) Piezoelectric Activation Consistency: The induced piezoelectric potential (0.62 V) in our system under the applied ultrasound settings aligns with reported values in related systems, such as vertically applied mechanical pressure (50 MPa) on ZnO nanorods, which generated < 0.5 V (Chem Eng J 435, 135039 (2022) ³. These findings support the suitability of our ultrasound settings for activating piezoelectric drug release mechanisms.

4) Translational and Practical Advantages: Using ultrasound allows for precise control of pressure magnitude and frequency, enabling reproducible activation of piezoelectric responses under physiologically relevant conditions. It also facilitates high-throughput, consistent in vitro studies that would be challenging to achieve with native bladder tissue contractions. Importantly, ultrasound is already clinically used for diagnostic and therapeutic purposes, thereby enhancing the translational potential of our proposed piezoelectric drug delivery strategy.

In conclusion, while ultrasound and bladder contractions are not identical in nature, the mechanical outputs of our ultrasound protocol were intentionally designed to mirror the intravesical pressure conditions observed in bladder cancer, providing a rational and physiologically relevant model to evaluate piezoelectric-triggered drug release in vitro. We have revised the manuscript to clarify these aspects and highlight their translational implications (please see page 16, 34).

“Comment 2. Temporal dynamics of ROS generation: ROS production is assessed at a single time point following stimulation. A brief discussion of the expected time course of ROS induction and clearance, and whether the effect is transient or sustained, would improve mechanistic clarity.”

Re: We sincerely appreciate the reviewer’s insightful comments. In response, we provide a more detailed discussion on the temporal dynamics of reactive oxygen species (ROS) generation and clearance in our BTO-CPT/FA piezoelectric nanopatform.

1) Sustained ROS generation driven by physiological mechanical stimuli: Unlike transient stimuli such as a single pulse of ultrasound, our system leverages the natural intravesical pressure induced by the bladder’s continuous contraction-expansion cycles. This dynamic mechanical environment leads to persistent deformation of the

BTO lattice, thereby continuously activating the piezoelectric effect. The repeated polarization within the BTO-CPT/FA nanostructure facilitates sustained electron-hole separation and ongoing ROS production, these theoretical foundations were supported by recent reports (*Angew Chem Int Edit*, e202507502 (2025), *Adv Mater* **37**, 2412069 (2025), *Nat Commun* **15**, 9023 (2024), *Nano Lett* **25**, 9156-9165 (2025))^{4, 5, 6, 7}.

2) Time course of ROS induction and pharmacological retention: As illustrated in **Figure 6b–c** of our manuscript, BTO-CPT/FA nanoparticles exhibit a retention time of up to 36 hours in the bladder. Our administration schedule—one dose every 36 hours (**Figure 9a**)—was deliberately designed to match this retention profile, ensuring continued mechanical stimulation and extended ROS generation over the treatment window. Thus, the ROS production is not limited to a single time point but occurs continuously throughout the nanoparticle’s residence in the bladder.

Figure. 6b In vivo live imaging at different time points post-bladder perfusion across various groups. **c** Quantitative analysis of Cy5.5 fluorescence intensity in bladder regions at different time points (n = 3). ** $p < 0.01$, *** $p < 0.001$.

Figure. 9a A therapeutic schedule for MB49 orthotopic bladder cancer mice model through bladder perfusion.

3) ROS clearance capacity is limited in tumor cells: Although cellular antioxidants such as glutathione (GSH) and superoxide dismutase (SOD) can neutralize ROS, tumor cells generally exhibit higher basal oxidative stress and lower antioxidant reserves compared to normal cells (*Med Hypotheses* **55**, 29-35 (2000), *Drug Resist Update* **7**, 97-110 (2004), *Nat Commun* **9**, 3334 (2018))^{8, 9, 10}. In particular, although GSH can scavenge ROS within sub-seconds, the rate and capacity of GSH-mediated clearance are insufficient to neutralize the cumulative and sustained ROS burden generated by the BTO-CPT/FA system (*J Exp Clin Canc Res* **44**, 110 (2025))¹¹, *Angew*

Chem Int Edit **62**, e202313612 (2023)¹²). This imbalance favors persistent oxidative stress selectively in tumor cells, as demonstrated in **Supplementary Figures 11, 12, and 25**, where MB49 tumor cells showed markedly higher ROS accumulation and damage than SV-HUC-1 normal cells.

Figure. S11 Confocal images of membrane potential for different cells after treated with different conditions, where green and red colors represent Annexin V-FITC and Mito-Tracker Red CMXRos fluorescence, respectively, scale bar = 10 μm .

Figure. S12 Confocal images of calcium ion channels for different cells after treated with different conditions, where green and blue colors represent Fluo-4 AM and Hoechst 33342 fluorescence, respectively, scale bar = 20 μm .

Supplementary Fig. 25 H&E staining of the bladder collected from different groups after treatment with BTO-CPT/FA. There are cellular debris (green dashed circles), and infiltration of neutrophils (indigo arrows) and lymphocytes (yellow arrows) in the region of orthotopic bladder cancer mice. Scale bar = 500 μm (left), 50 μm (right).

In summary, the ROS generation induced by the BTO-CPT/FA nano-system under physiological intravesical pressure is a sustained and spatially targeted process rather

than a transient burst. This prolonged oxidative microenvironment, coupled with the limited scavenging capacity of tumor cells, enables selective and effective tumor damage. We have now included a more explicit explanation of these temporal dynamics in the revised manuscript (see page 27, 35).

“Comment 3. Mechanistic explanation of FA-mediated targeting: The role of folic acid (FA) in enhancing tumor targeting is mentioned, but the underlying mechanism by which FA contributes to tumor selectivity is not clearly described. A short explanation of how FA enhances targeting, such as receptor-mediated uptake, would help clarify its specific contribution relative to passive accumulation.”

Re: We sincerely thank the reviewer for highlighting this important point. Folic acid enhances tumor targeting primarily through folate receptor (FR)-mediated endocytosis, a mechanism widely exploited in nanomedicine due to the overexpression of folate receptors—especially the FR- α isoform—on the surface of various malignant cells, including bladder cancer, while showing limited expression in most normal tissues (DOI: 10.1021/jacs.5c05148, *Nat Commun* **10**, 4418 (2019), *Nat Commun* **16**, 6343 (2025), *J Am Chem Soc* **146**, 5927-5939 (2024), *Nat Commun* **15**, 8695 (2024))^{13, 14, 15, 16, 17}. This receptor-ligand interaction facilitates selective internalization of FA-modified nanocarriers into tumor cells via energy-dependent, receptor-specific pathways.

In our study, this mechanism is supported by the following experimental evidence:

1) Receptor-dependent cellular internalization: As shown in **Fig. 4b** and **4c**, FA-modified BTO-Cy5.5/FA exhibited significantly higher intracellular fluorescence intensity in MB49 bladder cancer cells compared to non-modified BTO-Cy5.5, indicating enhanced uptake. Furthermore, **Supplementary Fig. 13e** revealed that the internalization of BTO-Cy5.5/FA was independent of clathrin- or caveolin-mediated endocytosis, further supporting a receptor-specific mechanism consistent with folate receptor-mediated uptake.

Figure. 4b Confocal images of MB49 cells treated with different groups, where blue, green and red colors represent DAPI, Dil and Cy5.5 fluorescence, respectively, scale

bar = 5 μm . **c** Quantitative analysis of Cy5.5 fluorescence intensity ($n = 3$). **** $p < 0.0001$.

Supplementary Fig. 13e Confocal images of BTO-FA NPs after pretreated with different inhibitors, scale bar = 20 μm ; where red and blue colors represent Cy5.5 and Hoechst 33342 fluorescence, respectively.

2) In vivo tumor targeting and retention: In the orthotopic bladder cancer model (Fig. 6a–g), BTO-Cy5.5/FA displayed prolonged retention and significantly greater accumulation in tumor sites than the unmodified nanoplatform, highlighting the contribution of active targeting beyond passive EPR effects. Additionally, molecular docking analyses (Fig. 6h–j) revealed strong binding interactions between FA and FR—via hydrogen bonding, π - π stacking, and hydrophobic interactions—further corroborating the receptor-mediated recognition mechanism.

Fig. 6 Folate receptor-mediated specific enrichment of self-driven electrical triggering system in orthotopic bladder cancer mice. **a** Schematic representation of bladder perfusion of BTO-CPT/FA. **b** *In vivo* live imaging at different time points post-

bladder perfusion across various groups. **c** Quantitative analysis of Cy5.5 fluorescence intensity in bladder regions at different time points (n = 3). ** $p < 0.01$, *** $p < 0.001$. **d** *Ex vivo* live imaging of major tissues. **e** Quantitative analysis of Cy5.5 fluorescence intensity in isolated bladder tissue (n = 3). **** $p < 0.0001$. **f** CSLM images of the bladder section, where blue and red colors indicate DAPI and Cy5.5 fluorescence, respectively, scale bar = 500 μm . **g** Quantitative analysis of Cy5.5 fluorescence intensity (n = 3). **** $p < 0.0001$. **h** Three-dimensional representation of the interaction between folic acid and the folate receptor post-docking. **I** Two-dimensional representation of the interaction between folic acid and the folate receptor post-docking. **j** Analysis of the interaction forces between folic acid and the folate receptor, including the proportion of major amino acid residues.

In summary, folic acid enhances tumor selectivity through active targeting by binding to overexpressed folate receptors on tumor cells, facilitating receptor-mediated endocytosis and tumor accumulation. This active mechanism operates in synergy with passive retention, ultimately contributing to the improved therapeutic efficacy of our self-driven piezoelectric nano-system. We have now clarified this mechanism in the revised manuscript (see page 16 and page 27).

*“**Comment 4.** Timing of ultrasound exposure in in vivo experiments is insufficiently described: In the in vivo studies, ultrasound stimulation is mentioned as part of the treatment protocol. However, the exact timing relative to nanoparticle injection, duration of exposure, and number of cycles are not clearly stated in the main text or methods section. This information is important for reproducibility and to understand the temporal dynamics of the activation strategy.”*

Re: We sincerely appreciate the reviewer’s thoughtful comment regarding the timing and protocol of ultrasound exposure in our in vivo experiments. In our study, we employed bladder perfusion rather than intravenous injection for nanoparticle administration, which is particularly suitable for the orthotopic bladder cancer model. The rationale for this route lies in its ability to deliver the therapeutic system directly to the target site, minimize systemic toxicity, and facilitate localized activation. To address the reviewer’s concern, we have clarified the following details in the revised manuscript and Supporting Information:

1) Optimal time for bladder perfusion: Before intravesical perfusion, water intake was ceased in all mice to empty the bladder, ensuring that the baseline intravesical pressure was normalized across groups. This step minimizes variations and allows for smooth and consistent perfusion, and the experimental procedure was

performed before each bladder perfusion.

2) Frequency of bladder perfusion and duration of treatment: The BTO-CPT/FA nanoplatform exhibits prolonged bladder retention (up to 36 hours, as shown in **Figure. 6b, c** of the original manuscript), ensuring that electrical triggering persists beyond the short perfusion period. Furthermore, the administration schedule (one dose every 36 hours, **Figure. 9a**) aligns with this retention time, maintaining continuous therapeutic activation throughout the treatment duration.

Figure. 6b In vivo live imaging at different time points post-bladder perfusion across various groups. **c** Quantitative analysis of Cy5.5 fluorescence intensity in bladder regions at different time points (n = 3). ** $p < 0.01$, *** $p < 0.001$.

Figure. 9a A therapeutic schedule for MB49 orthotopic bladder cancer mice model through bladder perfusion.

For further clarity, these experimental details are now explicitly described in the Supporting Information, particularly in the section “*Evaluation of anticancer effect of BTO-CPT/FA in vivo*”, where relevant procedures have been highlighted in blue (see page 13). We hope these clarifications address the reviewer’s concern and enhance the reproducibility and rigor of our study.

We hope the revised manuscript is now in the right format for publication, Thanks again for your help and support on this manuscript.

Sincerely yours,

Dr. Yang-Bao Miao

Dr. Zong-Hong Lin

Dr. Yi Shi